

# Assessment of Arctic Sea Ice Thickness Retrieval Ability of the Chinese HY-2B Radar Altimeter

Zhaoqing Dong[1,2], Lijian Shi[2,3], Mingsen Lin[2,3], Tao Zeng[2,3], Suhui Wu[2]

[1]Hohai University, Nanjing, 210003, China
[2]National Satellite Ocean Application Service, Beijing, 100081, China
[3]Key Laboratory of Space Ocean Remote Sensing and Application (MNR), Beijing, 100081, China

*Correspondence to*: Lijian Shi (shilj@mail.nsoas.org.cn)

**Abstract.** In the context of global warming, sea ice changes have received increasing attention as "indicators" and "amplifiers" of climate change. With the development of satellite altimeters, satellite altimeter technologies have been
increasingly used to retrieve Arctic sea ice thicknesses and have achieved rapid development and application. At present, the CryoSat-2 radar altimeter and Ice, Cloud and land Elevation Satellite-2 (ICESat-2) laser altimeter are the main data sources used in Arctic sea ice thickness retrievals. With the continuous development of the China Ocean Dynamic Environment Satellite Series (HY-2), it is of great significance to explore the potential application of this dataset in Arctic sea ice thickness retrievals. In this study, we first estimated the Arctic radar freeboard and sea ice thickness values during two sea
ice growing cycles (from October 2019 to April 2020 and from October 2020 to April 2021) using the China HY-2B radar altimeter and then compared the results with the Alfred Wegener Institute (AWI) CryoSat-2 sea ice freeboard and sea ice thickness products recorded during the same period. The accuracies of the HY-2B radar freeboard and sea ice thickness were then verified with the Operation IceBridge (OIB) airborne data and ICESat-2 laser altimeter data, and the random uncertainties in the HY-2B sea ice freeboard and sea ice thickness results were finally estimated. Although the spatial
distributions of the HY-2B radar freeboard and sea ice thickness results agreed well with those of AWI CryoSat-2, the deviation between the HY-2B radar freeboard and CryoSat-2 radar freeboard data was within 2 cm, while the deviation between the HY-2B sea ice thickness data and CryoSat-2 sea ice thickness data was within 0.2 m. In addition, the growth trends of the HY-2B radar freeboard and sea ice thickness were slower than those of AWI CryoSat-2. This finding was related to the applied sea surface height anomaly (SSHA) extraction method. Comparisons with the OIB sea ice freeboard
and sea ice thickness values recorded in April 2019 showed that the correlation between the HY-2B sea ice freeboard retrievals and OIB sea ice freeboard data was 0.58, the root mean square error (RMSE) was 0.17 m, and the mean absolute error (MAE) was 0.14 m. The correlation between the HY-2B sea ice thickness retrieval and OIB sea ice thickness data was 0.41, the RMSE was 2.05 m, and the MAE was 1.91 m. Based on the Gaussian error propagation theory, we estimated the uncertainties of the HY-2B sea ice freeboard and sea ice thickness data: the uncertainty of the former ranged from 8.5 cm to
12.0 cm, while the uncertainty of the latter ranged from 26.8 cm to 37.7 cm. Due to the influence of the SSHA uncertainty ($\sigma_{SSA}$) and the number of observation points inside the grid, the uncertainties in the HY-2B sea ice freeboard and sea ice thickness data were higher at low latitudes than at high latitudes.



## 1 Introduction

Arctic sea ice is an important factor of the global climate system and plays an important role in maintaining its energy
balance. By reflecting most of the solar shortwave radiation, sea ice reduces the absorption of solar shortwave radiation by
seawater and blocks outward longwave radiation from leaving the ocean, thus regulating the overall radiation budget of the
Earth. Sea ice also regulates the exchanges of heat, momentum and water vapour between the polar atmosphere and oceans
(Thomas, D. N et al., 2010; Xu et al., 2017). Due to the special air-ice-sea feedback mechanism, the Arctic has exhibited
warming temperatures at more than twice the global average increasing rate. This phenomenon is known as the "Arctic
amplification effect" (Serreze et al., 2009). Studies have shown that global warming has led to decreases in the extent and
thickness of Arctic sea ice and that the ice age of multiyear ice has gradually decreased (Comiso et al., 2008; Lindell et al.,
2016; Kwok, 2018; IPCC, 2019; Kacimi et al., 2022; Meier et al., 2022). Models predict that the Arctic will be ice-free in
summer by the middle of the 21st century (Notz et al., 2020). As sea ice melts, large amounts of fresh water will be injected
into the deep convection area of the North Atlantic Ocean, reducing the strength of the Atlantic meridional overturning
circulation (AMOC) and thus transporting heat northwards towards the Arctic; in turn, this will lead to persistent anomalies
in the sea ice extent (Liu et al., 2018; Halloran et al., 2020). The predicted decrease in Arctic sea ice will also change the
living environment of Arctic mammals, and these changes will not be conducive to the survival or development of Arctic
mammals, such as polar bears and walruses (IPCC, 2019). Due to the rapid retreat of sea ice, trans-Arctic shipping routes
have become increasingly navigable (Stephenson et al., 2015; Cao et al., 2022). At the same time, the reduction in Arctic sea
ice has improved the convenience of exploiting natural resources in the Arctic, and these activities will have an important
impact on the economy of the Arctic and on regions beyond the Arctic.

Sea ice thickness, as the third dimension of sea ice, can be combined with sea ice density to calculate sea ice volume to
better understand changes in sea ice. However, sea ice thickness is also a difficult parameter to measure. The recent
development of satellite altimeters has made it possible to obtain sea ice thickness data over continuous and large ranges. To
date, the available international altimeter satellites that obtain polar sea ice thickness observations include the European
Remote Sensing Satellite 1 (ERS)-1, ERS-2, Envisat, Ice, Cloud and land Elevation Satellite (ICESat), CryoSat-2 (CS-2),
Saral, Sentinel-3A, Sentinel-3B and ICESat-2 (IS-2). Laxon et al. (2003) estimated Arctic sea ice thickness for the first time
with the ERS-1/2 altimeter and verified their findings with submarine sonar data, thus confirming the feasibility of using
satellite altimeters to retrieve sea ice thickness data. Kwok et al. (2004) estimated the Arctic sea ice thickness for the first
time in 2004 using the Geoscience Laser Altimeter System (GLAS) on the ICESat satellite, further demonstrating the
advantage of altimeter data in estimating Arctic sea ice thicknesses. Giles et al. (2008) estimated the Arctic sea ice thickness
using the Envisat altimeter and analysed its variation pattern in winter from 2002 to 2007; the authors found that the area
where the sea ice thickness showed a decreasing and thinning trend was mainly in the Beaufort Sea. Tilling et al. (2016)
released near-real-time CS-2 sea ice thickness products with time periods of 2, 14 and 28 days. Also based on CS-2 data, R.
Ricker et al. (2014) set threshold ranges for the pulse peak (PP), stack standard deviation (SSD) and stack kurtosis (K) terms





to separate the lead, sea ice and open water components, compared and analysed the effects of different retracking thresholds on the sea ice thickness, and estimated the uncertainties of the sea ice freeboard and sea ice thickness. Shen et al. (2020) retrieved Arctic sea ice freeboard data based on Sentinel-3A records and analysed the differences and consistency between the results and the corresponding CS-2 data. The results showed that the Sentinel-3A sea ice freeboard was generally lower

than that indicated by CS-2. Petty et al. (2020) generated gridded monthly IS-2 sea ice thickness products and compared them with various monthly sea ice thickness estimates obtained from the European Space Agency (ESA)'s CS-2 satellite mission, with IS-2 showing consistently lower thicknesses. With the continuous progress of Arctic sea ice remote sensing technologies, a wide variety of sea ice thickness products have become available to the scientific community (Sallila et al., 2019). CS-2 radar altimeters, ICESat and IS-2 laser altimeters cover almost the entire Arctic Ocean due to their large orbital

inclinations and are thus the main data sources for estimating sea ice thicknesses. However, few reports have explored the retrieval of sea ice thickness by Chinese altimeters among recent studies of polar sea ice thickness. With the continuous development of China's Marine Dynamic Environment Satellite (Haiyang-2B, HY-2B), the HY-2B satellite can be used as a supplementary means to observe polar sea ice. Therefore, it is of great significance to obtain reliable Arctic sea ice thickness products based on the HY-2B radar altimeter to provide data support for the study of long-term changes in Arctic sea ice

thickness.

In this study, we used the HY-2B radar altimeter to retrieve Arctic radar freeboard and sea ice thickness data and compared the results with the CS-2 data released by the Alfred Wegener Institute (AWI) during the same period. Finally, we compared the results with Operation IceBridge (OIB) airborne data and IS-2 laser altimeter data. In Section 2, we introduce the data used in this study. In Section 3, we introduce the determination method of the sea surface height anomaly (SSHA) and the

retrieval process of sea ice thickness in detail. In Section 4, we compare the Arctic HY-2B radar freeboard and sea ice thickness data with AWI CS-2 data and IS-2 data. In Section 5, we discuss the influence of different SSHA determination schemes on the HY-2B radar freeboard results and estimate the uncertainties in the HY-2B radar freeboard and sea ice thickness results. Finally, in Section 6, we summarize the conclusions.

## 2 Study area and data

### 2.1 HY-2B radar altimeter

The HY-2B satellite was successfully launched on October 25, 2018. It is China's second polar-orbiting marine dynamic environmental satellite and the second marine operational satellite in China's civil space infrastructure program. Its main mission is to monitor and survey the marine environment, obtain a variety of marine dynamic environmental parameters, including sea surface winds, wave heights, sea surface heights, sea surface temperatures and other elements, and take into

account the observation of sea ice. The HY-2B satellite integrates both active and passive microwave remote sensors and carries loads such as a radar altimeter, microwave scatterometer, scanning microwave radiometer, correction radiometer, ship identification system and data collection system. The HY-2B radar altimeter is a dual-band pulse-limited radar altimeter



comprising the Ku band and C band. The main parameters of the HY-2B radar altimeter are shown in Table 1 (Jiang et al., 2019; National Satellite Ocean Application Service, 2019).

The National Satellite Ocean Application Service (NSOAS) has released level-1, level-2 and fusion data products compiled through the preprocessing, data-retrieval and statistical averaging of the HY-2B altimeter level-0 data. The level-2 products are divided into Interim Geophysical Data Records (IGDR), Sensing Geophysical Data Records (SGDR) and Geophysical Data Records (GDR). Because the SGDR product contains waveform data and applies a retrack algorithm, the HY-2B SGDR data were used to retrieve Arctic sea ice thicknesses in this study. The spatial coverage of the HY-2B SGDR data in

April 2020 is shown in Fig. 1.

## 2.2 CryoSat-2 radar altimeter

CS-2 was launched by the ESA in April 2010 with an orbital altitude of approximately 717 km, an orbital inclination of 92° and a repeat cycle period of 369 days. It has a 30-day subcycle and can realize monthly observations of the Arctic with a coverage of 88°N/S. CS-2 carries a Ku-band synthetic aperture interferometric radar altimeter (SIRAL) that can obtain the

surface elevations of ground objects. Compared to conventional radar altimeters, CS-2 can achieve monthly observations of the Arctic with a coverage range of 88°N/S. This SIRAL uses delayed Doppler radar altimeter technology to reduce the satellite observation footprint to approximately 0.3 km along-track and 1.5 km across-track.

Currently, there are four main kinds of CS-2 sea ice thickness products: those from the ESA, the Centre for Polar Observation and Modeling (CPOM) (Laxon et al., 2003; Tilling et al., 2017), the AWI (Ricker et al., 2014; Hendricks et al.,

2020) and the National Snow and Ice Data Center (NSIDC) (Kurtz et al., 2014; Kurtz et al., 2017). These products are constructed using different sea ice freeboard retrieval methods. We mainly used the level-2 L2I along-track data published by the ESA and the monthly average sea ice thickness product published by the AWI.

## 2.3 ICESat-2 laser altimeter

The Advanced Terrain Laser Altimeter System (ATLAS) onboard IS-2 is a low-pulse energy laser (operating wavelength:

532 nm) that uses photon-counting technology to emit pulses at a repetition rate of 10 kHz (Degnan, 2002). The photon detector accurately calculates the round-trip time of these photons from the satellite to the ground and back to obtain distance measurements. We used the snow freeboard data of ATL20 products in the study (version 003 (Petty et al., 2021)); these products were by the National Aeronautics and Space Administration (NASA). The ATL20 snow freeboard was calculated by subtracting the local sea surface height (SSH) from the sea ice elevation. The average value of the specular reflected

elevation of the inter-ice channel collected in the 10-km segment where the measurement point was located was used as the SSH estimation value (Kwok, Petty, Bagnardi et al., 2021). The 10-km segments were selected to minimize the impact of the sea surface slope on the sea ice freeboard height estimations, as SSHs are generally constant within 10-km segments in polar regions north of 60°N. If SSH data were not available within a segment, the total freeboard estimate was not provided, thus assuring the reliability of the total freeboard estimates. Finally, the total freeboard height was gridded into a 25-km spatial



grid, and the average value of the total freeboard height of all observation points in the grid was used as the total freeboard height of that grid.

Assuming hydrostatic equilibrium, Kacimi et al. (2022) used the IS-2 ATL10 snow freeboard product (version 004 (Kwok et al., 2021)), combined with the Arctic snow depths retrieved by obtaining the difference between IS-2 and CS-2 on the ice-snow reflection interface (Kwok et al., 2020), to retrieve Arctic monthly average sea ice thickness products. We used these

products to evaluate the sea ice thicknesses retrieved from HY-2B and CS-2.

### 2.4 OIB airborne data

The airborne OIB experiment is an aerial remote sensing polar-region observation project started by NASA in 2009. Its initial purpose is to compensate for the data gaps that arise during the operation of ICESat and IS-2 satellites and to carry out large-scale sea ice detection experiments in the Arctic from March to May and in the Antarctic from October to November

every year. The flight path of the OIB in the Arctic in April 2019 is shown in Fig. 2. In this study, we used IceBridge L4-level data (IDCSI4) to evaluate the sea ice freeboard and sea ice thickness data retrieved by HY-2B and CS-2. In addition, we gridded the OIB data to a 25-km polar stereographic grid and set no fewer than 100 observation points inside each grid to optimally solve the limited representation problem of the OIB data.

### 2.5 Auxiliary data

In this study, we used auxiliary data, including data representing the sea ice concentration (SIC), sea ice type, mean sea surface (MSS) height, snow depth and snow density. The SIC (version OSI-401-b) and sea ice type (version OSI-403-b) data were released by the European Organization for Meteorological Satellites (EUMETSAT) Ocean and Sea Ice Satellite Application Facility (OSI-SAF). The MSS data were released by the Denmark Technical University (DTU).

### 2.5.1 Sea ice concentration

Rasmus Tonboe et al. (2016) used the brightness temperatures of the 19-V, 37-V and 37-H channels in the Special Sensor Microwave-Imager/Sounder (SSMIS) scanning radiometer to retrieve SICs with a hybrid algorithm constructed from the Bristol algorithm and bootstrap algorithm. To ensure optimum performances over both marginal and consolidated ice and to retain the virtues of each algorithm, the Bristol algorithm is given low weights at low concentrations, while the opposite is the case for high-ice-concentration regions (Rasmus et al., 2016). The SIC data are provided as a daily average grid product

with the 10-km Lambert azimuthal grid. We used these SIC data to screen the altimeter data, and altimeter observations corresponding to areas with SICs greater than 70% were used in the sea ice freeboard calculations.

### 2.5.2 Sea ice type

We used sea ice type data to distinguish first-year ice (FYI) from multiyear ice (MYI). Signe et al. (2021) used the gradient ratio (GR) of 19/37 in Advanced Microwave Scanning Radiometer 2 (AMSR-2) microwave radiometer data and the



scattering coefficient in the Advanced Scatterometer (ASCAT) microwave data to calculate the ice type probability. The sea
        ice type data are provided as a daily average grid product with a 10-km Lambert azimuthal grid.

### 2.5.3 MSS height

In this study, we employed the DTU18 MSS model to eliminate errors due to unresolved gravity features, intersatellite biases
and remaining satellite orbit errors; this model can precisely determine the instantaneous elevation of lead (Skourup et al.,
2017). The DTU18 MSS model is fused with the data of several satellite altimeters, such as TOPEX/Poseidon (T/P), Jason-1
        (J1), Jason-2 (J2), ERS-1, ERS-2, ENVISAT, ICESat, Geosat, Geosat Follow-On (GFO) and CryoSat-2 (Andersen et al.
        2018a; Andersen et al. 2018b).

### 2.5.4 Snow depth

Hendricks et al. (2020) obtained a composite snow depth product (hereafter referred to as the AWI snow depth product) by
fusing W99 climatology snow depths with the daily average AMSR-2 snow depths of the University of Bremen. To merge
        these two datasets, the authors created a monthly average AMSR-2 snow depth product to match the W99 climatology snow
        depths from October to April. They then low-pass filtered the monthly average AMSR-2 snow depths with a Gaussian filter
        with a size of 8 grid cells, removed negative snow depth values and limited the upper range to 60 cm. Finally, they created a
        regional weighting factor to ensure a smooth transition between the two types of data in the borderline area. Since the W99
climatology snow depths on FYI are higher, they had to be corrected by a coefficient of 0.5 (Kwok et al. 2015). However,
        the AMSR-2 snow depths on FYI did not need to be modified, so the authors introduced a total scaling factor to correct the
        contribution of W99 (Hendricks et al., 2020). The AWI snow depth data are provided as a monthly averaged grid product in
        which the Equal Area Scalable Earth Grid version 2 (EASE2) is used for the Northern Hemisphere with a spatial resolution
        of 25 km.
Kwok et al. (2020) used the freeboard differences measured by IS-2 (by measuring the height between the local sea level and
        the air-snow interface) and CS-2 (by measuring the height between the local sea level and the snow-sea ice interface) to
        invert the snow depth data in the Arctic basin from October 2018 to April 2019 (hereafter referred to as the Kwok snow
        depths). We gridded the Kwok snow depth data to the 25-km polar stereographic grid with a temporal resolution of 1 month.
        The valid region of these Kwok snow depth data is the Arctic Ocean, defined as the region bounded by the gateway into the
Pacific Ocean (the Bering Strait), the Canadian Arctic Archipelago (CAA), Greenland (Flam Strait) and the Barents Sea
        (Kwok et al., 2020).

### 2.5.5 Snow density

To minimize differences in sea ice thicknesses at the beginning of the sea ice growing season, we used the evolving snow
density values proposed by Mallett et al. (2020); these values are consistent with the snow densities used in the AWI CS-2
sea ice thickness product. The specific snow density values are shown in Table 2.





## 3 Method

In this section, we describe the sea ice thickness retrieval method applied for the SGDR data of the HY-2B pulse-limited radar altimeter in detail and compare the parameters involved in the retrieval process with those in the CS-2 L2I data released by the ESA.

**3.1 Retrieval process**

The technical process of retrieving sea ice thickness based on HY-2B SGDR data is shown in Fig. 3. The specific retrieval process is as follows:

(1) Invalid values in the SGDR data and data south of 60°N were eliminated. Due to the influence of instrument noise, atmospheric factors and tidal factors during the propagation of pulse signals, it was necessary to consider the dry and wet

tropospheric delay correction, inverse barometric correction, ionospheric correction, ocean tidal correction, equilibrium tidal correction, ocean load tidal correction, earth tidal correction and polar tidal correction when calculating the surface elevation.

(2) The SICs of the data points in all HY-2B orbits were interpolated using the SIC data; in this process, the altimeter observations corresponding to areas with SICs greater than 70% were used in the sea ice freeboard calculations. Sea ice was classified into FYI, MYI and ambiguous ice using the sea ice type data, and ambiguous ice was not used for the subsequent

sea ice thickness retrievals.

(3) The geoid fluctuations were eliminated by subtracting the MSS height from the obtained surface elevations of ground objects, that is, the derived relative elevations of ground objects $h$ (Ollivier et al., 2012; Zhang et al., 2021). The residual of sea surface height was eliminated by subtracting the average value of every 25 km ($h_{25km}$) from the observation data every 25 km along the track (Kwok et al., 2007; Zhang et al., 2021), as shown in Eq. (1). The relative surface elevation, $h_r$, was

obtained after eliminating the residuals, as shown in Fig. 4(a) and (b). Eq. (1) can be expressed as follows:

$$h_r = h - h_{25km}, \tag{1}$$

where $h_r$ is the relative surface elevation after eliminating residuals, $h$ is the relative elevation of ground objects, and $h_{25km}$ is the average value every 25 km.

(4) If more than or equal to 9 observation points were available per 25 km in the track data, the average of the 9 lowest

values was taken as the SSHA. Otherwise, the SSHA was considered to be 0. All the observed values $h_r$ inside each 25-km segment were subtracted from the SSHA to obtain the radar freeboard height, as shown in Eq. (2) and Fig. 4(b) and (c). Since the HY-2B SGDR product has been retracked, we did not retrack HY-2B again in this study. Eq. (2) can be expressed as follows:

$$f_r = h_r - SSHA, \tag{2}$$

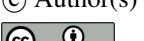



where $f_r$ is the radar freeboard, $h_r$ is the relative surface elevation after eliminating the residual, and *SSHA* is the sea surface height anomaly.

(5) Because the speed of electromagnetic waves is attenuated when the waves pass through a snow layer, a wave propagation speed correction for the radar freeboard based on the snow depth data is necessary; in this correction, we assumed that electromagnetic waves can completely penetrate the snow depth, as shown in Eq. (3). Therefore, we used the AWI snow

depth data to obtain wave propagation information for comparison with the AWI CS-2 sea ice freeboard data:

$$f = f_r + 0.22 \times h_s, \qquad (3)$$

where $f$ is the sea ice freeboard, $f_r$ is the radar freeboard, and $h_s$ is the snow depth.

(6) The sea ice freeboard data are converted to sea ice thickness data by assuming hydrostatic equilibrium, as shown in Eq. (4). We gridded the sea ice thickness to the 25-km polar stereographic grid with a temporal resolution of 1 month:

$$T = \frac{\rho_w}{\rho_w - \rho_i} \bullet f + \frac{\rho_s}{\rho_w - \rho_i} \bullet h_s, \qquad (4)$$

where $T$ is the sea ice thickness, $\rho_s$ is the snow density, $\rho_{seawater}$ is the seawater density, and $\rho_i$ is the sea ice density. We used a fixed FYI density estimate of 916.7 kgm$^{-3}$ and an MYI density estimate of 882 kgm$^{-3}$ (Alexandrov et al., 2010).

### 3.2 Comparison of along-track freeboard estimates

The orbit settings for HY-2B and CS-2 are different in that it is impossible to compare their radar freeboard estimates from

the same position at the same time, so we compared the radar freeboard estimates of HY-2B and CS-2 on adjacent tracks within the Beaufort Sea, as shown in Fig. 5. Table 3 summarizes the mean and standard deviation values of the relative surface elevation, SSHA, and radar freeboard estimates based on HY-2B and CS-2. Fig. 5(a) and (e) show the orbit positions of HY-2B and CS-2 obtained on April 4, 2020, and March 13, 2020, covering the Beaufort Sea and the northern Canadian Archipelago, respectively, to compare the relative surface elevation, SSHA, and radar freeboard estimates. In addition, both

orbits cover the FYI (grey) and MYI (black) regions. Fig. 5(b), (c), (f) and (g) show that the relative surface elevations of HY-2B and CS-2 in these two periods are 0±0.31 m/0±0.25 m and 0.081±0.17 m/0.087±0.25 m, respectively. We found that the relative surface elevation of HY-2B was slightly lower than that of CS-2. The SSHAs of HY-2B and CS-2 in the two periods are -0.26±0.13 m/-0.20±0.18 m and -0.051±0.029 m/-0.069±0.066 m, respectively. We found that the SSHAs estimated by HY-2B were lower than those estimated by CS-2, and the SSHA dispersion estimated by HY-2B was higher

than that estimated by CS-2. Fig. 5(d) and (h) show the radar freeboard estimates of HY-2B and CS-2 in the two periods, respectively. We found that the radar freeboard estimates of HY-2B were larger than those of CS-2. There were some abnormally high values caused mainly by the relatively low SSHA values. In addition, because the tracks of HY-2B and CS-2 did not overlap completely, differences in relative surface elevation, SSHA, and radar freeboard estimates were also caused by the location and time period differences between these two satellites.





## 4 Results

In this section, we used the method proposed in Section 3 to retrieve the HY-2B radar freeboard and sea ice thickness data during the two periods of interest (from October 2019 to April 2020 and from October 2020 to April 2021). We also compared the results with the CS-2 radar freeboard and sea ice thickness data released by AWI during the same periods and analysed the differences between the HY-2B and CS-2 products with regards to different sea ice types. Finally, we verified the results using OIB airborne data and ICESat-2 laser altimeter data.

### 4.1 Comparison with AWI CS-2 radar freeboard data

Based on the HY-2B SGDR data, we analysed the HY-2B monthly average radar freeboard data collected from October 2019 to April 2020 while also comparing them with the AWI CS-2 radar freeboard recorded during the same period, as shown in Fig. 6. The spatial patterns of the HY-2B and CS-2 data were in broad agreement; that is, thicker radar freeboards occurred north of the Canadian Archipelago, while thinner radar freeboards occurred in other seas. Despite this good spatial consistency, the HY-2B radar freeboards were generally thicker than those of AWI CS-2, except in April 2020. The mean deviations between the HY-2B radar freeboard and the AWI CS-2 radar freeboard data ranged from -2.4 cm to 1.9 cm from October 2019 to April 2020. The smallest mean deviation was -2.4 cm, indicating that the HY-2B radar freeboards were smaller than those of AWI CS-2 in April 2020. However, the largest mean deviation was 1.9 cm in October 2019. The mean absolute error (MAE) between the HY-2B radar freeboard and the AWI CS-2 radar freeboard was within 10 cm. Because the spatial coverage of HY-2B can reach only 81° N/S while that of CS-2 can reach 88° N/S, the monthly average radar freeboard derived based on the HY-2B retrievals have no observation data in the Arctic central region. Because HY-2B cannot cover other Arctic seas except the central Arctic region, the HY-2B radar freeboard results are sparse, especially in early winter (October 2019 to December 2019).

Table 4 shows the mean and modal radar freeboards of HY-2B and AWI CS-2 from October 2019 to April 2020 and from October 2020 to April 2021. For comparison, only the overlapping data points in the two satellite products were considered. In late spring, the CS-2 radar freeboard was thicker than the HY-2B radar freeboard. The mode HY-2B freeboard was thicker than that of AWI CS-2 except in March 2020, April 2020, October 2020, and April 2021. The AWI CS-2 mean freeboard was larger than the modal freeboard in all months (Schwegmann et al., 2016). The mean HY-2B freeboard was thicker than the modal HY-2B freeboard except in October 2019, January 2020, November 2020, and January 2021.

Fig. 7 shows the seasonal variation trends of the HY-2B and AWI CS-2 radar freeboards from October 2019 to April 2020 and from October 2020 to April 2021. We calculated the average HY-2B radar freeboard and AWI CS-2 radar freeboard values over the common area. Because the growth trend of the HY-2B radar freeboard was slower than that of the AWI CS-2 radar freeboard, the HY-2B radar freeboard values were higher than the AWI CS-2 radar freeboard values in winter, while the opposite pattern was observed in late spring.



To assess the deviation between the HY-2B and AWI CS-2 radar freeboards on various sea ice types, we list the differences in FYI, MYI and total sea ice between the two monthly average radar freeboards in Table 5. The deviation between the HY-2B radar freeboard and the AWI CS-2 radar freeboard on FYI was larger than that on MYI, with deviations of approximately 2 cm on FYI and 1 cm on MYI. In addition, the deviation between the HY-2B radar freeboard and the AWI CS-2 radar freeboard changed from positive to negative with time. In April, the deviations between HY-2B and AWI CS-2 were negative on FYI, MYI and total sea ice, indicating that the HY-2B radar freeboards were smaller than the AWI CS-2 radar freeboards. In general, the HY-2B radar freeboards exhibited a deviation of approximately 2 cm with respect to the CS-2 radar freeboards. This deviation may have been limited by the accuracy of the extracted HY-2B SSHAs. Because the HY-2B SSHAs were lower than those of CS-2, the HY-2B radar freeboards were higher than those of CS-2.

## 4.2 Comparison of sea ice thickness with AWI CS-2 data

Fig. 8 shows the spatial comparison between the HY-2B Arctic sea ice thickness and the AWI CS-2 sea ice thickness from October 2019 to April 2020. The spatial patterns of the two sea ice thickness products exhibited broad agreement; thicker sea ice occurred north of the Canadian Archipelago, while thinner sea ice occurred in the Eurasian continental marginal sea and Baffin Bay. At the same time, both products showed similar seasonal changes in which the Arctic sea ice thickness gradually thickened. Although the spatial distribution was consistent, the HY-2B sea ice thickness was larger than that of AWI CS-2 except in April 2020. This was mainly due to the thicker HY-2B radar freeboards than those of AWI CS-2. The mean deviations between the HY-2B sea ice thickness and the AWI CS-2 sea ice thickness ranged from -0.207 m to 0.155 m from October 2019 to April 2020. The mean deviations between the HY-2B and AWI CS-2 sea ice thicknesses were -0.207 m in April 2020 and 0.155 m in October 2019. The MAE between the HY-2B sea ice thickness and the AWI CS-2 sea ice thickness was within 0.9 m.

Table 6 lists the calculated monthly mean and modal sea ice thickness values derived from HY-2B and AWI CS-2 from October 2019 to April 2020 and from October 2020 to April 2021. For comparison, only the overlapping data points in the two satellite products are considered. We found that the HY-2B sea ice thicknesses were thicker than the CS-2 sea ice thicknesses in early winter, while the CS-2 sea ice thicknesses were greater than the HY-2B sea ice thicknesses in late spring. This result was related to the slower growth rate of the HY-2B sea ice thickness than of the CS-2 sea ice thickness. Except in March and April, the monthly mean HY-2B sea ice thickness was always thicker than the monthly mean AWI CS-2 sea ice thickness. Except in October 2019, the HY-2B mode thickness was always thinner than the AWI CS-2 mode thickness. Finally, except in October 2020, the mean CS-2 sea ice thickness was thicker than the modal thickness. The mean HY-2B sea ice thickness was also thicker than the modal thickness except in April 2021.

Fig. 9 shows the time series of the HY-2B and AWI CS-2 monthly mean sea ice thicknesses during two sea ice growing cycles (from October 2019 to April 2020 and from October 2020 to April 2021) averaged over the overlapping regions. We calculated the average HY-2B sea ice thickness and AWI CS-2 sea ice thickness values over the common area. Similarly, we found that the seasonal sea ice thickness trend was similar to that of the radar freeboard results. Because the growth trend of





the HY-2B sea ice thickness was slower than that of the AWI CS-2 sea ice thickness, the HY-2B sea ice thickness was
thicker than the AWI CS-2 radar freeboard except in late spring. The growth rate of the AWI CS-2 sea ice thickness was
approximately twice that of the HY-2B sea ice thickness.

To assess the deviation between the HY-2B sea ice thickness and the AWI CS-2 sea ice thickness among various sea ice
types, we derived the deviations on FYI, MYI, and total sea ice, as listed in Table 7. On FYI, the HY-2B sea ice thicknesses
were thicker than the AWI CS-2 sea ice thicknesses except in April. On MYI, the HY-2B sea ice thicknesses were thinner
than the AWI CS-2 sea ice thicknesses except in October 2020 and November 2020. In addition, the deviation between the
HY-2B sea ice thickness and the AWI CS-2 sea ice thickness changed from positive to negative over time. In general, the
HY-2B sea ice thickness had a deviation of approximately 0.2 m with respect to the CS-2 sea ice thickness. This deviation
was mainly affected by the accuracy of the retrieved radar freeboard values.

### 4.3 Comparison with independent data

We used the HY-2B SGDR data collected in April 2019 to retrieve Arctic sea ice freeboard values and compared the OIB
airborne observation data with these HY-2B sea ice freeboard values and the AWI CS-2 sea ice freeboard values, as shown
in Fig. 10. Because the HY-2B radar altimeter can cover only the 81°N/S region, only 13 grids could be evaluated when
overlapped with the OIB airborne data collected in the same period. The correlation between the HY-2B sea ice freeboard
data and OIB data was 0.58, the root mean square error (RMSE) was 0.17 m, and the MAE was 0.14 m. The correlation
between the AWI CS-2 sea ice freeboard data and the OIB sea ice freeboard data was 0.84, with an RMSE of 0.10 m and an
MAE of 0.081 m. Based on hydrostatic equilibrium, the AWI snow depth data were used to convert the HY-2B sea ice
freeboard into sea ice thicknesses, which were then verified against the OIB sea ice thickness, as shown in Fig. 11. The
correlation between the HY-2B sea ice thickness and the OIB data was 0.41, with an RMSE of 2.05 m and an MAE of 1.91
m. However, the correlation between the AWI CS-2 sea ice thicknesses and the OIB sea ice thicknesses was 0.80, with an
RMSE of 1.00 m and an MAE of 0.75 m.

IS-2 laser altimeters have a range that reaches the snow surface on sea ice and therefore are not impacted by the uncertain
scattering horizons within snow layers (Magruder et al., 2020). The spatial resolution (approximately 11 m of the
measurement footprint (Fons et al., 2021)) of these altimeters are much higher than those of CS-2 (approximately 0.3 km
along-track and 1.5 km across-track) and HY-2B (approximately 1.9 km across-track), thus providing independent all-Arctic
snow freeboard and sea ice thickness data that can be compared with the HY-2B and CS-2 retrievals. The IS-2 snow
freeboard values were subtracted from the Kwok snow depths to obtain the IS-2 sea ice freeboard values. To compare these
values with the IS-2 sea ice freeboard data, we used the Kwok snow depth to perform a wave propagation speed correction
for the HY-2B and AWI CS-2 radar freeboard data. Fig. 12 shows monthly comparisons between the HY-2B sea ice
freeboard and IS-2 sea ice freeboard data and between the CS-2 sea ice freeboard and IS-2 sea ice freeboard data from
October 2019 to April 2020 and from October 2020 to April 2021, respectively. The RMSEs obtained between the HY-2B
sea ice freeboard data and the IS-2 sea ice freeboard data ranged from 0.10 m to 0.13 m, and the MAEs ranged from 0.07 m





to 0.08 m. The RMSEs between the CS-2 sea ice freeboard and IS-2 sea ice freeboard data ranged from 0.05 m to 0.07 m, and the MAEs ranged from 0.04 m to 0.05 m. The RMSEs and MAEs of the HY-2B sea ice freeboard and IS-2 sea ice freeboard data were twice those obtained for the CS-2 data. In general, the HY-2B sea ice freeboard data were credible.

Assuming hydrostatic equilibrium, the HY-2B and CS-2 sea ice freeboard heights were converted to sea ice thicknesses using the Kwok snow depth, and the results were compared with the IS-2 sea ice thicknesses. Fig. 13 shows comparisons of the HY-2B and CS-2 sea ice thicknesses with the IS-2 sea ice thicknesses from October 2019 to April 2020 and from October 2020 to April 2021, respectively. The RMSEs derived between the HY-2B sea ice thickness and IS-2 sea ice thickness data ranged from 0.87 m to 1.11 m, and the MAEs ranged from 0.60 m to 0.70 m. The RMSEs derived between the

CS-2 and IS-2 data ranged from 0.34 m to 0.54 m, and the MAEs ranged from 0.25 m to 0.4 m. The RMSEs and MAEs between the HY-2B sea ice thickness and IS-2 sea ice thickness data were more than twice those obtained for the CS-2 data. The sea ice thickness error was thus related not only to the sea ice freeboard and snow depth values but also to the sea ice type and snow density (Ricker et al., 2014).

## 5 Discussion

In this section, we first compare the effects of the SSHAs extracted under different parameter schemes on the HY-2B radar freeboard retrievals. We then discuss the random uncertainties of the HY-2B sea ice freeboard and sea ice thickness results.

### 5.1 Influence of different SSHA determination schemes on the HY-2B radar freeboard data

Ricker et al. (2014) believed that the random uncertainty of radar freeboard data can be determined by the speckle noise and actual accuracy of sea level height data. Therefore, in this work, it was crucial to accurately extract SSHAs in the HY-2B

radar freeboard retrievals. We adopted 10 schemes to determine these SSHAs and applied them to retrieve the HY-2B radar freeboard values. The specific parameter schemes are listed in Table 8. Moreover, the HY-2B radar freeboard retrievals were compared to the AWI CS-2 radar freeboard values collected during the same period. The mean deviation, MAE and SSHA values retrieved between the two satellites under different schemes from October 2019 to April 2020 and from October 2020 to April 2021 are listed in Table 9. As the table shows (Schemes 1-5 or 6-10), the mean deviation and MAE values first

decreased and then increased with the gradual increase in SSHA, indicating that a larger SSHA does not necessitate a smaller mean deviation or MAE. In Schemes 1-5, during the two sea ice growing cycles (from October 2019 to April 2020 and from October 2020 to April 2021), although the SSHA values of Scheme 5 were largest, all of which were greater than -0.03 m, the mean deviations were all less than 0, indicating that the HY-2B radar freeboard retrievals were generally lower than the AWI CS-2 radar freeboards; in addition, under this scheme, the MAE is larger than that obtained under Scheme 4. Finally,

according to the mean deviation and MAE values, we used Scheme 4 to extract SSHAs to retrieve the HY-2B radar freeboards. It is worth noting that the SSHAs extracted by Scheme 4 resulted in relatively slow HY-2B sea ice freeboard and




sea ice thickness growth rates compared to CS-2. Therefore, the HY-2B sea ice freeboard and sea ice thickness values were lower than those of CS-2 in late spring, especially in March and April, as shown in Tables 4 and 6.

### 5.2 Uncertainty of HY-2B radar freeboard and sea ice thickness data

The speckle noise caused by instrument system errors was found to be $\sigma_{SGDR} = 0.02\ m$ (National Satellite Ocean Application Service, 2019), and the SSHA uncertainty was assumed to be determined by the standard deviation of observation points within a moving 25-km window. If only one observation point or no observation point was observed within the 25-km sliding window, the grid point was ignored. The sea ice freeboard value was obtained after a wave propagation speed correction was applied to the radar freeboard, so the sea ice freeboard uncertainty can be expressed as
shown in Eq. (5):

$$\sigma_f = \sqrt{\sigma_{SGDR}^2 + \sigma_{SSA}^2 + (0.22 \times \sigma_{h_s})^2}\ , \tag{5}$$

where $\sigma_{SGDR} = 0.02\ m$, $\sigma_{SSA}$ is the standard deviation of observation points within a 25-km moving window, $\sigma_f$ is the sea ice freeboard uncertainty, and $\sigma_{h_s}$ is the AWI snow depth uncertainty.

We calculated the partial derivative of Eq. (4) to obtain the weights of the single-variable variances to obtain the contribution
of each variable to the thickness uncertainty, as shown in Eq. (6)-(9).

$$\frac{\partial T}{\partial f_r} = \frac{\rho_w}{\rho_w - \rho_i}\ , \tag{6}$$

$$\frac{\partial T}{\partial \rho_i} = \frac{(f_r + 0.22 h_s) \times \rho_w + h_s \times \rho_s}{(\rho_w - \rho_i)^2}\ , \tag{7}$$

$$\frac{\partial T}{\partial h_s} = \frac{\rho_s + 0.22 \rho_w}{\rho_w - \rho_i}\ , \tag{8}$$

$$\frac{\partial T}{\partial \rho_s} = \frac{h_s}{\rho_w - \rho_i}\ , \tag{9}$$

The sea ice thickness uncertainty can be divided into random uncertainty and systematic uncertainty. Ricker et al. (2014) hypothesized that the uncertainties of the modified W99 snow depth and snow density values resulting from interannual variabilities are systematic and cannot be regarded as random uncertainty. However, the AWI snow depth product is a composite snow depth product obtained by integrating the W99 climatology snow depths and the daily average AMSR-2 snow depths of Bremen University. Therefore, we also assumed that the uncertainties in the AWI snow depth and snow





density products were systematic. Due to the variability in seawater density, the contribution of its uncertainty was ignored (Kurtz et al., 2014; Ricker et al., 2014). We calculated the mixed uncertainty of the sea ice thickness via Gaussian error propagation, as shown in Eq. (10):

$$
\begin{aligned}
\sigma_T &= \sqrt{(\frac{\partial T}{\partial f} \times \sigma_f)^2 + (\frac{\partial T}{\partial \rho_i} \times \sigma_{\rho_i})^2 + (\frac{\partial T}{\partial h_s} \times \sigma_{h_s})^2 + (\frac{\partial T}{\partial \rho_s} \times \sigma_{\rho_s})^2} \\
&= \sqrt{(\frac{\rho_w}{\rho_w - \rho_i} \times \sigma_f)^2 + (\frac{(f_r + 0.22h_s) \times \rho_w + h_s \times \rho_s}{(\rho_w - \rho_i)^2} \times \sigma_{\rho_i})^2 + (\frac{\rho_s + 0.22\rho_w}{\rho_w - \rho_i} \times \sigma_{h_s})^2 + (\frac{h_s}{\rho_w - \rho_i} \times \sigma_{\rho_s})^2}
\end{aligned}
\tag{10}
$$

where $\sigma_{\rho_i}$ is the uncertainty of the sea ice density, $\sigma_{\rho FYI} = 35.7\ kg/m^3$, $\sigma_{\rho MYI} = 23\ kg/m^3$, and $\sigma_{\rho_s} = 50\ kg/m^3$.

We gridded the HY-2B sea ice freeboard uncertainty and sea ice thickness uncertainty to a 25-km polar stereographic grid, as shown in Eq. (11):

$$
\sigma_{\bar{f}, \bar{T}} = \sqrt{\frac{1}{\sum_{i=1}^{N} \frac{1}{\sigma_{[f_i, T_i]}^2}}},
\tag{11}
$$

where $\sigma_{\bar{f}, \bar{T}}$ is the gridded sea ice freeboard uncertainty or sea ice thickness uncertainty and $N$ is the number of HY-2B observation points inside each 25-km grid cell.

We compared the HY-2B sea ice freeboard uncertainty and AWI CS-2 sea ice freeboard uncertainty from October 2019 to April 2020 and from October 2020 to April 2021, and the results are shown in Fig. 14. Because more data points were obtained in high-latitude areas than in low-latitude areas, the uncertainty of the HY-2B sea ice freeboard results was larger in low-latitude areas than in high-latitude areas. The uncertainty of the HY-2B sea ice freeboard results was larger than that of the CS-2 results. This was mainly because the SSHA uncertainty ($\sigma_{SSA}$) of HY-2B was larger than that of CS-2. Table 10

summarizes the average HY-2B sea ice freeboard uncertainty and CS-2 sea ice freeboard uncertainty values derived over the common area. The HY-2B sea ice freeboard uncertainty values ranged from 8.5 to 12.0 cm, while the CS-2 sea ice freeboard uncertainty values ranged from 1.2 to 1.5 cm.

Fig. 15 shows the comparison of the HY-2B and AWI CS-2 sea ice thickness uncertainties from October 2019 to April 2020 and from October 2020 to April 2021. We found that the spatial distribution of the HY-2B sea ice thickness uncertainty was

similar to that of the sea ice freeboard uncertainty; that is, the uncertainty was larger at low latitudes than at high latitudes. In addition, the uncertainty in the sea ice thickness in HY-2B was larger than that in CS-2. This was mainly due to the higher uncertainty of the HY-2B sea ice freeboard values than of the CS-2 values. Table 11 summarizes the HY-2B sea ice freeboard uncertainty and CS-2 sea ice freeboard uncertainty values over the common area. The HY-2B sea ice thickness uncertainties ranged from 26.8 cm to 37.7 cm, while the CS-2 sea ice thickness uncertainties ranged from 12.1 cm to 15.4 cm.





## 6 Conclusion

In this study, we first used Chinese HY-2B radar altimeter data to estimate Arctic sea ice freeboard and sea ice thickness values with a new retrieval method and then compared the results to the AWI CS-2 sea ice freeboard and sea ice thickness values recorded during the same period. The accuracy of the findings was verified with independent data sources including NASA OIB airborne data and IS-2 laser altimeter data. Finally, the random uncertainties in the HY-2B sea ice freeboard and sea ice thickness results were estimated. The main conclusions are as follows:

(1) The spatial distributions of the HY-2B radar freeboard and AWI CS-2 radar freeboard results had good consistency, but there were still some differences in the numerical values and temporal evolution. The mean deviations between the HY-2B radar freeboard and AWI CS-2 data ranged from -2.4 cm to 1.9 cm from October 2019 to April 2020. Except in April 2020, the HY-2B radar freeboard values were thicker than the AWI CS-2 radar freeboard values. The errors (mean deviation and MAE values) between the HY-2B radar freeboard and AWI CS-2 radar freeboard data on FYI were larger than those on MYI. In addition, the growth trend of the HY-2B radar freeboard values was slower than that of the AWI CS-2 radar freeboard values. Similarly, the spatial distributions of the HY-2B sea ice thickness and AWI CS-2 data exhibited good consistency, but we still identified some differences in their numerical and temporal evolution patterns. The mean deviations between the HY-2B sea ice thickness and AWI CS-2 sea ice thickness data ranged from -0.207 m to 0.155 m from October 2019 to April 2020. Except in April 2020, the HY-2B sea ice thicknesses were greater than the AWI CS-2 sea ice thicknesses. The increasing trend of the HY-2B sea ice thickness was slower than that of the AWI CS-2 sea ice thickness.

(2) Comparisons with the OIB sea ice freeboard and sea ice thickness values obtained in April 2019 showed that the correlation between the HY-2B sea ice freeboard retrievals and OIB values was 0.58, the RMSE was 0.17 m, and the MAE was 0.14 m. The correlation between the HY-2B sea ice thickness retrievals and OIB sea ice thicknesses was 0.41, the RMSE was 2.05 m, and the MAE was 1.91 m. Moreover, the comparisons with independent data recorded by the IS-2 laser altimeter from October 2019 to April 2020 and from October 2020 to April 2021 showed that the RMSEs ranged from 0.10 to 0.13 m, while the MAEs ranged from 0.07 m to 0.08 m. The RMSE and MAE values derived between the HY-2B sea ice freeboard and IS-2 sea ice freeboard results were twice those derived between the CS-2 and IS-2 data. The RMSE between the HY-2B sea ice thickness and IS-2 sea ice thickness ranged from 0.87 m to 1.11 m, and the MAE ranged from 0.60 m to 0.70 m. The RMSE and MAE values derived between the HY-2B sea ice thickness and IS-2 sea ice thickness are more than twice those obtained between the CS-2 and IS-2 data. The sea ice thickness error was related not only to the sea ice freeboard and snow depth values but also to the sea ice type and snow density.

(3) Based on the Gaussian error propagation theory, we estimated the uncertainties in the HY-2B sea ice freeboard and sea ice thickness values from October 2019 to April 2020 and from October 2020 to April 2021 and found that the HY-2B sea ice freeboard uncertainty values ranged from 8.5 cm to 12.0 cm, while the uncertainties in the HY-2B sea ice thickness ranged from 26.8 cm to 37.7 cm. Due to the influence of $\sigma_{SSA}$ and the number of observation points within the grid, the uncertainties in the HY-2B sea ice freeboard and sea ice thickness were higher at low latitudes than at high latitudes.





In this study, we preliminarily tried to use HY-2B radar altimeter data to retrieve Arctic sea ice thicknesses and obtained reliable results. However, the shortcoming of this work is that we did not accurately distinguish between floes and lead. In the future, we will develop a higher-accuracy classification algorithm to classify floes and lead and use this improved algorithm to retrieve sea ice freeboard and sea ice thickness data. Moreover, the HY-2B SGDR data used in this work retained only the measurements of the suboptimal maximum likelihood estimation (SMLE) retracking algorithm, which is applicable only to the ocean surface. Although the offset centre of gravity (OCOG) retracking algorithm is applicable to nonocean surfaces, including land and sea ice, it is not saved in SGDR data and thus needs to be obtained from HY-2B L1b data. Moreover, it is necessary to recalculate the satellite altitude of the OCOG retracking algorithm using fine-orbit determination data and recalculate various environmental correction terms, including the wet and dry troposphere correction, ionospheric correction, ocean tidal correction, polar tide correction and earth tide correction terms. We will reprocess the HY-2B L1b data in the future to obtain more suitable products for polar sea ice thickness retrievals.

*Data availability.* The HY-2B SGDR data are available at ftp://osdds-ftp.nsoas.org.cn/, provided by the NSOAS (last access: 30 June 2022). The radar freeboard and sea ice thickness data corresponding to CryoSat-2 Level 2I are available at ftp://science-pds.cryosat.esa.int/, provided by the ESA (last access: 30 June 2022). The CryoSat-2 radar freeboard, sea ice thickness, and snow depth data are available at ftp://ftp.awi.de/sea_ice/, provided by the AWI (Ricker et al., 2014; Hendricks et al., 2020) (last access: 30 June 2022). The ATL20 products (version 003) for the ICESat-2 laser altimeter are available at https://nsidc.org/data/ATL20/versions/3, provided by the NSIDC (Petty et al., 2021) (last access: 30 June 2022). The ICESat-2 Arctic monthly average sea ice thickness product and the snow depth product retrieved by ICESat-2 and CryoSat-2 are available at https://icesat-2.gsfc.nasa.gov/sea-ice-data/kacimi-kwok-2022 (Kwok et al., 2020; Kacimi et al., 2022) (last access: 30 June 2022). The IceBridge L4-level data (IDCSI4) are available at https://nsidc.org/data/NSIDC-0708/versions/1/, provided by the NSIDC (Kurtz et al., 2013) (last access: 30 June 2022). The sea ice concentration and sea ice type data are available at https://osi-saf.eumetsat.int, provided by the OSI-SAF (Rasmus Tonboe et al., 2016; Signe et al., 2021) (last access: 30 June 2022). The DTU18 MSS data are available at ftp://ftp.space.dtu.dk/pub/, provided by the DTU (Andersen et al. 2018a; Andersen et al. 2018b) (last access: 30 June 2022).

*Author contributions.* Data curation, Z.D. and L.S.; writing, Z.D. and L.S.; methodology, Z.D. L.S. M.L. T.Z. and S.W.; validation, Z.D. T.Z. and S.W.; funding acquisition, L.S. and M.L. All authors have read and agreed to the published version of the manuscript.

*Competing interests.* The authors declare that they have no conflicts of interest.





*Acknowledgements.* The research is funded by the National Key Research and Development Program of China (grant numbers 2021YFC2803300 and 2018YFC1407200) and the Impact and Response of Antarctic Seas to Climate Change (grant number IRASCC2020-2022-No. 01-01-03).

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

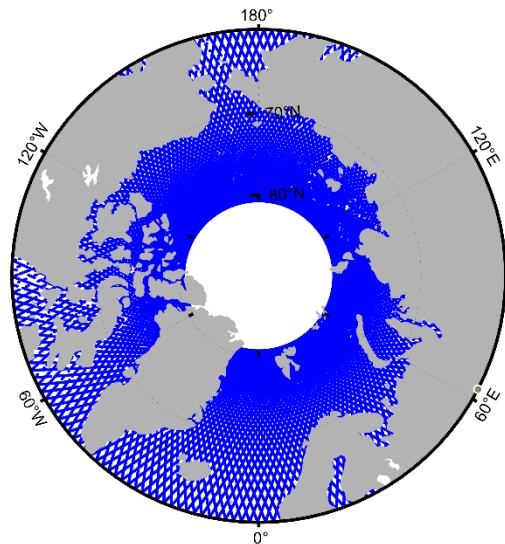

**Figure 1: An example of the accurate data representation & universal readability of figures.**

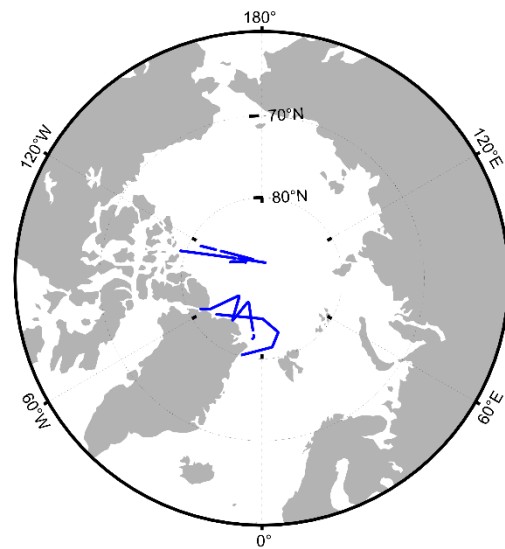

**Figure 2: Trajectory of the sea ice flight experiments conducted over the Arctic. The blue line shows the OIB track in April 2019.**



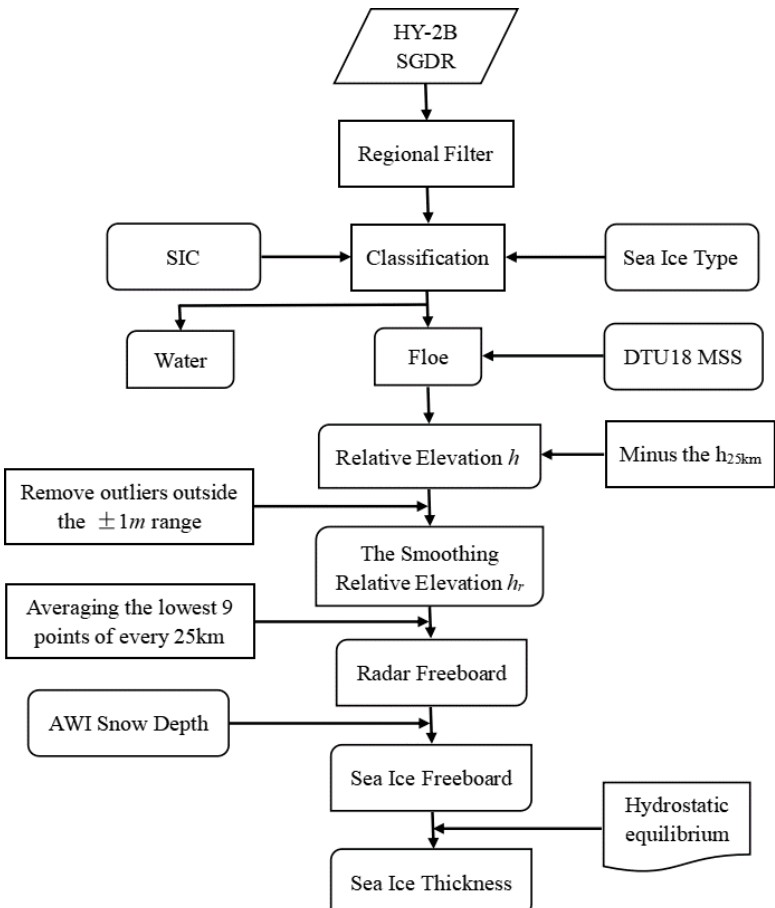

**Figure 3: A flowchart of the sea ice thickness retrieval algorithm.**






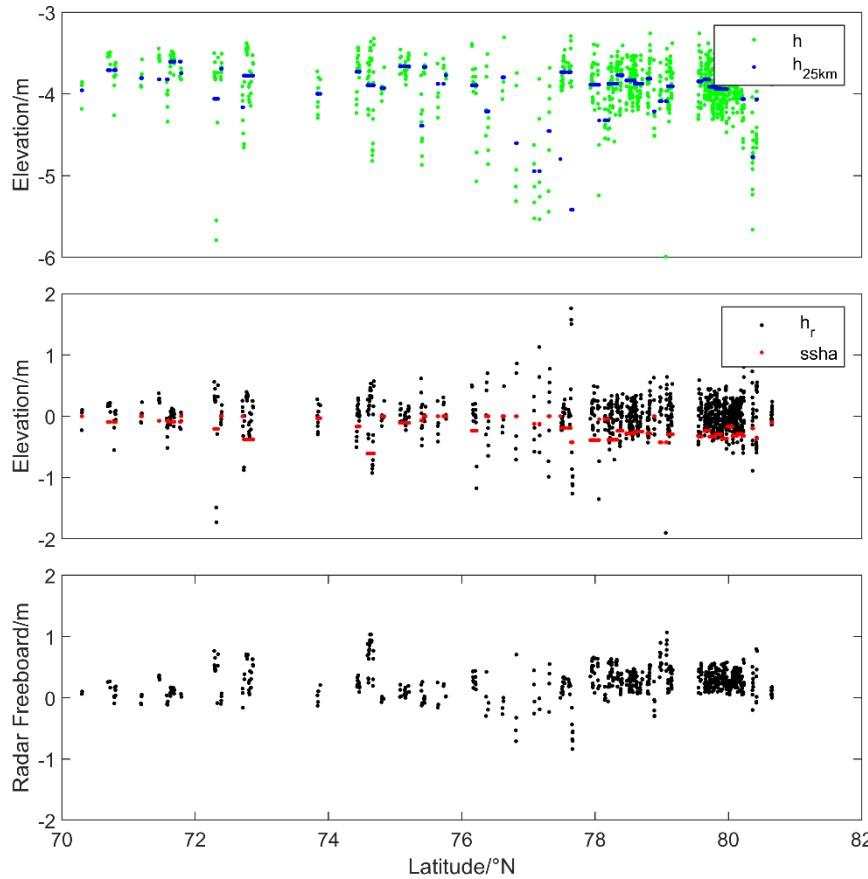

**Figure 4: A sample of the HY-2B elevation profile obtained for of track number 14418 on April 4, 2020. The green points in panel (a) are the relative elevation ($h$) values; the blue points in panel (a) are the $h_{25km}$ values, defined as the 25-km running mean of $h$; the black points in panel (b) are the modified relative elevation ($h_r$) values; the red points in panel (b) are the sea surface height anomaly (SSHA) values; and the black points in panel (c) are the radar freeboard values.**





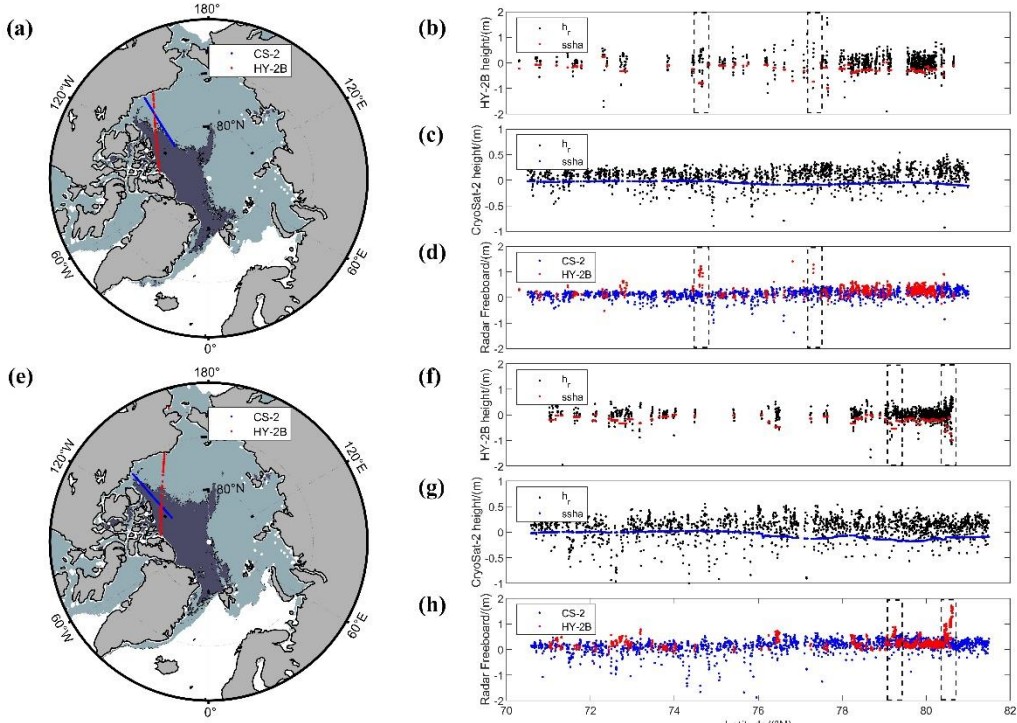

**Figure 5: (a) (e) Cryosat-2 (blue) and HY-2B (red) tracks (acquired on April 4, 2020, and March 13, 2020, respectively) selected for comparison. FYI regions: light shading, MYI regions: dark grey shading. (b) (f) HY-2B sea ice floe relative surface elevations (black dots) and SSHAs (red dots) corresponding to the tracks shown in panels (a) and (e), respectively. (c) (g) Cryosat-2 sea ice floe (black dots) relative surface elevations and SSHAs (red dots) corresponding to the tracks shown in panels (a) and (e), respectively. (d) (h) Cryosat-2 (blue) and HY-2B (red) radar freeboard values corresponding to the tracks shown in panels (a) and (e), respectively.**






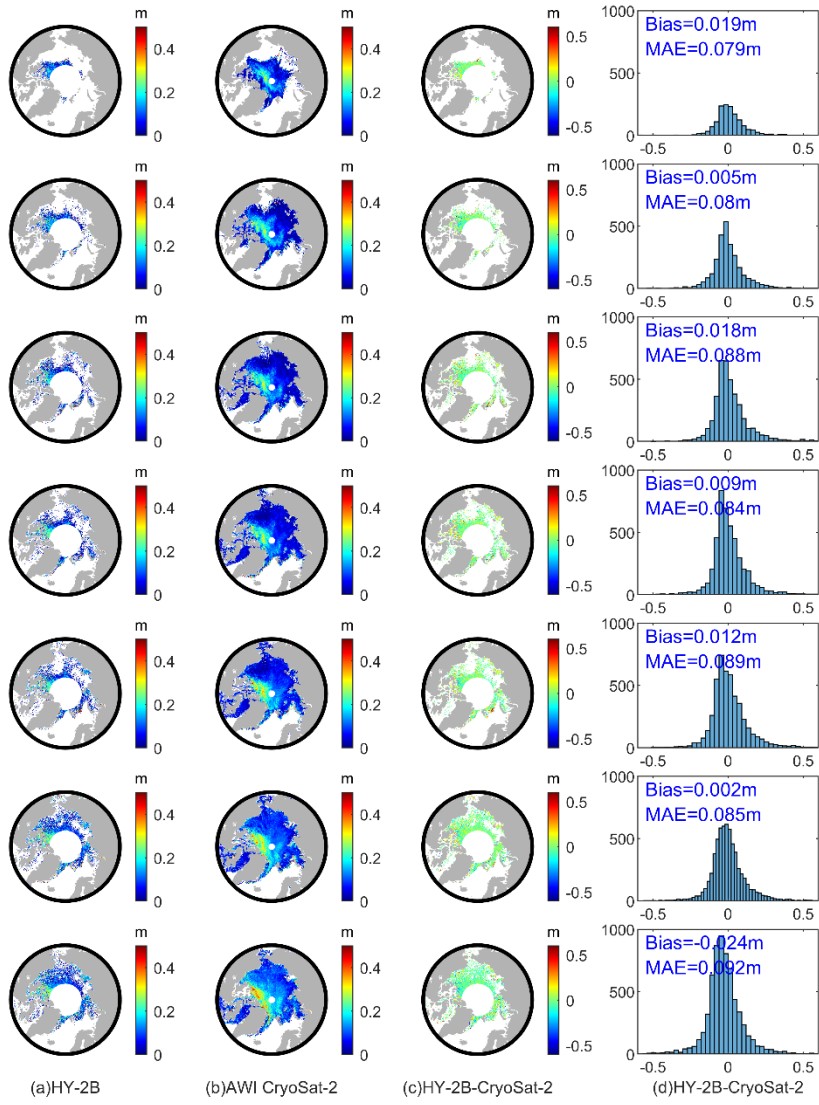

(a)HY-2B          (b)AWI CryoSat-2          (c)HY-2B-CryoSat-2          (d)HY-2B-CryoSat-2

**Figure 6. Comparisons and differences between HY-2B radar freeboard and AWI Cryosat-2 radar freeboard values recorded from October 2019 to April 2020, (a) HY-2B radar freeboard values, (b) Cryosat-2 radar freeboard values, (c) the spatial difference between the HY-2B and Cryosat-2 radar freeboard values, and the (d) histogram of differences between the HY-2B and Cryosat-2 radar freeboard values.**




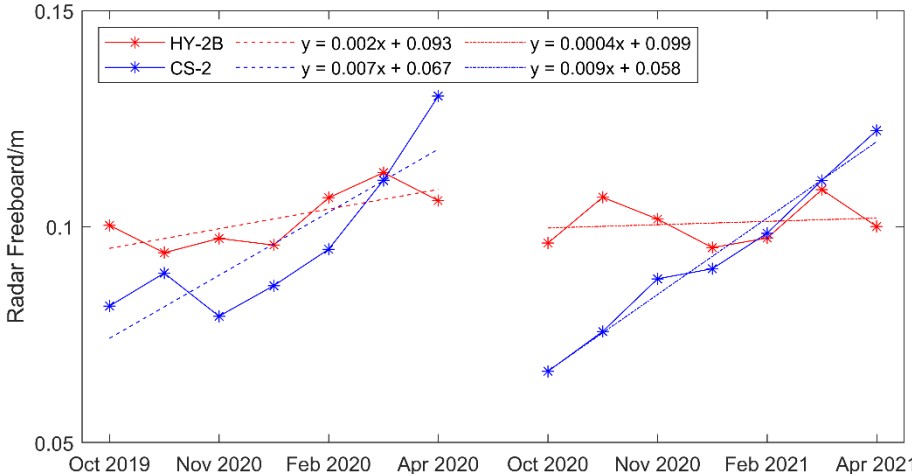

**Figure 7: Seasonal variation trends of HY-2B and CryoSat-2 radar freeboard values from October 2019 to April 2020 and from October 2020 to April 2021.**




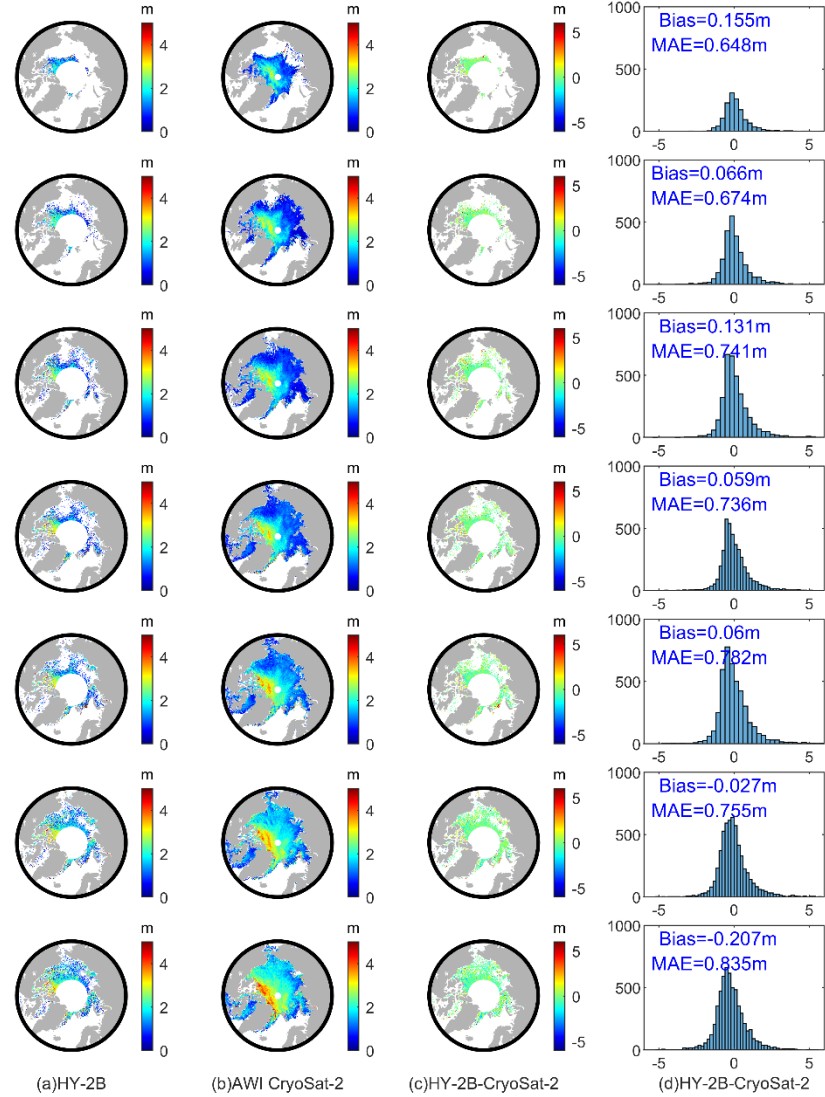

**Figure 8:** Comparisons and differences between HY-2B sea ice thickness and AWI Cryosat-2 sea ice thickness values from October 2019 to April 2020, (a) HY-2B sea ice thicknesses, (b) Cryosat-2 sea ice thicknesses, (c) spatial differences between HY-2B and Cryosat-2 sea ice thicknesses, and (d) a histogram of the differences between the HY-2B and Cryosat-2 sea ice thicknesses.



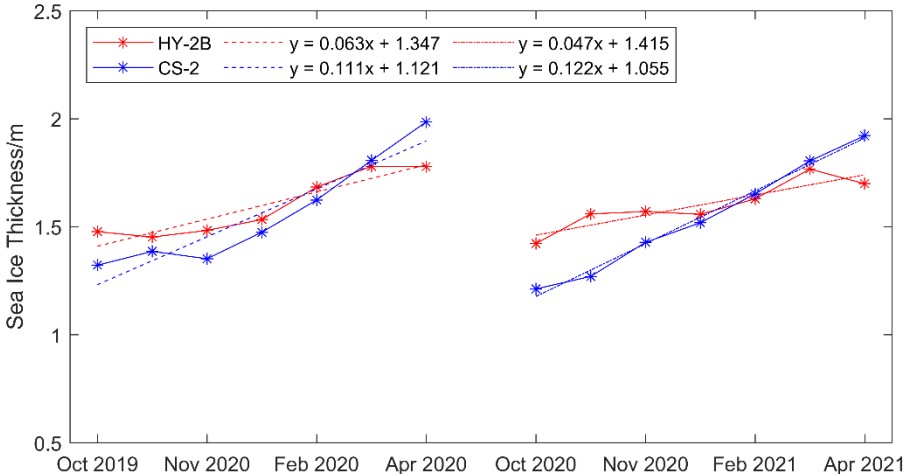

**Figure 9: Seasonal variation trends of HY-2B and CryoSat-2 sea ice thicknesses from October 2019 to April 2020 and from October 2020 to April 2021.**


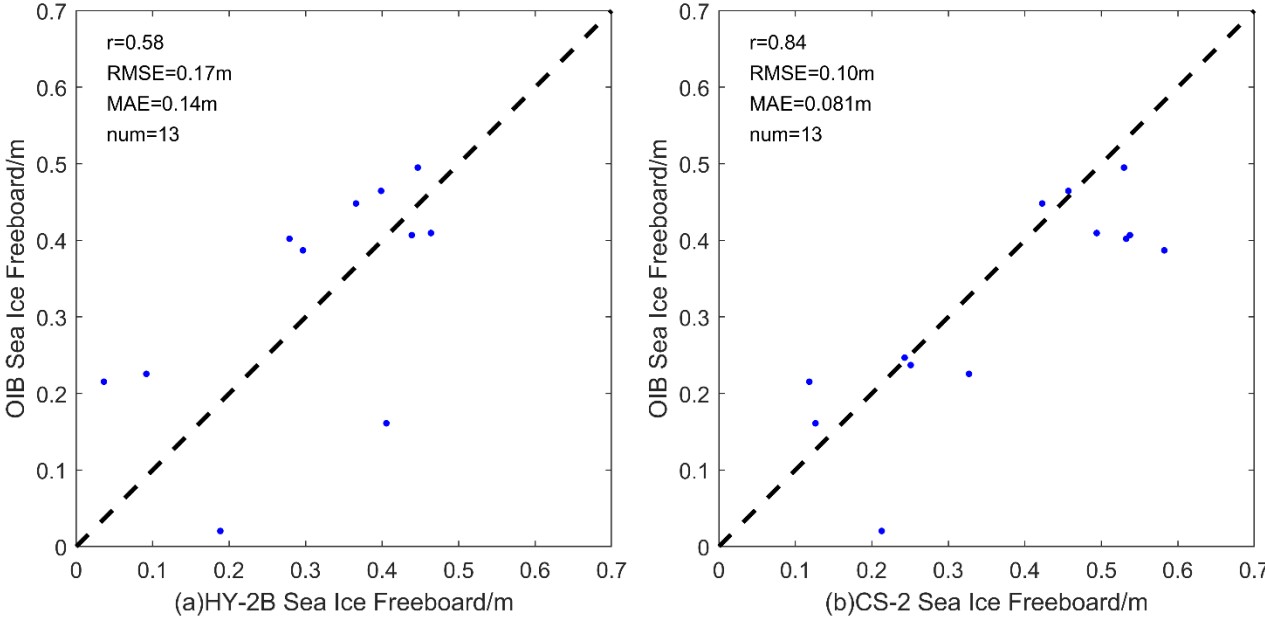

**Figure 10: Comparison between two kinds of sea ice freeboard products and the OIB sea ice freeboard values collected in April 2019: (a) the HY-2B sea ice freeboard values and (b) the AWI CryoSat-2 sea ice freeboard values.**



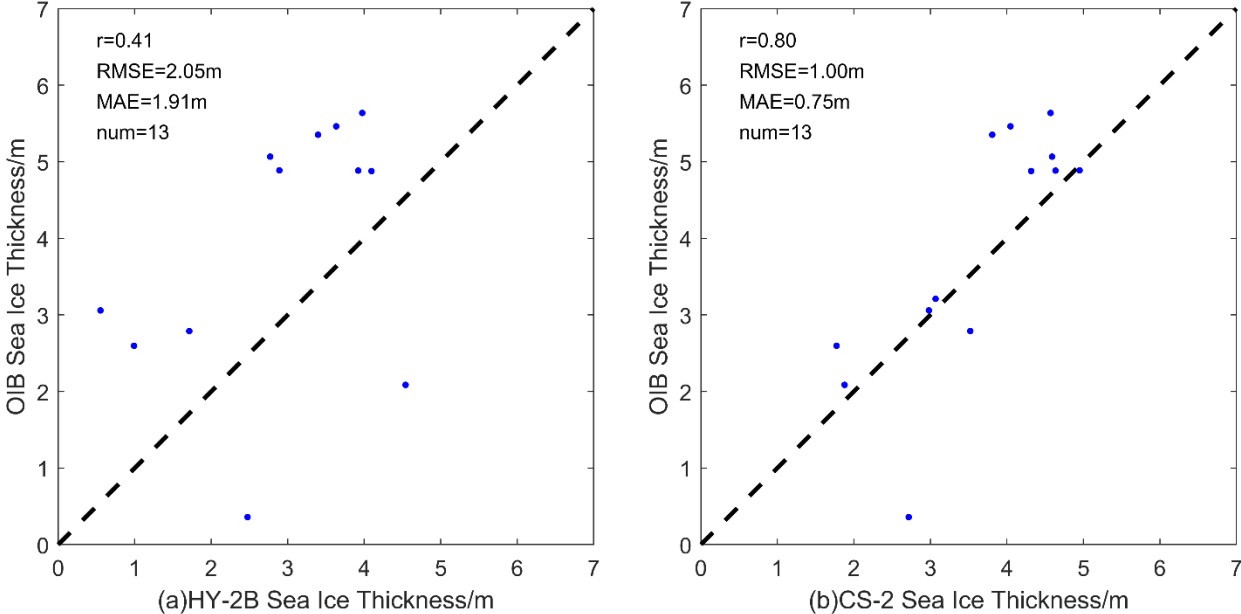


**Figure 11: Comparison between two kinds of sea ice thickness products and the OIB sea ice thickness values collected in April 2019: (a) the HY-2B sea ice thickness values and (b) the AWI CryoSat-2 sea ice thickness values.**





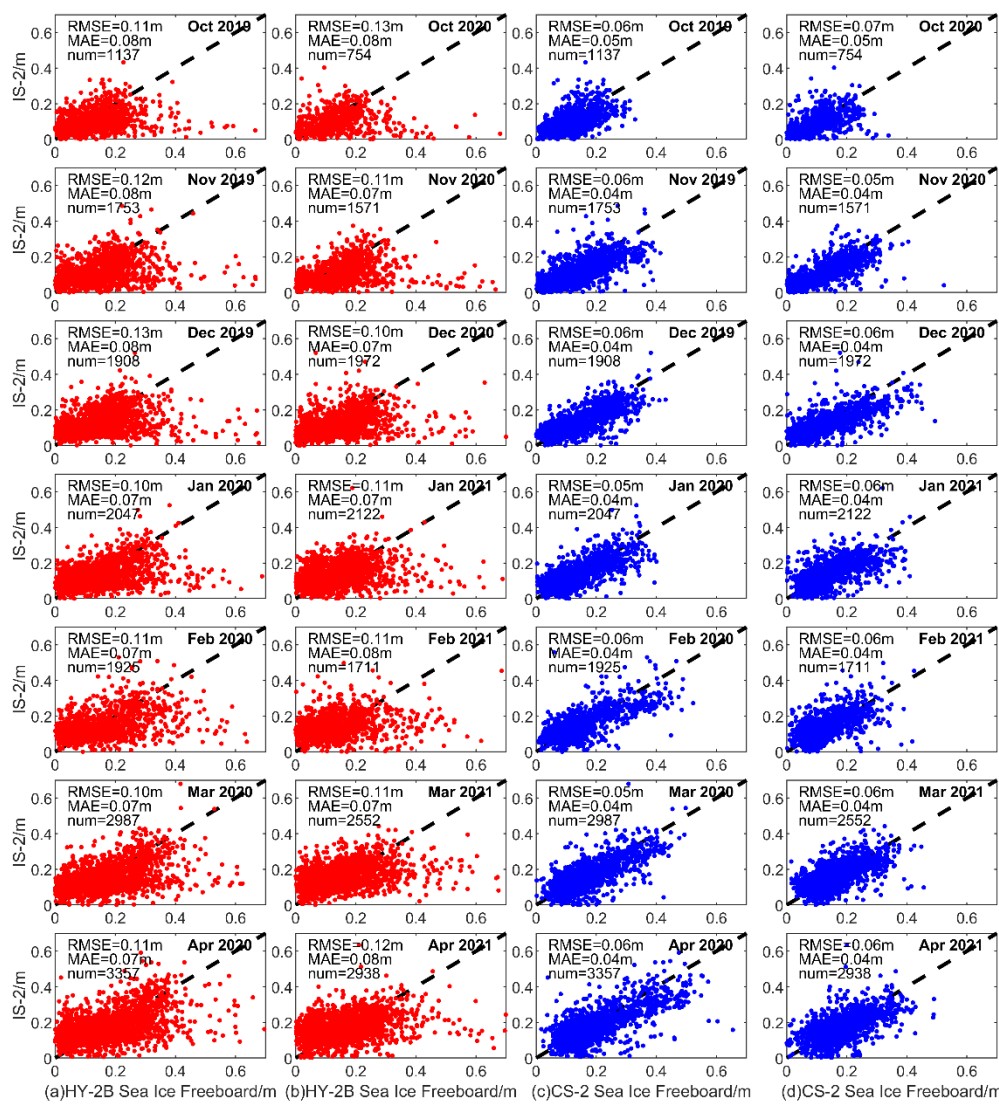

**Figure 12: Monthly comparisons between HY-2B sea ice freeboard and ICESat-2 sea ice freeboard values and between CryoSat-2 sea ice freeboard and ICESat-2 sea ice freeboard values: panels (a) and (c) show comparisons from October 2019 to April 2020, and panels (b) and (d) show comparisons from October 2020 to April 2021.**



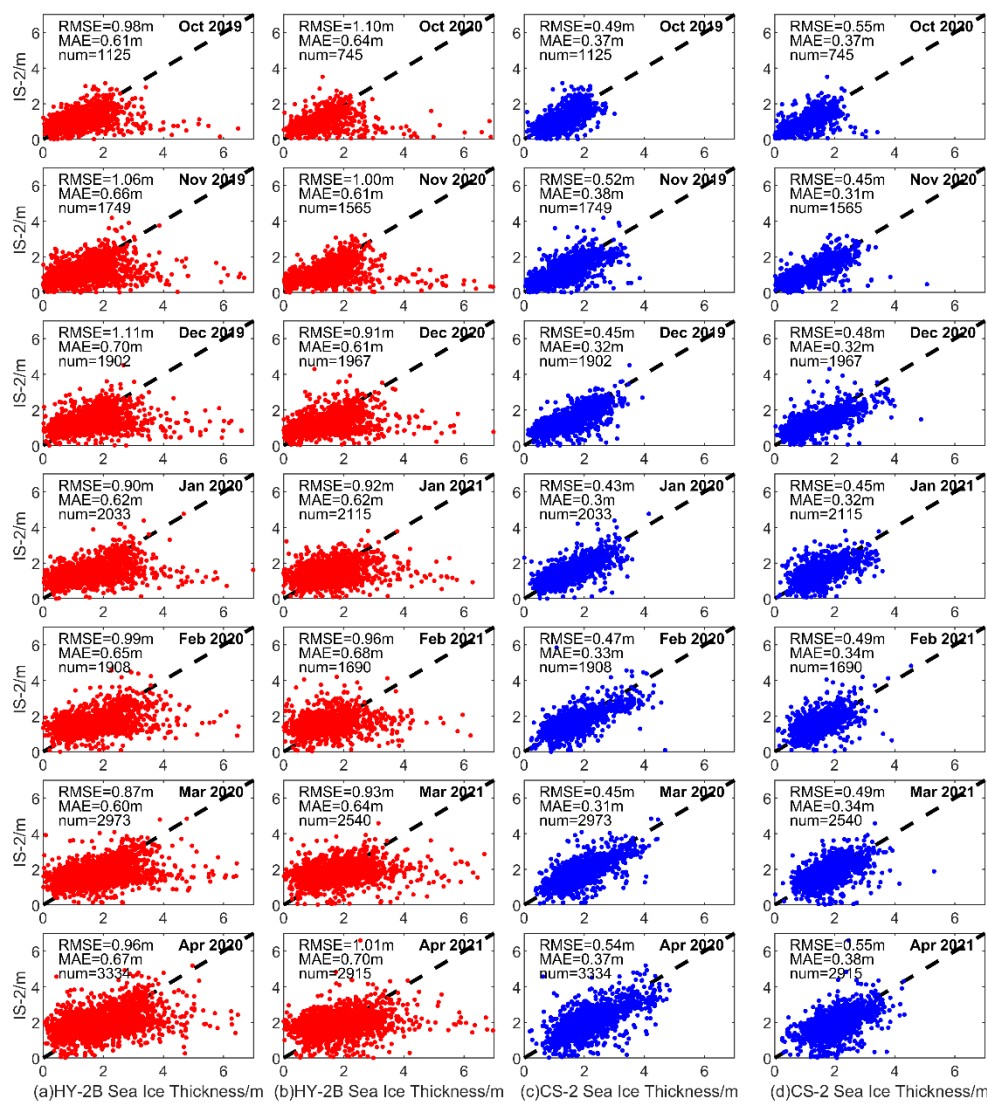

**Figure 13: Monthly comparisons between HY-2B sea ice thickness and ICESat-2 sea ice thickness values and between CryoSat-2 sea ice thickness and ICESat-2 sea ice thickness values: panels (a) and (c) show comparisons from October 2019 to April 2020, and panels (b) and (d) show comparisons from October 2020 to April 2021.**



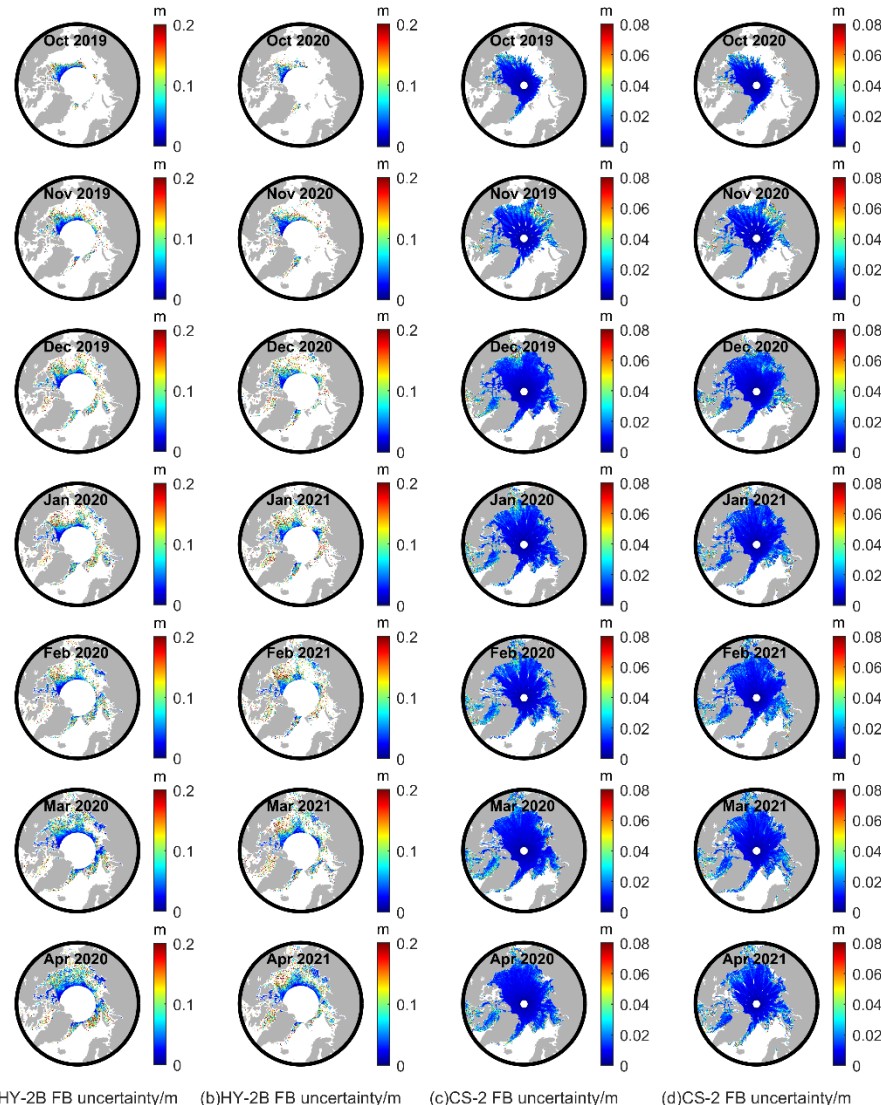

**Figure 14: Monthly comparisons between HY-2B sea ice freeboard uncertainties and CS-2 sea ice freeboard uncertainties from October 2019 to April 2020 and from October 2020 to April 2021: panels (a) and (c) show comparisons from October 2019 to April 2020, and panels (b) and (d) show comparisons from October 2020 to April 2021.**





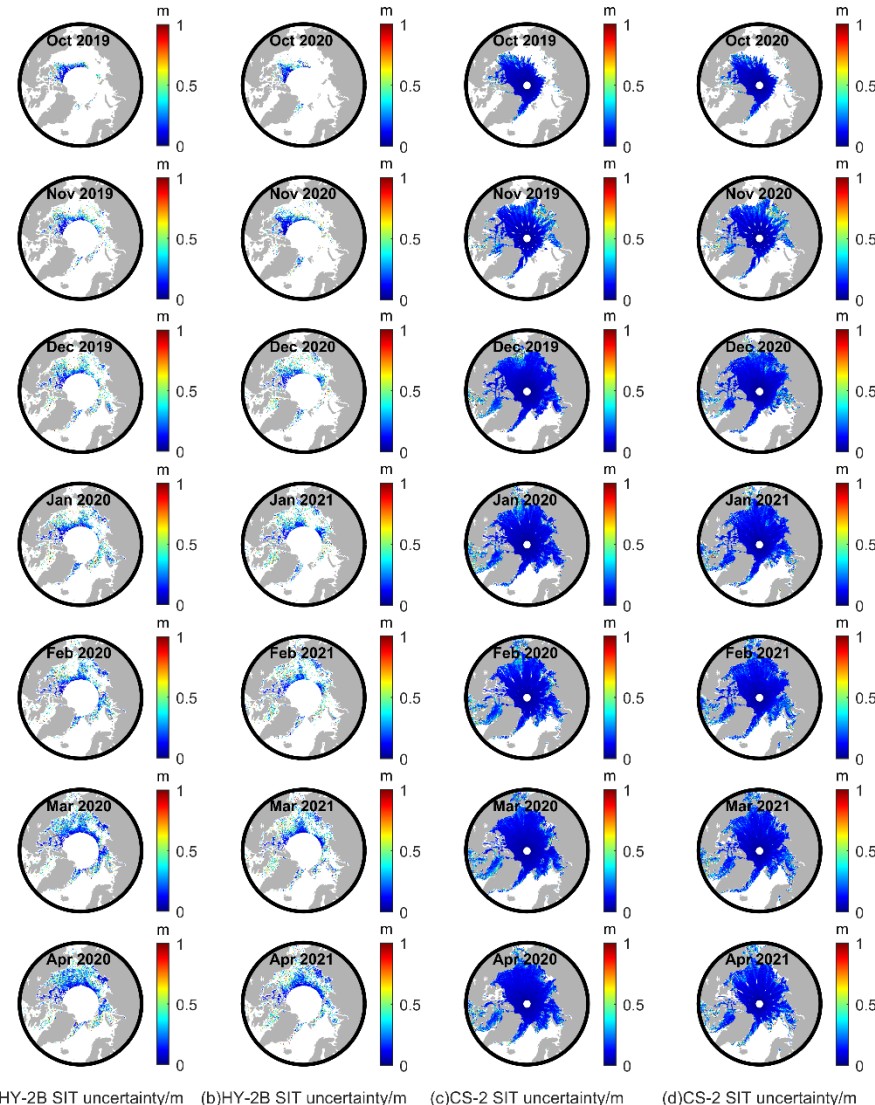

**Figure 15: Monthly comparisons between HY-2B sea ice thickness uncertainties and CS-2 sea ice thickness uncertainties from October 2019 to April 2020 and from October 2020 to April 2021: panels (a) and (c) show comparisons from October 2019 to April 2020, and panels (b) and (d) show comparisons from October 2020 to April 2021.**



**Table 1: HY-2B radar altimeter main parameters.**

| Parameter | Value | |
|---|---|---|
| Band | Ku | C |
| Centre frequency | 13.58 GHZ | 5.25 GHz |
| Chirp signal bandwidth | 320/80/20 MHz | 160/40/10 MHz |
| Footprint diameter | 1.9 km | 10 km |
| Bandwidth | 102.4 us | |
| Waveform bin number | 128 | |
| Range accuracy | < 2 cm | |
| Spatial coverage | 81°N/S | |


**Table 2: Time series of snow density values.**

| Date/month | October | November | December | January | February | March | April |
|---|---|---|---|---|---|---|---|
| Snow density/kg.m$^{-3}$ | 274.51 | 281.01 | 287.51 | 294.02 | 300.51 | 307.01 | 313.51 |

**Table 3: Comparison of the mean and standard deviation values of the relative surface elevation ($h_r$), sea surface height anomaly (SSHA), and radar freeboard estimates ($f_r$) from HY-2B and Cryosat-2.**

| Unit: m | HY-2B | | CryoSat-2 | |
|---|---|---|---|---|
| | 13 March 2020 | 4 April 2020 | 13 March 2020 | 4 April 2020 |
| $h_r$ | 0±0.25 | 0±0.31 | 0.087±0.25 | 0.081±0.17 |
| SSHA | -0.20±0.18 | -0.26±0.13 | -0.069±0.066 | -0.051±0.029 |
| $f_r$ | 0.24±0.25 | 0.27±0.23 | 0.16±0.27 | 0.13±0.18 |


**Table 4: Mean and modal radar freeboard values of HY-2B and CryoSat-2 over the common area.**

| Month | Mean/mode (Unit: m) | | | |
|---|---|---|---|---|
| | 2019.10-2020.04 | | 2020.10-2021.04 | |
| | HY-2B | CryoSat-2 | HY-2B | CryoSat-2 |
| October | 0.100/0.118 | 0.082/0.059 | 0.096/0.041 | 0.067/0.051 |
| November | 0.094/0.069 | 0.089/0.058 | 0.107/0.117 | 0.076/0.027 |
| December | 0.097/0.046 | 0.079/0.038 | 0.102/0.058 | 0.088/0.052 |
| January | 0.096/0.097 | 0.086/0.049 | 0.095/0.103 | 0.090/0.065 |
| February | 0.107/0.067 | 0.095/0.061 | 0.097/0.057 | 0.099/0.078 |
| March | 0.113/0.087 | 0.111/0.092 | 0.109/0.098 | 0.111/0.090 |
| April | 0.106/0.104 | 0.130/0.106 | 0.100/0.088 | 0.122/0.102 |



**Table 5: Differences in the monthly mean radar freeboard values of HY-2B and CryoSat-2 on FYI, MYI and total sea ice.**

| Month | 2019.10-2020.04 | | | 2020.10-2021.04 | | |
|---|---|---|---|---|---|---|
| Unit: m | FYI | MYI | ALL | FYI | MYI | ALL |
| 10 | 0.032 | 0.0071 | 0.019 | 0.068 | 0.020 | 0.030 |
| 11 | 0.012 | -0.012 | 0.0047 | 0.040 | 0.023 | 0.031 |
| 12 | 0.025 | -0.0010 | 0.018 | 0.026 | -0.0085 | 0.014 |
| 01 | 0.0093 | 0.013 | 0.0094 | 0.012 | -0.0068 | 0.0049 |
| 02 | 0.016 | 0.0052 | 0.012 | 0.0034 | -0.0074 | -0.0011 |
| 03 | 0.0046 | -0.0018 | 0.0018 | 0.0018 | -0.0089 | -0.0022 |
| 04 | -0.017 | -0.053 | -0.024 | -0.017 | -0.035 | -0.022 |
| mean | 0.012 | -0.0060 | 0.0058 | 0.019 | -0.0034 | 0.0077 |
| MAE | 0.017 | 0.013 | 0.013 | 0.024 | 0.016 | 0.015 |


**Table 6: Mean and modal freeboard values of HY-2B and CryoSat-2 sea ice thicknesses in the common area.**

| Month | Mean/mode (Unit: m) | | | |
|---|---|---|---|---|
| | 2019.10-2020.04 | | 2020.10-2021.04 | |
| | HY-2B | CryoSat-2 | HY-2B | CryoSat-2 |
| October | 1.477/1.278 | 1.322/1.023 | 1.422/0.549 | 1.212/1.313 |
| November | 1.452/0.541 | 1.386/0.794 | 1.560/0.543 | 1.270/0.551 |
| December | 1.483/0.591 | 1.352/0.891 | 1.571/0.547 | 1.427/0.968 |
| January | 1.534/0.627 | 1.475/1.081 | 1.558/0.621 | 1.518/1.150 |
| February | 1.684/0.670 | 1.624/1.290 | 1.629/0.638 | 1.652/1.189 |
| March | 1.780/1.390 | 1.807/1.794 | 1.767/0.786 | 1.805/1.542 |
| April | 1.778/0.848 | 1.986/1.824 | 1.699/1.780 | 1.921/1.862 |



**Table 7: Differences in the monthly mean sea ice thicknesses of HY-2B and CryoSat-2 on FYI, MYI and total sea ice.**




| Unit: m | 2019.10-2020.04 | | | 2020.10-2021.04 | | |
| --- | --- | --- | --- | --- | --- | --- |
| month | FYI | MYI | ALL | FYI | MYI | ALL |
| October | 0.456 | -0.0353 | 0.155 | 0.725 | 0.0575 | 0.210 |
| November | 0.164 | -0.137 | 0.0662 | 0.414 | 0.123 | 0.289 |
| December | 0.221 | -0.110 | 0.131 | 0.260 | -0.140 | 0.143 |
| January | 0.115 | -0.103 | 0.0589 | 0.113 | -0.155 | 0.0390 |
| February | 0.136 | -0.203 | 0.0598 | 0.0313 | -0.166 | -0.0228 |
| March | 0.0229 | -0.279 | -0.0274 | 0.0089 | -0.204 | -0.0381 |
| April | -0.146 | -0.654 | -0.207 | -0.184 | -0.435 | -0.222 |
| mean | 0.138 | -0.217 | 0.0338 | 0.195 | -0.131 | 0.0569 |
| MAE | 0.180 | 0.217 | 0.101 | 0.248 | 0.183 | 0.138 |

**Table 8: Schemes for determining SSHAs.**

| Number | Scheme |
| --- | --- |
| 1 | If there are more than 3 observation points per 25-km segment in every track, the average of the 3 lowest values is taken as the SSHA. Otherwise, the SSHA is set to 0. |
| 2 | If there are more than 5 observation points per 25-km segment in every track, the average of the 5 lowest values is taken as the SSHA. Otherwise, the SSHA is set to 0. |
| 3 | If there are more than 7 observation points per 25-km segment in every track, the average of the 7 lowest values is taken as the SSHA. Otherwise, the SSHA is set to 0. |
| 4 | If there are more than 9 observation points per 25-km segment in every track, the average of the 9 lowest values is taken as the SSHA. Otherwise, the SSHA is set to 0. |
| 5 | If there are more than 11 observation points per 25 km segment in every track, the average of the 11 lowest values is taken as the SSHA. Otherwise, the SSHA is set to 0. |
| 6 | If there are more than 3 observation points per 25-km segment in every track, the median of the 3 lowest values is taken as the SSHA. Otherwise, the SSHA is set to 0. |
| 7 | If there are more than 5 observation points per 25-km segment in every track, the median of the 5 lowest values is taken as the SSHA. Otherwise, the SSHA is set to 0. |
| 8 | If there are more than 7 observation points per 25-km segment in every track, the median of the 7 lowest values is taken as the SSHA. Otherwise, the SSHA is set to 0. |
| 9 | If there are more than 9 observation points per 25-km segment in every track, the median of the 9 lowest values is taken as the SSHA. Otherwise, the SSHA is set to 0. |
| 10 | If there are more than 11 observation points per 25-km segment in every track, the median of the 11 lowest values is taken as the SSHA. Otherwise, the SSHA is set to 0. |


**Table 9: Table of differences between the AWI CryoSat-2 radar freeboard and HY-2B radar freeboard values retrieved by different SSHA determination schemes.**



| Unit: m | Oct 2019-Apr 2020 | | | Oct 2020-Apr 2021 | | |
|---|---|---|---|---|---|---|
| | Mean deviation | MAE | SSHA | Mean deviation | MAE | SSHA |
| 1 | 0.1491 | 0.1491 | -0.0817 | 0.1454 | 0.1454 | -0.0689 |
| 2 | 0.0809 | 0.0809 | -0.0593 | 0.0796 | 0.0796 | -0.0495 |
| 3 | 0.0355 | 0.0355 | -0.0459 | 0.0361 | 0.0361 | -0.0385 |
| 4 | 0.00576 | 0.0127 | -0.0364 | 0.00774 | 0.0150 | -0.0307 |
| 5 | -0.0158 | 0.0158 | -0.0295 | -0.0131 | 0.0191 | -0.0249 |
| 6 | 0.1430 | 0.1430 | -0.0793 | 0.1391 | 0.1391 | -0.0668 |
| 7 | 0.0699 | 0.0699 | -0.0553 | 0.0686 | 0.0686 | -0.0459 |
| 8 | 0.0202 | 0.0232 | -0.0401 | 0.0212 | 0.0237 | -0.0337 |
| 9 | -0.011 | 0.0122 | -0.0297 | -0.00863 | 0.0173 | -0.0252 |
| 10 | -0.0328 | 0.0328 | -0.0224 | -0.0291 | 0.0291 | -0.0193 |

**Table 10: Mean sea ice freeboard uncertainties of HY-2B and CryoSat-2 in the common area.**

| Unit: m | Oct 2019-April 2020 | | Oct 2020-April 2021 | |
|---|---|---|---|---|
| | HY-2B | CS-2 | HY-2B | CS-2 |
| October | 0.085 | 0.014 | 0.095 | 0.013 |
| November | 0.10 | 0.015 | 0.10 | 0.014 |
| December | 0.11 | 0.012 | 0.11 | 0.014 |
| January | 0.11 | 0.013 | 0.11 | 0.014 |
| February | 0.10 | 0.015 | 0.12 | 0.015 |
| March | 0.093 | 0.013 | 0.11 | 0.015 |
| April | 0.099 | 0.013 | 0.10 | 0.015 |


**Table 11: Mean sea ice thickness uncertainties of HY-2B and CryoSat-2 in the common area.**

| Unit: m | Oct 2019-April 2020 | | Oct 2020-April 2021 | |
|---|---|---|---|---|
| | HY-2B | CS-2 | HY-2B | CS-2 |
| October | 0.268 | 0.128 | 0.299 | 0.124 |
| November | 0.343 | 0.152 | 0.328 | 0.132 |
| December | 0.377 | 0.121 | 0.375 | 0.135 |
| January | 0.362 | 0.126 | 0.367 | 0.137 |
| February | 0.347 | 0.142 | 0.377 | 0.144 |
| March | 0.319 | 0.129 | 0.355 | 0.147 |
| April | 0.337 | 0.137 | 0.337 | 0.154 |