# Peer review of "Feasibility of Retrieving Arctic Sea Ice Thickness from The Chinese HY-2B Ku-band Radar Altimeter"

_EGUsphere, 2022_

## Referee Comment (RC1)

**Review of**

**"Assessment of Arctic Sea Ice Thickness Retrieval Ability of the Chinese HY-2B Radar Altimeter"**

**By Dong et al.**

The manuscript presents a feasibility study of using the China Ocean Dynamic Environment Satellite Series 2b (HY-2B) radar altimeter for estimating sea ice thickness. The study compares the retrieved radar freeboards, derived sea ice freeboards (using various snow depth products), and derived sea ice thickness with auxiliary data in the form of either airborne observations (Operation IceBridge) or other satellite observations (using CryoSat-2 (CS2) or ICESat-2 (IS2) observations).

At the current state of the manuscript, I believe that more work is necessary before publication. The study could benefit from additional analysis, since the methodology is not convincing, which the results also illustrate. Furthermore, the readability of the manuscript is low, and I believe the figures require more work, to better present the results. Overall, I believe a manuscript that shows the retrieval ability of HY-2B would be beneficial, as it could provide yet another means of estimating thickness from altimeters. Unfortunately, I do not believe that the paper at its current state has presented the retrieval ability of HY-2B, thus requiring major revisions and additions prior to any publication.

**General comments**

I am not convinced of the soundness of the freeboard retrieval methodology presented in this study. While they refer to the study of Kwok et al. (2007) for the retrieval methodology, that study used laser observations rather than radar, which are not as impacted by speckle and noise as radar observations. They also refer to the study of Zhang et al. (2021) that applied a similar (although not entirely same processing chain) to Envisat freeboards, and produced reasonable results, however I am in earnest also not fully convinced of their methodology either. There are some assumptions that I question, and I therefore hope that the authors can present some more results regarding the validity of their methodology – and why they decide to use this methodology instead of the more commonly used methodology of classifying waveforms as either leads/floes based on shape (e.g., Tilling et al. 2018, Ricker et al. 2014, Hendricks and Ricker 2020). This includes for example:

- Can you be sure that the lowest 9 points are in truth from leads? What if all points within the segment are from floes (or negative freeboards, e.g., Figure 4)? If you select them as leads, you will bias the SSHA.
- How big of an impact do you estimate this SSHA methodology to make (e.g., the behavior you point out in Figure 5b+d+f+h)? I appreciate the authors work on investigating different 'SSHA determination schemes', but this does not really question the overall method of using the lowest points, which are to most likely noise/speckle.
- And how can you be sure this methodology is applicable for radar freeboards – especially as you are already observing this biasing of the HY-2B SSHA?

I appreciate that the authors have mentioned in the conclusion that using lead/floe discrimination is expected to be future work, but if you are already aware of several limitations of the applied methodology (and your results also show this in the comparison with Operation IceBridge data), then I would expect a more exhaustive discussion on the applicability of this method.

I do believe that this paper would benefit from investigating the actual waveforms of the radar altimeter to assess the sea ice thickness retrieval ability. Other papers that have assessed such for new sensors (e.g., AltiKa or Sentinel-3) have both employed a lead/floe discrimination algorithm and derived freeboard from here (Armitage and Ridout, 2015; Shen et al., 2020). I encourage the authors to conduct such an analysis in case the radar waveforms are available for the authors. It would not require the authors to compute elevation afterwards using a new or different re-tracker but would simply provide a way of filtering and selecting 'true' leads (after validating the analysis with e.g., SAR or optical data). From there, they can use the leads to

interpolate between and compute the SSHA in a more robust manner. This will also likely provide more comparable freeboard observations. I think such an analysis is crucial to assess HY-2B's retrieval ability.

The study also does not discuss much regarding why CS2 and HY-2B could be different, but merely states that they are. Several aspects could contribute to this e.g., change in footprint or retrieval methodology (re-tracking and SSHA/freeboard retrieval). I do not believe the authors have discussed this aspect thoroughly enough to truly show that HY-2B is 'reliable and consistent with CS2', even if they make a short sentence regarding re-trackers in the conclusion. However, studies have shown that change of re-tracker alone for CS2 introduce significant differences (e.g., Ricker et al. 2014, Landy et al. 2021 etc.), so simply comparing freeboard products from two different satellites using different methods all together warrants at least a discussion.

Overall, I believe the discussion must be expanded. The results of the study are primarily presented, but not discussed in further detail including the aspects that are causing some of these large discrepancies. Especially the less encouraging results with OIB and discrepancies between both HY-2B/CS2 and HY-2B/IS2 should be discussed rather than only presented. What are causing these differences? What do we expect? What will impact this – spatial/temporal coverage? Measurement modes? There are many differences between CS2 and HY-2B that could be causing the discrepancies, and while there is some consistency between H2-YB and CS2, this should still be discussed. It seems the authors are focusing more on positive/agreeable results that the more disheartening ones.

I also wondered why the authors did not compare with Sentinel-3, since Sentinel-3 has the same coverage (till 81.5°N/S) as HY-2B? I know that Sentinel-3's dedicated Sea Ice preprocessors have become available in August 2022 (https://scihub.copernicus.eu/dhus/#/home, they can be found by selecting the product types "SR_2_LAN_HY", "SR_2_LAN_SI", "SR_2_LAN_LI" (respectively for Hydrology, Sea Ice and Land Ice), and that they will reprocess whenever possible aiming for end 2022. It might be that it is not available for the period of the study and in case they are not, this could be a suggestion for a future comparison study as well.

**Specific comments**

*Overall*

There appears to be some issues with the referencing/citation software used throughout the article. In several instances (e.g., line 150, "Rasmus Tonboe et al. (2016)"; line 158, "Signe et al., (2021)") the citation is written with either both first and last name of the first-author, or by only first name of the first author. This should be corrected – while it may be an issue with LaTeX, I believe it is the responsibility of the author to correct this.

I also encourage you to do a thorough proof-reading and edit, especially in the results, discussion, and conclusion sections. There are many unnecessary repetitions, and by removing these and constraining the text, it would improve the readability of the manuscript.

All acronyms should be checked whether they are properly named and that they are explained when appearing first in the text. I have written some examples in 'technical corrections', but the authors should check the entire manuscript.

*Figures and tables*

The manuscript includes a lot of figures and tables. I encourage the authors to consider ways to limit the figures/tables and more concisely present the results. Also, some figures could be provided in an appendix or as supporting information.

Not all figures include sub-figure labels (e.g., Figure 4). Also, several figures have sub-figure labels given in the x-axis labels – I encourage the authors to include the sub-figure labels in the top left corner instead. However, in case they keep the current format, they should include a space between the sub-figure label and the x-axis label. Also, the manner of which units are displayed in the labels and in legends do not follow The Cryosphere guidelines (e.g., currently have no spaces between the units and the numbers).

Figure 6, 8, 14, 15: Consider a different colormap to ensure readers with color vision deficiencies can correctly interpret your findings (as per The Cryosphere submission guidelines). 'Jet' or 'rainbow' are not encouraged as colormaps. Also, I think the maps are too small – it's hard to see what is going on. I suggest enlarging the maps and use only 1 color-bar for the entire map. This will reduce the space between the maps, allowing to increase the size too.

Figure 1 + 2, 7 + 9, and 10 + 11: I suggest combining the figures – just to reduce the sheer number of figures. It would just generate several subplots. This would of course require some slight alteration of the text to ensure that the sequence of which figures are referenced are still in the correct order.

*Open data and research*

I was not able to access the FTP folder with the SDGR HY-2B data; neither using explorer nor filezilla. Therefore, I was not able to investigate the data available nor whether potential lower-level products were available for the authors to conduct a waveform analysis. I encourage the authors to ensure the data is available through the link, and/or provide the actual data used in the publication through a database (raw and processed data if possible) with associated DOI. I also encourage the authors to make the code available for reproducibility and transparency.

However, I thank the authors for providing the link to all other data including date of access! Refreshing to see in a publication and right in line with the expectations of the science community regarding open research.

*Abstract*

I encourage the authors to shorten the abstract. 7 lines about the general background of satellite altimetry and using more than 10 lines on a summary of the results seems a bit too much. At the same time, I believe only the main results (with one or two numerical results) would be enough to underline the points made. Future work/limitations of study should be provided in the abstract as well.

*Introduction*

Line 52: "Sea ice thickness, as the third dimension of sea ice (…)": what is meant by sea ice density here? Surely, sea ice thickness can in combination with sea ice area be used to calculate the sea ice volume, but sea ice density cannot be combined with sea ice thickness to compute sea ice volume.

Line 68. "Shen et al. (2020) (…)": Lawrence et al. (2019) showed that this was mostly a result of the processing chain of Sentinel-3 not having included zero-padding or Hamming-weighting. I believe this should be mentioned, since the study of Lawrence et al. (2019) in which these corrections were applied showed greater consistency.

*Data*

Line 105. What type of re-tracking algorithm is used and what are potential impacts of using this re-tracker (limitations that other studies have shown? etc.)?

Line 113-117: Baseline-D ESA and AWI use different re-tracking algorithms which will have different results. As such, your comparison is not 'consistent' in the way that the products are not the same and any differences you observe between the HY-2B and AWI/ESA could be different depending on re-tracker. A suggestion could be to investigate the Climate Change Initiative (CCI) products, where both trajectory and gridded products are available – and they use the same processing chain (ftp://ftp.awi.de/sea_ice/projects/cci/crdp/v3p0-rc1/cryosat2/nh/).

Line 162. How come you use DTU18MSS and not the new (and likely improved) DTU21MSS? Available here (https://doi.org/10.11583/DTU.19383221.v1): https://data.dtu.dk/articles/dataset/DTU21_Mean_Sea_Surface/19383221

Line 164. What is meant by "this model can precisely determine the instantaneous elevation of lead"? I do not believe that the referenced study (Skourup et al., 2017) has stated this. A mean sea surface represents the sea level due to constant phenomena and represents the position of the ocean surface averaged over an appropriate time period to remove annual, semi-annual, seasonal, and spurious sea surface height signals – I am not sure what you mean by the statement. Surely, if the model could determine the instantaneous elevation of a lead, we could just interpolate the model along the floe observations and not care about the individual lead observations along the track? The sentence should be corrected.

Line 190. You already have a lot of tables. Instead of providing a table with the exact densities for each month, instead you could consider just writing the equation in the section.

*Methodology*

Section 3.1: Figure 3 shows a removal of 'outliers' using $\pm 1$ m, but this is not mentioned in the text.

Line 198. What is meant by "invalid values". How are they considered invalid?

Line 198-201. Which exact corrections (providers) are you using? This should be stated.

Line 207-213. The purpose of removing the residuals should be stated. This is not commonly applied for the lead/floe retrieval methodologies, and as such can add to the reasons why the results differ.

Line 215. Why are you assuming the SSHA to be 0? And what is the impact of this? Figure 4+5 show that SSHA is not necessarily zero (zero-elevation is only relative to MSS). Instead, could you consider nan (not-a-number) the data and just interpolate where data is not available?

Line 226. Isn't the 0.22 value depending on snow density and thus varying, even in the AWI product? It is based on slower wave propagation, given by $\left(\frac{c}{c_s} - 1\right)$, where c is the speed of light and c is the speed of light through snow. Here, $c_s = c \cdot \eta_s$, where $\eta_s$ depends on the snow density (which you vary when using Mallet et al. (2020)). If so, this should be corrected. If not, please explain exactly what this value is and where it comes from.

Line 229. How is the gridding performed?

*Results*

Line 237-249: "Fig 5(a) (….)." This entire paragraph could benefit from some rephrasing, as it was confusing and long to read. Also, check the manuscript guidelines for The Cryosphere again to ensure that the way you write the different expressions ("0$\pm$0.31m") follows the guidelines.

Line 264-265. I think you need to consider what you mean by 'smallest' and 'largest' deviation. Sure, in actual numbers, -2.4 cm is smallest, and 1.9 cm is largest. But the deviation from zero must mean that the -2.4 cm is largest! So, this needs to be rephrased for clarity.

Line 336 -358. You mention the Kwok snow depth product is used, but in Figure 3 you mention that the AWI snow depth is used for the processing of the radar freeboards. Then, later you mention that the Kwok snow depth is used for the ICESat-2 freeboards to obtain the sea ice freeboard and for the CS2 to obtain sea ice freeboard. How come you decide to use this snow depth product suddenly, and not just use the AWI snow depth throughout?

Line 274-275. The fact that HY-2B modal freeboard is thicker than the mean is not expected – we usually expect a skewed distribution with an abundance of thinner freeboards compared to the thicker freeboards. This should be discussed further.

Line 337-338: Be careful when discussing spatial resolution of IS2. The footprint of IS2 is approximately 11 m, yes, but the ATL10/ATL20 products are processed data that is not of this spatial resolution.

Line 358. You mention sea ice thickness error? What is meant by this – this has not been discussed previously.

Figure 13. For almost all subfigures, we observe HY-2B to generate significantly thicker sea ice thickness than IS2 (CS2 also does not observe this behavior). This seem to showcase that some overestimation of thickness from HY-2B, which I think could be discussed more in the text for the IS2 comparison.

*Discussion*

Line 371-374. Long and confusing sentence, however it sounds like it is saying that in general HY-2B is observing smaller freeboards than the AWI freeboards – but, when comparing otherwise you said that HY-2B was higher than CS2 freeboard (I assume this is the ESA observations then). I suggest you rephrase this sentence for clarity, and then discuss further the fact that you are comparing observations with different re-trackers – what can you really get from this? And are you comparing along-track or gridded? This should also be mentioned.

Line 386. If you are following Mallet et al. (2020), the 0.22 value should be changing with density.

Line 393, Equation 8. I am intrigued by why you are including the impact of $+0.22\rho_w$, when e.g., the study of Ricker et al. 2014 did not. This may explain my comment further down regarding line 415-425. I suggest you check up on this to make sure that this is the correct partial derivative.

Line 404. The uncertainty of sea ice densities should be cited.

Line 415-425. Something seems odd in the uncertainty determination. Ricker et al. (2014) determined freeboard uncertainties of 6 cm (12 cm) for FYI (MYI), but for sea ice thickness, they estimate 60 cm (120 cm) (increase by factor 10). This is not observed in your estimation, where the CS2 sea ice thickness uncertainty is only 12.1-15.4 cm. Also, your freeboard uncertainties to thickness uncertainties do not increase by factor 10 for HY-2B observations? Something seems off here. I suggest going through this again and ensuring that the units are correct.

*Conclusion*

Line 442-44: I am not convinced by this methodology, when the comparisons and correlations with OIB are so low compared with CS2. A correlation of 0.58 for HY-2B but 0.84 for CS2 for freeboards, and 0.41 for HY-2B but 0.80 for CS2 on thickness is quite a big difference – and for me just shows that the method used here is not that applicable for sea ice, and not reliable.

Line 460-468: A lot of new information provided here which is not discussed or even commented upon in the text. I strongly encourage the authors to include a discussion (in the discussion section) on the impact of re-tracking methodology applied to the HY-2B freeboards (and how it differs from the CS2 re-tracking). There are also some considerations for what should be done in future work, which are not discussed in the text but only here. Finally, the authors conclude that they have used a sub-optimal re-tracking algorithm which is only applicable for ocean (not sea ice?). Makes me question why they did this analysis with this algorithm, when it is not suitable, and the results clearly show this too.

**Technical corrections**

Line 113: "main" – not sure they are the only main products? The CCI product could be mentioned here. Furthermore, the upcoming releases of CryoTEMPO (http://cryosat.mssl.ucl.ac.uk/tempo/downloads.html) are expected to be a favorable product to be used in the future by the science community.

Line 116. What is meant by L2I? The intermediate product is not provided to the public, but rather the L2 product is.

Line 123. "were by " -> "were provided by"

Line 148. "Denmark Technical University (DTU)" -> "the Technical University of Denmark (DTU)"

Line 171. "W99" – "Warren et al. 1999 (W99)": reference the publication and explain the acronym.

Line 171. "AMSR-2" -> "Advanced Microwave Scanning Radiometer-2 (AMSR-2)"

Line 202. "were interpolated" -> "were linearly interpolated" (or however it was interpolated)

Line 204. "was not used" -> "was not considered"

Line 206-207. "from the obtained surface elevations of ground objects (…)": consider rephrasing this sentence, as it is not clear.

Line 215-216. "(…) were subtracted from the SSHA to obtain": The SSHA are subtracted from the segment height, not the other way around. This sentence should be rephrased.

Line 231: "$\rho_{seawater}$" -> "$\rho_w$" (following the equation)

Line 298-299. "Because the (…)": I wouldn't say that the lower SSHA necessitates higher freeboards, because they are only higher if the elevations of the floes are of the same height as CS2 or higher. If the floe elevations were also lower, HY-2B would not necessarily have higher freeboards. Consider rephrasing it.

Line 451-452. You mention sea ice error again, but you have merely referenced this by the paper of Ricker et al. (2014) and not done more sensitivity studies on this. I suggest removing this sentence or discuss this more in the text if you aim to keep it.

**References**

Armitage, T. W. K., and Ridout, A. L. (2015), Arctic sea ice freeboard from AltiKa and comparison with CryoSat-2 and Operation IceBridge, Geophys. Res. Lett., 42, 6724– 6731, doi:10.1002/2015GL064823.

Hendricks, S. and Ricker, R. (2020): Product User Guide & Algorithm Specification: AWI CryoSat-2 Sea Ice Thickness (version 2.3)

Lawrence, I.S., Thomas W.K. Armitage, Michel C. Tsamados, Julienne C. Stroeve, Salvatore Dinardo, Andy L. Ridout, Alan Muir, Rachel L. Tilling, Andrew Shepherd, (2019) Extending the Arctic sea ice freeboard and sea level record with the Sentinel-3 radar altimeters, Advances in Space Research, Volume 68, Issue 2, doi: 10.1016/j.asr.2019.10.011.

Ricker, R., Hendricks, S., Helm, V., Skourup, H., and Davidson, M.: Sensitivity of CryoSat-2 Arctic sea-ice freeboard and thickness on radar-waveform interpretation, The Cryosphere, 8, 1607–1622, https://doi.org/10.5194/tc-8-1607-2014, 2014.

Shen , X., Chang-Qing Ke, Hongjie Xie, Mengmeng Li & Wentao Xia (2020) A comparison of Arctic sea ice freeboard products from Sentinel-3A and CryoSat-2 data, International Journal of Remote Sensing, 41:7, 2789-2806, DOI: 10.1080/01431161.2019.1698078

Tilling, R.L., Andy Ridout, Andrew Shepherd, Estimating Arctic sea ice thickness and volume using CryoSat-2 radar altimeter data, (2018) Advances in Space Research, Volume 62, Issue 6, doi: 10.1016/j.asr.2017.10.051.

---

## Author Comment (AC1)

Dear Editors and Reviewers,

Thank you for considering our manuscript, and the reviewers' comments concerning our manuscript entitled Assessment of Arctic Sea Ice Thickness Retrieval Ability of the Chinese HY-2B Radar Altimeter (manuscript ID: egusphere-2022-870). The comments are all valuable and very helpful for improving upon our paper. We have now carefully reviewed and addressed all of comments which we hope meet with approval, with revisions to the original manuscript shown in red. The primary corrections in the paper and the responds to the reviewer's comments are as flowing:

**Responses to Reviewer's Comments:**

**Reviewer #1:**

**General comments**

According to your valuable comments, we change the retrieval method of HY-2B. If there are more than 15 points within 25 km section, we take the average of the 15 lowest values as the SSHA. Otherwise set as nan, and interpolate the SSHA along track. We find that the correlation between the retrieved sea ice freeboard and OIB sea ice freeboard is 0.77, with a RMSE is 0.13 m, and a MAE is 0.12 m. Therefore, we update our method and find the optimal threshold of 15 points within 25 km segment. If there are more than 15 points within 25 km section, we take the average of the 15 lowest values as the SSHA, otherwise set as nan, and interpolate the SSHA along track.

**GC0: I am not convinced of the soundness of the freeboard retrieval methodology presented in this study.** While they refer to the study of Kwok et al. (2007) for the retrieval methodology, that study used laser observations rather than radar, which are not as impacted by speckle and noise as radar observations. They also refer to the study of Zhang et al. (2021) that applied a similar (although not entirely same processing chain) to Envisat freeboards, and produced reasonable results, however I am in earnest also not fully convinced of their methodology either. There are some assumptions that I question, and **I therefore hope that the authors can present some more results regarding the validity of their methodology – and why they decide to use this methodology instead of the more commonly used methodology of classifying waveforms as either leads/floes based on shape** (e.g., Tilling et al. 2018, Ricker et al. 2014, Hendricks and Ricker 2020). This includes for example:

• **Can you be sure that the lowest 9 points are in truth from leads? What if all points within the segment are from floes (or negative freeboards, e.g., Figure 4)? If you select them as leads, you will bias the SSHA.**

**Response:** Thank you for the valuable comments. The HY-2B altimeter uses two different tracking modes: suboptimal maximum likelihood estimation (SMLE) and offset center of gravity (OCOG). The two tacking modes can exchange according to the observation

surfaces. The HY-2B Level-2 altimetry products (SGDR products) we used do not have OCOG data.

Pulse peakiness (PP) reflects the acuteness of the shape of the returned waveform peak. Lead-reflected waveforms have a peakier shape with higher power than those reflected by ice. Consequently, PP has been widely used to distinguish leads from ice floes in pulse limited radar altimeter data (Peacock et al., 2004; Connor et al., 2009) and SAR altimeter data (Laxon et al., 2013).

As shown in Figure 1 and Figure 2, we used PP value greater than 10 to obtain the SSHAs using nearest interpolation but did not yield satisfactory results. In addition, we only extracted 4 lead points on a track number of 14418 (1118 floe points) according to PP value greater than 10. We also used other PP thresholds (3, 7 and 15) to extract lead points (461, 11 and 2), but didn't obtain satisfactory results.

[Figure]

Figure 1. A sample of the HY-2B elevation profile obtained for of track number 14418 on April 4, 2020. The green points in panel (a) are the relative elevation (h) values; the blue points in panel (a) are the $h_{25km}$ values, defined as the 25-km running mean of h; the black points in panel (b) are the modified relative elevation ($h_r$) values; the red points in panel (b) are the sea surface height anomaly (SSHA) values; the azury points in panel (b) are lead points extracted by PP values greater 10; and the black points in panel (c) are the radar freeboard values.

[Figure]

Figure 2. Radar freeboard retrieved by PP value greater than 10 in April 2020.

Additionally, we attempted to extract leads using MODIC lead products (Hoffman et al., 2019, obtained at ftp://frostbite.ssec.wisc.edu) to retrieve the radar freeboard, but we were not successful, as shown in Figure 3 and Figure 4. We think there are three possible reasons. Firstly, the SGDR products have been re-tracked for Brown model, which isn't suitable for the retrack of sea ice. The retrack algorithm is different from other retrack algorithms, such as TFMRA (threshold first maximum retracker algorithm) and SAMOSA+ (Synthetic Aperture Radar Altimetry MOde Studies and Applications +). Secondly, the SGDR products have rare effective data points in each track, resulting in rare lead points being extracted using these two methods as well. Consequently, there is a significant error in the interpolation of the SSHAs caused by the rare leads. Thirdly, the classification of floe and lead isn't precise, which we need to improve the algorithm of classification.

[Figure]

Figure 3. A sample of the HY-2B elevation profile obtained for of track number 14418 on April 4, 2020. The green points in panel (a) are the relative elevation (h) values; the blue points in panel (a) are the $h_{25km}$ values, defined as the 25-km running mean of h; the black points in panel (b) are the modified relative elevation ($h_r$) values; the red points in panel (b) are the sea surface height anomaly (SSHA) values; the azury points in panel (b) are lead points extracted by MODIC lead product; and the black points in panel (c) are the radar freeboard values.

[Figure]

Figure 4. Radar freeboard retrieved by MODIC lead product in April 2020.

Honestly, I can't sure that the lowest 9 points are in truth from leads. Radar freeboard will be lower if all points within the segment are from floes (or negative freeboards), especially leads are sparse in late winter and spring. According to Tilling et al. (2019), the quality of the interpolated SSH will depend on the number of available lead measurements. When the number of valid data points in each track of the data set is smaller, this method will generate a smaller deviation than the two previous approaches. We are so sorry for that we didn't retrieve the satisfactory results based on PP value. But we will try to use HY-2B L1 data to retrieve radar freeboard using waveform feature to extract lead, which will be described in detail in subsequent articles. The quality of the interpolated SSHA will be improved with more available lead measurements.

• **How big of an impact do you estimate this SSHA methodology to make (e.g., the behavior you point out in Figure 5b+d+f+h)? I appreciate the authors work on investigating different 'SSHA determination schemes', but this does not really question the overall method of using the lowest points, which are to most likely noise/speckle.**

**Response:** Thank you for the valuable comments. Figure 5 shows the normal distribution of radar freeboard difference between HY-2B and CS-2. We found that the total radar freeboard from HY-2B is slightly lower than for the AWI CS-2 products (-0.0068±0.14 m). The FYI radar freeboard from HY-2B is slightly higher than the AWI CS-2 products (0.0055±0.13 m). But the MYI radar freeboard from HY-2B is slightly lower than the AWI CS-2 products (-0.037±0.12 m). A higher radar freeboard is retrieved by this method in early winter since more points within the segment are lead; a lower radar freeboard is retrieved in late winter and spring since more points within the segment are floe. The HY-2B MYI is thinner than AWI CS-2. As the lead points in the MYI region are sparser

than those in FYI. To achieve the minimum error in radar freeboard, we investigate different "SSHA determination schemes".

[Figure]

Figure 5. Normal distribution of radar freeboard differences between HY-2B and CS-2 from Oct 2019 to Apr 2020 and from Oct 2020 to Apr 2021, (a) FYI, (b) MYI, (c) ALL.

- **And how can you be sure this methodology is applicable for radar freeboards – especially as you are already observing this biasing of the HY-2B SSHA?**

**Response:** Thank you for the valuable comments. As compared to the two previous approaches, the method of the 9 (or 15) lowest points is capable of retrieving Arctic sea ice freeboards with high accuracy in SGDR products. Therefore, we adopt it to retrieve Arctic radar freeboards. The SGDR products had been retracked by the Brown model. And the valid measurement points are sparse, resulting in the error of nearest interpolation is higher than the error of the method of 9 (or 15) lowest points.

**GC1:** I appreciate that the authors have mentioned in the conclusion that using lead/floe discrimination is expected to be future work, but **if you are already aware of several limitations of the applied methodology (and your results also show this in the comparison with Operation IceBridge data), then I would expect a more exhaustive discussion on the applicability of this method.**

**Response:** Thank you for the valuable comments. A lower radar freeboard is retrieved in late winter and spring, especially in MYI region since more points within the 25 km segment are floe. Radar freeboard might be low in the MYI region with a mean deviation of -0.037 m, as shown in Figure 5. This method also cannot reliably capture the seasonal variation of the radar freeboard. Since the radar freeboard is lower in March and April than it actually is (Line 260-267, 316-321).

**GC2: I do believe that this paper would benefit from investigating the actual waveforms of the radar altimeter to assess the sea ice thickness retrieval ability.** Other papers that have assessed such for new sensors (e.g., AltiKa or Sentinel-3) have both employed a lead/floe discrimination algorithm and derived freeboard from here (Armitage and Ridout, 2015; Shen et al., 2020). **I encourage the authors to conduct such an analysis in case the radar waveforms are available for the authors. It would not require the authors to compute elevation afterwards using a new or different re-tracker but would simply provide a way of filtering and selecting 'true' leads (after validating the analysis with e.g., SAR or optical data). From there, they can use the leads to interpolate between and compute the SSHA in a more robust manner. This will also likely provide more comparable freeboard observations. I think such an analysis is crucial to assess HY-2B's retrieval ability.**

**Response:** Thank you for the valuable comments. In SGDR product, SSHAs interpolated from PP values greater than 10 did not yield satisfactory results, as shown in Figure 2 and 3. In addition, we attempted to extract leads using MODIC lead products, but failed to attain the reasonable Arctic radar freeboard, as shown in Figure 4 and 5. **As a result of using the HY-2B L1 product, we first tried to extract lead by using PP values greater than 15 and to obtain SSHA using nearest interpolation. The Arctic radar freeboard of HY-2B L1b using the retrack algorithm of TFMRA 50 is shown in Figure 6. The retrieved Arctic radar freeboard is reasonable compared with the SGDR product, which will be described in detail in subsequent articles.** With more available lead measurements, the quality of the interpolated SSHA will be improved, resulting to retrieve reasonable radar freeboard. Meanwhile, we use an implementation of the Threshold First Maximum Retracker Algorithm (TFMRA) to estimate the range to the main scattering horizon for each waveform. The retrack algorithm is different from the Brown modal. Moreover, it is difficult to obtain reasonable radar freeboard using SGDR products since it has sparse measurement points and the error of interpolation SSHA is larger than the method of 9 (or 15) lowest points.

[Figure]

Figure 6. Radar freeboard retrieved by PP value greater than 15 using HY-2B L1 data in January 2020.

**GC3: The study also does not discuss much regarding why CS2 and HY-2B could be different, but merely states that they are. Several aspects could contribute to this e.g., change in footprint or retrieval methodology (re-tracking and SSHA/freeboard retrieval). I do not believe the authors have discussed this aspect thoroughly enough to truly show that HY-2B is 'reliable and consistent with CS2', even if they make a short sentence regarding re-trackers in the conclusion.** However, studies have shown that change of re-tracker alone for CS2 introduce significant differences (e.g., Ricker et al. 2014, Landy et al. 2021 etc.), **so simply comparing freeboard products from two different satellites using different methods all together warrants at least a discussion.**

**Response:** Thank you for the valuable comments. We think there are three reasons for these discrepancies. Firstly, the SGDR products have been retracked for Brown model, which isn't suitable for the retrack of sea ice. Secondly, the SGDR products have sparse effective data points in each track, resulting in rare lead points being extracted using these two methods as well. Consequently, there is a significant error in the interpolation of the SSHAs caused by the rare leads. The error of interpolation SSHA is larger than the assumption of 9 (or 15) lowest points. Thirdly, we didn't classify the floe and lead, but Robert et.al (2014) used waveform characteristics to accurately distinguished floes and leads (Line 285-287, 345-346).

**GC4: Overall, I believe the discussion must be expanded. The results of the study are primarily presented, but not discussed in further detail including the aspects that are causing some of these large discrepancies. Especially the less encouraging results with OIB and discrepancies between both HY-2B/CS2 and HY-2B/IS2 should be discussed rather than only presented. What are causing these differences? What do we expect? What will impact this – spatial/temporal coverage? Measurement modes? There are**

**many differences between CS2 and HY-2B that could be causing the discrepancies, and while there is some consistency between HY-2B and CS2, this should still be discussed. It seems the authors are focusing more on positive/agreeable results that the more disheartening ones.**

**Response:** Thank you for the valuable comments. We think there are three reasons for these discrepancies. Firstly, the SGDR products have been retracked for Brown model, which isn't suitable for the retrack of sea ice. Secondly, the SGDR products have rare effective data points in each track, resulting in rare lead points being extracted using these two methods as well. Consequently, there is a significant error in the interpolation of the SSHAs caused by the rare leads. Thirdly, we didn't classify the floe and lead, but Robert et.al (2014) used waveform characteristics to accurately distinguished floes and leads (Line 285-287, 345-346). We expect to use SGDR products to retrieve sea ice freeboard and provide experience for a more comprehensive assessment of HY-2B application potential using the L1 data.

The orbital inclination and the satellite altitude will affect the coverage area and repeat cycle of the satellite. The measurement mode of the satellite will affect the size of the measurement footprint. These will affect the spatial and temporal resolution of sea ice freeboard products.

The spatial patterns of the two sea ice thickness products exhibited broad agreement; thicker sea ice occurred north of the Canadian Archipelago, while thinner sea ice occurred in the Eurasian continental marginal sea and Baffin Bay. Since the height of the lead is usually lower than the height of the adjacent floes, this method is reasonable to where there are many lead points in the 25 km segment (Line 260-261).

**GC5: I also wondered why the authors did not compare with Sentinel-3, since Sentinel-3 has the same coverage (till 81.5° N/S) as HY-2B? I know that Sentinel-3's dedicated Sea Ice preprocessors have become available in August 2022 (https://scihub.copernicus.eu/dhus/#/home, they can be found by selecting the product types "SR_2_LAN_HY", "SR_2_LAN_SI", "SR_2_LAN_LI" (respectively for Hydrology, Sea Ice and Land Ice), and that they will reprocess whenever possible aiming for end 2022. It might be that it is not available for the period of the study and in case they are not, this could be a suggestion for a future comparison study as well.**

**Response:** Thank you for the valuable comments. We will conduct a comparative study between HY-2B and Sentinel-3 using HY-2B L1 data.

**Specific comments:**

*Overall*

**There appears to be some issues with the referencing/citation software used throughout the article. In several instances (e.g., line 150, "Rasmus Tonboe et al. (2016)"; line 158, "Signe et al., (2021)") the citation is written with either both first and last name of the first-author, or by only first name of the first author. This should be corrected – while it may be an issue with LaTeX, I believe it is the responsibility of the author to correct this.**

**Response:** Thank you for the valuable comments. We corrected all author names throughout the manuscript. We have revised 'Thomas, D. N' to 'Thomas' (line 29), 'R. Ricker' to 'Ricker' (line 52) and 'Rasmus Tonboe' to 'Rasmus' (line 144).

**I also encourage you to do a thorough proof-reading and edit, especially in the results, discussion, and conclusion sections. There are many unnecessary repetitions, and by removing these and constraining the text, it would improve the readability of the manuscript.**

**Response:** Thank you for the valuable comment. We have revised the manuscript to improve the readability of the manuscript (red fonts in the results, discussion, and conclusion sections).

**All acronyms should be checked whether they are properly named and that they are explained when appearing first in the text. I have written some examples in 'technical corrections', but the authors should check the entire manuscript.**

**Response:** Thank you for the valuable comment. We have checked all acronyms in the entire manuscript.

*Figures and tables*

**The manuscript includes a lot of figures and tables. I encourage the authors to consider ways to limit the figures/tables and more concisely present the results. Also, some figures could be provided in an appendix or as supporting information.**

**Response:** Thank you for the valuable comment. We have combined figure 1 and figure 2, figure 7 and figure 9, and figure 10 and figure 11 (figure 1, figure 7, figure 8). We have replaced table 2 with equation (1) (Line 187).

**Not all figures include sub-figure labels (e.g., Figure 4). Also, several figures have sub-figure labels given in the x-axis labels – I encourage the authors to include the sub-figure labels in the top left corner instead. However, in case they keep the current format, they should include a space between the sub-figure label and the x-axis label. Also, the manner of which units are displayed in the labels and in legends do not follow The Cryosphere guidelines (e.g., currently have no spaces between the units and the numbers).**

**Response:** Thank you for the valuable comment. We have supplied the sub-figure labels in figure 4. We also have kept a space between the units and the numbers in all figures.

**Figure 6, 8, 14, 15: Consider a different colormap to ensure readers with color vision deficiencies can correctly interpret your findings (as per The Cryosphere submission guidelines). 'Jet' or 'rainbow' are not encouraged as colormaps. Also, I think the maps are too small – it's hard to see what is going on. I suggest enlarging the maps and use only 1 color-bar for the entire map. This will reduce the space between the maps, allowing to increase the size too.**

**Response:** Thank you for the valuable comment. We have revised the colormap of figure 5, 6,11 and 12 with only 1 color-bar for the entire map.

**Figure 1 + 2, 7 + 9, and 10 + 11: I suggest combining the figures – just to reduce the sheer number of figures. It would just generate several subplots. This would of course require some slight alteration of the text to ensure that the sequence of which figures are referenced are still in the correct order.**

**Response:** Thank you for the valuable comment. We combine figure 1 and figure 2, figure 7 and figure 9, and figure 10 and figure 11. We also make some alteration of the text to ensure that the figures are referenced are still in the correct order.

*Open data and research*

**I was not able to access the FTP folder with the SDGR HY-2B data; neither using explorer nor filezilla. Therefore, I was not able to investigate the data available nor whether potential lower-level products were available for the authors to conduct a waveform analysis. I encourage the authors to ensure the data is available through the link, and/or provide the actual data used in the publication through a database (raw and processed data if possible) with associated DOI. I also encourage the authors to make the code available for reproducibility and transparency.** However, I thank the authors for providing the link to all other data including date of access! Refreshing to see in a publication and right in line with the expectations of the science community regarding open research.

**Response:** We are sorry for the confusion. You may need to register an account by https://osdds.nsoas.org.cn/register. Then, you can enter your account and password to log in to the official website to access the FTP folder with SDGR HY-2B data using filezilla (ftp://osdds-ftp.nsoas.org.cn/). The SGDR HY-2B data can also be accessed through https://osdds.nsoas.org.cn/MarineDynamic/ (Figure 7). You may need a VPN if you were not successful in logging in. We will share the main code in the internet.

[Figure]

Figure 7. The website of HY-2B satellite.

*Abstract*

**I encourage the authors to shorten the abstract. 7 lines about the general background of satellite altimetry and using more than 10 lines on a summary of the results seems a bit too much. At the same time, I believe only the main results (with one or two numerical results) would be enough to underline the points made. Future work/limitations of study should be provided in the abstract as well.**

**Response:** Thank you for the valuable comment. We have revised the abstract to 'With the continuous development of the China Ocean Dynamic Environment Satellite Series (Haiyang-2, HY-2), it is urgent to explore the potential application of the HY-2B in Arctic sea ice thickness retrievals. In this study, we first derive the Arctic radar freeboard and sea ice thickness during two cycles (from October 2019 to April 2020 and from October 2020 to April 2021) using HY-2B radar altimeter and compare the results with the Alfred Wegener Institute (AWI) CS-2 products. We evaluate our HY-2B sea ice freeboard and thickness products using Operation IceBridge (OIB) airborne data and ICESat-2 products. Finally, we estimate the uncertainties in the HY-2B sea ice freeboard and sea ice thickness. The radar freeboard deviation between HY-2B and CS-2 is within 0.02 m, whereas the sea ice thickness deviation between HY-2B and CS-2 is within 0.2 m. The HY-2B radar freeboards are generally thicker than AWI CS-2, except in spring (March and April). In spring, more of the lowest 15 points within 25 km segment are likely to originate from floes, while more points may originate from leads in early winter. We also find that the deviations of radar freeboard and sea ice thickness between HY-2B and CS-2 over MYI are larger than over FYI. The correlation between HY-2B sea ice freeboard retrievals and OIB values is 0.77, with a a root mean square error (RMSE) is 0.13 m and a mean absolute

error (MAE) is 0.12 m. The correlation between HY-2B sea ice thickness retrievals and OIB values is 0.65, with a RMSE is 1.86 m and a MAE is 1.72 m. The HY-2B sea ice freeboard uncertainty values range from 0.021 m to 0.027 m, while the uncertainties in the HY-2B sea ice thickness range from 0.44 m to 0.67 m.' (line 1-23).

*Introduction*

**Line 52: "Sea ice thickness, as the third dimension of sea ice (···)" : what is meant by sea ice density here? Surely, sea ice thickness can in combination with sea ice area be used to calculate the sea ice volume, but sea ice density cannot be combined with sea ice thickness to compute sea ice volume.**

**Response:** We apologize for the mistake. We have revised 'sea ice density' to 'sea ice area' (line 40).

**Line 68. "Shen et al. (2020) (···)" : Lawrence et al. (2019) showed that this was mostly a result of the processing chain of Sentinel-3 not having included zero-padding or Hamming-weighting. I believe this should be mentioned, since the study of Lawrence et al. (2019) in which these corrections were applied showed greater consistency.**

**Response:** Thank you for the valuable comment. We have revised 'Shen et al. (2020) retrieved Arctic sea ice freeboard data based on Sentinel-3A records and analyzed the differences and consistency between the results and the corresponding CS-2 data. The results showed that the Sentinel-3A sea ice freeboard was generally lower than that indicated by CS-2' to 'Shen et al. (2020) used Sentinel-3A to retrieve Arctic sea ice freeboard and analyzed the differences and consistency between Sentinel-3A and CS-2. The results showed that the Sentinel-3A sea ice freeboard was generally lower than that retrieved by CS-2. The differences between Sentinel-3A and CS-2 are mostly a result of the processing chain of Sentinel-3 not having included zero-padding or Hamming-weighting. The study of Lawrence et al. (2019) in which these corrections were applied showed greater consistency' (line 55-59).

*Data*

**Line 105. What type of re-tracking algorithm is used and what are potential impacts of using this re-tracker (limitations that other studies have shown? etc.)?**

**Response:** Thank you for the valuable comment. The SGDR products have been retracked for the Brown model, which is not suitable for sea ice. The inapposite retrack algorithm maybe result the residual error and abnormal values.

**Line 113-117: Baseline-D ESA and AWI use different re-tracking algorithms which will have different results. As such, your comparison is not 'consistent' in the way that the products are not the same and any differences you observe between the HY-2B and AWI/ESA could be different depending on re-tracker. A suggestion could be to**

**investigate the Climate Change Initiative (CCI) products, where both trajectory and gridded products are available – and they use the same processing chain (ftp://ftp.awi.de/sea_ice/projects/cci/crdp/v3p0-rc1/cryosat2/nh/).**

**Response:** Thank you for the valuable comment. Due to the trajectory products of CCI don't contain SSHA and the height of floe before subtracting SSHA, we can't compare them with HY-2B, so we adopt the Baseline-D products of ESA.

**Line 162. How come you use DTU18MSS and not the new (and likely improved) DTU21MSS?** **Available** **here** **(https://doi.org/10.11583/DTU.19383221.v1):** **https://data.dtu.dk/articles/dataset/DTU21_Mean_Sea_Surface/19383221**

**Response:** Thank you for the valuable comment. Since DTU21 (the version of 1 min) is much larger than DTU18 (the version of 2 mins), calculation efficiency is low. According to Figure 8 and Figure 9, the DTU 21 does not significantly improve our results. Therefore, we used the DTU 18 (the version of 2 mins) to process the HY-2B to compare with AWI.

[Figure]

Figure 8. Comparison between two kinds of sea ice freeboard products and the OIB sea ice freeboard values collected in April 2019: (a) the HY-2B sea ice freeboard retrieved by DTU 21 and (b) the AWI CS-2 sea ice freeboard.

[Figure]

Figure 9. Comparison between two kinds of sea ice freeboard products and the OIB sea ice freeboard values collected in April 2019: (a) the HY-2B sea ice freeboard retrieved by DTU 18 and (b) the AWI CS-2 sea ice freeboard.

**Line 164. What is meant by "this model can precisely determine the instantaneous elevation of lead"? I do not believe that the referenced study (Skourup et al., 2017) has stated this. A mean sea surface represents the sea level due to constant phenomena and represents the position of the ocean surface averaged over an appropriate time period to remove annual, semi-annual, seasonal, and spurious sea surface height signals – I am not sure what you mean by the statement. Surely, if the model could determine the instantaneous elevation of a lead, we could just interpolate the model along the floe observations and not care about the individual lead observations along the track? The sentence should be corrected.**

**Response:** We apologize for the confusion. To avoid misunderstanding, we have revised 'this model can precisely determine the instantaneous elevation of lead' to 'After removing the MSS, we are able to precisely determine the instantaneous elevation of lead' (line 158-159).

**Line 190. You already have a lot of tables. Instead of providing a table with the exact densities for each month, instead you could consider just writing the equation in the section.**

**Response:** Thank you for the valuable comment. We have replaced table 2 with equation (1) (line 178).

*Methodology*

**Section 3.1: Figure 3 shows a removal of 'outliers' using ±1 m, but this is not mentioned in the text.**

**Response:** Thank you for the valuable comments. We have revised 'The relative surface elevation, $h_r$, was obtained after eliminating the residuals, as shown in Fig. 4(a) and (b) 'to 'In addition, the relative surface elevations, $h_r$, outside the range $+1.0$ m to $-1.0$ m are removed from processing, as shown in Fig. 3 (a) and (b) '(line 200-201).

**Line 198. What is meant by "invalid values". How are they considered invalid?**

**Response:** 'Invalid values' is 'nan'. The original data may contain nan for each parameter, such as range, dry tropospheric delay correction and wet tropospheric delay correction. After subtraction, we removed the nan results (line 187).

**Line 198-201. Which exact corrections (providers) are you using? This should be stated.**

**Response:** Thank you for your valuable comments. We have revised 'it was necessary to consider the dry and wet tropospheric delay correction, inverse barometric correction, ionospheric correction, ocean tidal correction, equilibrium tidal correction, ocean load tidal correction, earth tidal correction and polar tidal correction when calculating the surface elevation' to 'it was necessary to consider the dry and wet tropospheric delay correction (National Centers for Environmental Prediction, NCEP), inverse barometric correction (NCEP), ionospheric correction (National Satellite Ocean Application Service, NSOAS), ocean tidal correction (Goddard Space Flight Center, GSFC, GOT4.10c), ocean load tidal correction (GSFC, GOT4.10c), earth tidal correction (Cartwright et al., 1973) and polar tidal correction (Wahr, 1985) when calculating the surface elevation' (line 188-192).

**Line 207-213. The purpose of removing the residuals should be stated. This is not commonly applied for the lead/floe retrieval methodologies, and as such can add to the reasons why the results differ.**

**Response:** According to your valuable comments, we supply the reason of removing the residuals. 'The residual error may be caused by the error of precise orbit determination and different tracking algorithm' (line 198-199).

**Line 215. Why are you assuming the SSHA to be 0? And what is the impact of this? Figure 4+5 show that SSHA is not necessarily zero (zero-elevation is only relative to MSS). Instead, could you consider nan (not-a-number) the data and just interpolate where data is not available?**

**Response:** Thank you for the valuable comments. We tried to set it as nan value and interpolated it along track. We found that the correlation between the retrieved sea ice

freeboard and OIB sea ice freeboard was 0.77, RMSE was 0.13 m, and MAE was 0.12 m. Therefore, we updated our method and re-found the optimal threshold of 15 points within 25 km segment, as shown in table 7 and 8 (in the manuscript). If there are more than 15 points within 25 km section, we take the average of the 15 lowest values as the SSHA, otherwise set as nan, and interpolate the SSHA along track.

If the points in the 25 km segment are less than 15, we assume the SSHA to be 0, which is maybe higher than the average of lowest points (negative). The HY-2B radar freeboard retrieved by the assumption of SSHA to be 0 is lower than the radar freeboard retrieved by the interpolation of SSHA, as shown in Figure 10.

[Figure]

Interpolation minus Assumption

Figure 10. The difference between the radar freeboard retrieved by assumption (If there are more than 9 observation points per 25-km segment in every track, the average of the 9 lowest values is taken as the SSHA. Otherwise, the SSHA is set to 0.) and the radar freeboard retrieved by interpolation (If there are more than 9 observation points per 25-km segment in every track, the average of the 9 lowest values is taken as the SSHA. Otherwise, the SSHA is set to nan and nearest interpolation is performed along track.) in April 2020.

As the points in the 25-km segment are less than 15, we attempted to interpolate SSHAs that are considered as nan. We used the AWI snow depth to obtain IS-2 sea ice freeboard and HY-2B and CS-2 sea ice freeboards. Figure. 11 shows monthly comparisons of sea ice freeboard between HY-2B retrieved by interpolation and IS-2 and between CS-2 and IS-2, respectively. The RMSE and MAE of interpolation are lower than the assumption. Therefore, we adopt the interpolation of SSHA to retrieve Arctic radar freeboard.

[Figure]

Figure 11. Monthly comparisons between HY-2B sea ice freeboard retrieved by interpolation and ICESat-2 sea ice freeboard and between CryoSat-2 sea ice freeboard and ICESat-2 sea ice freeboard: panels (a) show comparisons between HY-2B and ICESat-2 from October 2019 to April 2020 and from October 2020 to April 2021, and panels (b) shows comparisons between CS-2 and ICESat-2 from October 2019 to April 2020 and from October 2020 to April 2021.

**Line 226. Isn't the 0.22 value depending on snow density and thus varying, even in the AWI product? It is based on slower wave propagation, given by $(\dfrac{c}{c_s}-1)$ , where $c$ is the speed of light and $c_s$ is the speed of light through snow. Here, where depends on the snow density (which you vary when using Mallet et al. (2020)). If so, this should be corrected. If not, please explain exactly what this value is and where it comes from.**

**Response:** Thank you for the valuable comments. We have revised ' $f = f_r + 0.22 \cdot h_s$ ' to

' $f = f_r + (\dfrac{c}{c_S}-1) \cdot h_S$ ' and ' $c_S = c \cdot (1+5.1 \cdot 10^{-4} \rho_S)^{-1.5}$ ' (line 201 and 204).

**Line 229. How is the gridding performed?**

**Response:** We have revised 'We gridded the sea ice thickness to the 25-km polar stereographic grid with a temporal resolution of 1 month' to 'To obtain monthly grid values, we averaged all thickness measurements within a 25 km radius of the center of each grid cell, with all points receiving equal weighting' (line 257-258).

*Results*

**Line 237-249: "Fig 5(a) (....)." This entire paragraph could benefit from some rephrasing, as it was confusing and long to read. Also, check the manuscript guidelines for The Cryosphere again to ensure that the way you write the different expressions ("0±0.31m") follows the guidelines.**

**Response:** Thank you for the valuable comments. We have revised '0±0.31 m/0±0.25 m and 0.081±0.17 m/0.087±0.25 m' to '**0 m/0 m and 0.081 m/0.087 m**' (line 240). Also, we have revised '-0.26±0.13 m/-0.20±0.18 m and -0.051±0.029 m/-0.069±0.066 m' to '**-0.21 m/-0.11 m and -0.051 m/-0.069 m**' (line 242-243).

**Line 264-265. I think you need to consider what you mean by 'smallest' and 'largest' deviation. Sure, in actual numbers, -2.4 cm is smallest, and 1.9 cm is largest. But the deviation from zero must mean that the -2.4 cm is largest! So, this needs to be rephrased for clarity.**

**Response:** Thank you for your valuable comment. We have revised the description based on new statistics values (line 257-289).

**Line 336 -358. You mention the Kwok snow depth product is used, but in Figure 3 you mention that the AWI snow depth is used for the processing of the radar freeboards. Then, later you mention that the Kwok snow depth is used for the ICESat-2 freeboards to obtain the sea ice freeboard and for the CS2 to obtain sea ice freeboard. How come you decide to use this snow depth product suddenly, and not just use the AWI snow depth throughout?**

**Response:** We apologize for the confusion. We have changed to use AWI snow depth to obtain sea ice freeboard and sea ice thickness. We have removed the description of Kwok snow depth in the manuscript.

**Line 274-275. The fact that HY-2B modal freeboard is thicker than the mean is not expected – we usually expect a skewed distribution with an abundance of thinner freeboards compared to the thicker freeboards. This should be discussed further.**

**Response:** Thank you for your valuable comment. Before calculating the modals of radar freeboard, we obtain the HY-2B and CS-2 radar freeboard and sea ice thickness with only three decimal places. If it is close to the normal distribution, the mode maybe close to the mean, or even slightly greater than the mean (line 305-307).

**Line 337-338: Be careful when discussing spatial resolution of IS2. The footprint of IS2 is approximately 11 m, yes, but the ATL10/ATL20 products are processed data that is not of this spatial resolution.**

**Response:** Thank you for a kind reminder. We used the ATL 20 products to validate the HY-2B and CS-2 sea ice thickness products. The ATL 20 products are monthly grid products, and the spatial resolution is 25 km. We would be careful to discuss the spatial resolution of IS-2.

**Line 358. You mention sea ice thickness error? What is meant by this – this has not been discussed previously.**

**Response:** Thank you for the valuable comments. The sea ice thickness error is meant that RMSE and MAE between the HY-2B sea ice thickness and CS-2 sea ice thickness. We have revised "The sea ice thickness error was thus related not only to the sea ice freeboard and snow depth values but also to the sea ice type and snow density" to "The RMSE and MAE of sea ice thickness are thus related not only to sea ice freeboard and snow depth but also to sea ice type and snow density" (Line 353-354).

**Figure 13. For almost all subfigures, we observe HY-2B to generate significantly thicker sea ice thickness than IS2 (CS2 also does not observe this behavior). This seem to showcase that some overestimation of thickness from HY-2B, which I think could be discussed more in the text for the IS2 comparison.**

**Response:** Thank you for the valuable comments. We think that the overestimation of thickness from HY-2B may be caused by the error of precise orbit determination, Brown tracking algorithm and the algorithm of SSHA determination. In addition, the differences of measurement mode and footprint size maybe result the discrepancies between HY-2B and IS-2. We supplied the discussion in the text for the IS-2 comparison (line 346-348).

*Discussion*

**Line 371-374. Long and confusing sentence, however it sounds like it is saying that in general HY-2B is observing smaller freeboards than the AWI freeboards – but, when comparing otherwise you said that HY-2B was higher than CS2 freeboard (I assume this is the ESA observations then). I suggest you rephrase this sentence for clarity, and then discuss further the fact that you are comparing observations with different re-trackers – what can you really get from this? And are you comparing along-track or gridded? This should also be mentioned.**

**Response:** Thank you for your valuable comments. We have revised to 'The SSHA values of Scheme 8 are largest, both are greater than -0.1 m. The mean deviations of gridded radar freeboard between HY-2B and CS-2 are all less than 0, indicating that the HY-2B radar freeboard retrievals are generally lower than the AWI CS-2 radar freeboards.

In addition, the MAE of Scheme 8 is larger than that obtained under Scheme 7' (line 367-370).

**Line 386. If you are following Mallet et al. (2020), the 0.22 value should be changing with density.**

**Response:** Thank you for your valuable comments. We have revised ' $f = f_r + 0.22 \cdot h_s$ '

to ' $f = f_r + (\dfrac{c}{c_S} - 1) \cdot h_S$ ' and ' $c_S = c \cdot (1 + 5.1 \cdot 10^{-4} \rho_S)^{-1.5}$ ' (line 221 and 224).

**Line 393, Equation 8. I am intrigued by why you are including the impact of +0.22 ρ w, when e.g., the study of Ricker et al. 2014 did not. This may explain my comment further down regarding line 415-425. I suggest you check up on this to make sure that this is the correct partial derivative.**

**Response:** Thank you for the valuable comments. We have revised the gridding uncertainty budget for radar freeboard, sea ice freeboard and sea ice thickness in equation (7) to (13) (in the manuscript).

**Line 404. The uncertainty of sea ice densities should be cited.**

**Response:** Thank you for the valuable comments. We added the cite of Alexandrov et al. (2010) (line 406).

**Line 415-425. Something seems odd in the uncertainty determination. Ricker et al. (2014) determined freeboard uncertainties of 6 cm (12 cm) for FYI (MYI), but for sea ice thickness, they estimate 60 cm (120 cm) (increase by factor 10). This is not observed in your estimation, where the CS2 sea ice thickness uncertainty is only 12.1-15.4 cm. Also, your freeboard uncertainties to thickness uncertainties do not increase by factor 10 for HY-2B observations? Something seems off here. I suggest going through this again and ensuring that the units are correct.**

**Response:** Thank you for the valuable comments. We have revised the gridding uncertainty budget for radar freeboard, sea ice freeboard and sea ice thickness in equation (7) to (13).

*Conclusion*

**Line 442-44: I am not convinced by this methodology, when the comparisons and correlations with OIB are so low compared with CS2. A correlation of 0.58 for HY-2B but 0.84 for CS2 for freeboards, and 0.41 for HY-2B but 0.80 for CS2 on thickness is**

**quite a big difference – and for me just shows that the method used here is not that applicable for sea ice, and not reliable.**

**Response:** Thank you for the valuable comments. We are sorry for the results that are not reliable. We use the PP values to extract leads, and use them to obtain the SSHA using nearest interpolation. Finally, the result is not satisfactory. In the meanwhile, we also use the MODIC lead product to extract lead, but we don't retrieve the satisfactory result. The reason is that the valid measurement points are sparse, resulting the error of interpolation is larger than the error the average of lowest points within 25 km section. According to your valuable comments, we try to set it as nan value and interpolate it along track. We find that the correlation between the retrieved sea ice freeboard and OIB sea ice freeboard is 0.77, with a RMSE is 0.13 m, and a MAE is 0.12 m.

**Line 460-468: A lot of new information provided here which is not discussed or even commented upon in the text. I strongly encourage the authors to include a discussion (in the discussion section) on the impact of re-tracking methodology applied to the HY-2B freeboards (and how it differs from the CS2 re-tracking). There are also some considerations for what should be done in future work, which are not discussed in the text but only here. Finally, the authors conclude that they have used a sub-optimal re-tracking algorithm which is only applicable for ocean (not sea ice?). Makes me question why they did this analysis with this algorithm, when it is not suitable, and the results clearly show this too.**

**Response:** Thank you for the valuable comments. The SGDR products have been re-tracked for Brown modal. We are sorry for that we don't have the retrack values of Brown modal. We couldn't compare the retrack values of Brown modal with the CS-2. The SMLE algorithm is mainly aimed at ocean surface. The OCOG algorithm is mainly aimed at land, sea ice and ice sheet, but the OCOG data isn't contained in the SGDR product. These two re-tracking algorithms have some overlapping measurement points. We tried to explore the potential application of the SGDR product in this manuscript. We plan to reprocess the L1 data to comprehensively explore the potential of the HY-2B.

**Technical corrections**

**Line 113: "main" – not sure they are the only main products? The CCI product could be mentioned here. Furthermore, the upcoming releases of CryoTEMPO (http://cryosat.mssl.ucl.ac.uk/tempo/downloads.html) are expected to be a favorable product to be used in the future by the science community.**

**Response:** According to your valuable comment, we supplied the introductions of the CCI product and the CryoTEMPO (line 111-112).

**Line 116. What is meant by L2I? The intermediate product is not provided to the public, but rather the L2 product is.**

**Response:** According to your valuable comment, we have revised 'L2I' to 'L2' (line 113).

**Line 123. "were by " -> "were provided by".**

**Response:** According to your valuable comment, we have revised 'were by' to 'were provided by' (line 120).

**Line 148. "Denmark Technical University (DTU)" -> "the Technical University of Denmark (DTU)".**

**Response:** According to your valuable comment, we have revised 'Denmark Technical University' to 'the Technical University of Denmark' (line 142).

**Line 171. "W99" – "Warren et al. 1999 (W99)": reference the publication and explain the acronym.**

**Response:** According to your valuable comment, we have revised 'W99' to 'Warren et al. 1999 (W99)' (line 164).

**Line 171. "AMSR-2" -> "Advanced Microwave Scanning Radiometer-2 (AMSR-2)".**

**Response:** Thank you for the valuable comment. We had added the 'Advanced Microwave Scanning Radiometer-2 (AMSR-2)' (line153).

**Line 202. "were interpolated" -> "were linearly interpolated" (or however it was interpolated).**

**Response:** Thank you for the valuable comment. We have revised 'were interpolated using the SIC data' to 'were obtained using nearest interpolation' (line 193).

**Line 204. "was not used" -> "was not considered".**

**Response:** Thank you for the valuable comment. We have revised 'was not used' to 'was not considered' (line 195).

**Line 206-207. "from the obtained surface elevations of ground objects (…)": consider rephrasing this sentence, as it is not clear.**

**Response:** Thank you for the valuable comment. We have revised 'The geoid fluctuations were eliminated by subtracting the MSS height from the obtained surface elevations of ground objects' to 'The mean sea-surface (MSS) height product DTU18 (Andersen et al. 2018a; Andersen et al. 2018b) is subtracted from the geolocated surface elevations to remove the geoid fluctuations' (line 196-197).

**Line 215-216. "(…) were subtracted from the SSHA to obtain": The SSHA are subtracted from the segment height, not the other way around. This sentence should be rephrased.**

**Response:** Thank you for the valuable comment. We have revised 'All the observed values $h_r$ inside each 25-km segment were subtracted from the SSHA to obtain the radar freeboard height' to 'The SSHA was subtracted from the observed values $h_r$ inside each 25-km segment to obtain the radar freeboard height' (line 208).

**Line 231: " ρ $_{seawater}$" -> " ρ $_w$"   (following the equation).**

**Response:** Thank you for the valuable comment. We have revised ' ρ $_{seawater}$' to ' ρ $_w$' (line 230).

**Line 288-289. "Because the (…)": I wouldn't say that the lower SSHA necessitates higher freeboards, because they are only higher if the elevations of the floes are of the same height as CS2 or higher. If the floe elevations were also lower, HY-2B would not necessarily have higher freeboards. Consider rephrasing it.**

**Response:** Thank you for the valuable comment. We have revised 'Because the HY-2B SSHAs were lower than those of CS-2, the HY-2B radar freeboards were higher than those of CS-2' to 'It should be noted that the HY-2B SSHAs are lower than those of the CS-2, the elevations of the floes are slightly higher than the CS-2, and the HY-2B radar freeboards are higher than those of the CS-2' (line 288-289).

**Line 451-452. You mention sea ice error again, but you have merely referenced this by the paper of Ricker et al. (2014) and not done more sensitivity studies on this. I suggest removing this sentence or discuss this more in the text if you aim to keep it.**

**Response:** Thank you for the valuable comment. We have removed the sentence.

---

## Author Comment (AC2)

Dear Editors and Reviewers,

Thank you for considering our manuscript, and the reviewers' comments concerning our manuscript entitled Assessment of Arctic Sea Ice Thickness Retrieval Ability of the Chinese HY-2B Radar Altimeter (manuscript ID: egusphere-2022-870). The comments are all valuable and very helpful for improving upon our paper. We have now carefully reviewed and addressed all of comments which we hope meet with approval, with revisions to the original manuscript shown in red. The primary corrections in the paper and the responds to the reviewer's comments are as flowing:

**Responses to Reviewer's Comments:**

**Reviewer #2:**

**General comments**

**GC0: The HY-2B satellite is carrying a dual frequency Ku- and C- band radar altimeter. The data are used for deriving the sea ice thickness and it is compared to similar retrievals from CryoSat2 and airborne campaign data (Operation Ice Bridge). This is an urgent and welcome topic. However, important details are missing from the MS about the instrument and data and discussion of the processing steps and choices made. I did not see it stated clearly, but this study is not using the dual-frequency capability of HY-2B when deriving the freeboard? Only Ku-band? I think that the answers to these questions are important for evaluating the MS and the novelty of it.**

**Response:** Thank you for the valuable comments. We have supplied the descriptions of instrument, data, and processing steps and choices made (line 87-93, 96-98, 196-197, 200-201, 209-210).

In this manuscript, we didn't use C-band of HY-2B to retrieve radar freeboard, just only Ku-band. **We have revised the title to 'Assessment of Arctic Sea Ice Thickness Retrieval Ability of the Chinese HY-2B Ku-band Radar Altimeter'.** Currently, there is no research about using C-band to retrieve radar freeboard. We are searching the possibility of C-band to retrieve radar freeboard based on HY-2B L1 data. We will describe in detail in subsequent articles.

**GC1: The snow data used in the processing are the same as the AWI snow depth (line 224). This is logical when HY-2B data are compared to AWI CS2 data. Are all the other processing steps and auxiliary data used in the processing identical to the AWI processing so that a proper comparison can be made? In line 24 it is mentioned that some of the differences were due to the applied sea surface height anomaly extraction method. Is it due to differences in the methodology or sensor specific differences? Please clarify.**

**Response:** Thank you for the valuable comments. The sea ice concentration, sea ice type data used in the processing are identical to the AWI processing. But the MSS model used is the DTU 18, which is not identical to the DTU 15 of the AWI. We used DTU 15 to calculate the radar freeboard and compared with the radar freeboard calculated by DTU 18. We found that the difference between the two was small and the impact on the results was negligible, so we adopted DTU 18 to retrieve radar freeboard. The other processing steps are different with the AWI, including the retrack algorithm, the extraction of SSHA, and the criterion of filtering. These different processing steps can result in the discrepancies between both HY-2B and CS-2. We think it is due to differences in the methodology, sensor specific differences and measurement mode. We have supplied the reason of the differences in the section of result (line 263-264, 267-268, 297-300, 346-348). We will improve the processing method to obtain radar freeboard with higher precision using HY-2B L1 data.

**GC2: A number of studies focused on Ku- and Ka-band radar altimeter applications from AltiKa and CryoSat and for preparation of the planned CRISTAL mission. This MS does not continue that discussion using C- and Ku-band radar altimetry for sea ice applications. Anyway, I think that some justification of the choices made in this paper are needed, for example, it is implicitly understood that radar scattering is at the snow-ice interface. Is that a good assumption? And how would that differ for Ku- and C-band. Would the different foot-print sizes at the two frequencies have an impact for the derived ice thickness?**

**Response:** Thank you for the valuable comments. We assumed that the radar pulses penetrate through any snow cover on ice floes and scatter from the snow-ice interface, which has been shown in laboratory experiments where the snow cover on sea ice is cold and dry (Beaven et al., 1995; Tilling et al., 2018). Despite some evidence that the scattering horizon migrates as temperature rises (Willatt et al., 2010), Tilling et al. (2017) did not observe any bias in their thickness retrieval when compared to year-round ice draft data, and so they thought that the impact of this effect was not significant (line 215-220). The differences between C band and Ku band are mainly reflected in footprint size, waveform (retrack algorithm) and radar penetration factor. Two different frequencies maybe have an impact for the radar penetration factor, which will impact the radar freeboard. The different footprint sizes at the two frequencies also have an impact for the spatial resolution of radar freeboard. The spatial aliasing situation has long existed in sea ice thickness (Geiger et al., 2015), where the different instrument footprint sizes bring about artificial modes in thickness distribution (Zhou et al., 2020) .

The HY-2B altimeter uses two different tracking modes: suboptimal maximum likelihood estimation (SMLE) and offset centre of gravity (OCOG). The two tacking modes can exchange according to the observation surfaces. The HY-2B Level-2 altimetry products (SGDR products) we used do not have OCOG data. The measurement data point is sparse.

We will explore the possibility of C-band to retrieve radar freeboard using L1 data. Because the L1 data can contribute to assess the comprehensive retrieval ability of C-band without missing OCOG data.

**GC3: The snow depth dataset is a strange combination of retrieved snow depth over first-year ice and snow depth climatology over multiyear ice and some smoothing and filtering. What does the different snow depths and ice densities in first-year ice and multiyear ice areas mean for the retrieved ice thickness? How does this compare to the variation in derived freeboards for the ice thickness variability? What is the impact of the fixed ice densities and the ice type classification with its different snow depths etc.? Is the snow depth needed at all? To me it seems that the snow data are massaged until the "right" ice thickness is achieved.**

**Response:** Thank you for the valuable comments. Based on the snow depth and density measured at Soviet drifting stations on multiyear Arctic sea ice from 1954 to 1991, a two-dimensional quadratic model named W99 Climatology Snow Depth model is fitted to represent the geographical and seasonal variation in snow depth for a particular month, irrespective of the year (Warren et al., 1999). The climatology was based on observations from drift stations in a period where the Arctic Ocean was dominated by multiyear sea ice. It is therefore likely that the reduction of multiyear sea ice in the recent decade (Nghiem et al., 2007) may have impacted the distribution of snow depth in areas that are now more often covered by seasonal sea ice (Ricker et al., 2014). Rostosky et al., (2018) analyzed the correlation between the gradient ratio of different channels and OIB snow depth data and tried to expand the snow depth retrieval onto MYI. Since the OIB flight is conducted in March and April of every year in the Arctic, the retrieved snow depth on the MYI of the algorithm is only applicable to March and April of spring. Therefore, Hendricks et al. (2020) introduced a monthly snow depth and density parametrization based on merging of the W99 snow climatology and daily snow depth over first-year sea ice from AMSR2 data provided by the Institute for Environmental Physics of the University Bremen (IUP).

Dong et al. (2022) found that the spatial patterns in six products (W99, AWI, Bremen, Kwok, Neural Network and LSTM) are in broad agreement; that is, snow cover is thicker over the sea ice of northern Greenland and the northern Canadian Archipelago, while snow cover is thinner over the sea ice of the Eurasian marginal Sea and Baffin Bay region. The spatial pattern of snow depth is similar with sea ice thickness.

The radar freeboard ($f$) can be converted into sea-ice thickness $T$ depending on the snow depth ($h_S$) and the densities of snow ($\rho_S$), sea ice ($\rho_i$) and sea water ($\rho_W$), as shown in Eq. (5).

$$T = \frac{\rho_w}{\rho_w - \rho_i} \cdot f + \frac{\rho_s}{\rho_w - \rho_i} \cdot h_s \tag{5}$$

where $T$ is the sea ice thickness, $\rho_w$ is the seawater density, $\rho_s$ is the snow density, $\rho_i$ is the sea ice density and $h_s$ is the snow depth. Consistent with the approach of Laxon et al. (2013) we use ice densities ($\rho_i$) of 916.7 kg m$^{-3}$ for first-year ice (FYI) and 882.0 kg m$^{-3}$ for MYI (Alexandrov et al., 2010). Furthermore, we assume a value of 1024 kg m$^{-3}$ for the water density ($\rho_w$) (Ricker et al., 2014).

**Specific comments:**

**Line 13: "… it is of great significance…" this is a subjective statement without real justification. Please reformulate.**

**Response:** Thank you for the valuable comment. We have revised to 'it is urgent to explore the potential application of this dataset in Arctic sea ice thickness retrievals' (line 8-9).

**Line 14: delete "values".**

**Response:** According to your valuable comment, we have removed the vocabulary of 'values' (line 10).

**Line 15: "… ice growing cycles" replace by "winters".**

**Response:** Thank you for the valuable comment. We have revised 'sea ice growing cycles' to 'cycles', as we distinguish it to winter and spring (line 10).

**Line 17: delete "recorded".**

**Response:** According to your valuable comment, we have removed the vocabulary of 'recorded' (line 17).

**Line 18: replace "verified" with "compared"..**

**Response:** Thank you for the valuable comment. We have removed the sentence.

**Line 40: delete "effect".**

**Response:** According to your valuable comment, we have removed the vocabulary of 'effect' (line 30).

**Line 45: It is confusing and inaccurate what is written about AMOC. Please reformulate or delete.**

**Response:** According to your valuable comment, we have removed the sentence.

**Line 52: add "and extent" after "density".**

**Response:** According to your valuable comment, we have revised 'density' to 'extent' (line 40).

**Line 57: use "derive" instead of "estimate", also line 61.**

**Response:** According to your valuable comment, we have revised 'estimated' to 'derived' (line 45 and 47).

**Line 75: "few reports" please list these few reports as references.**

**Response:** According to your valuable comment, we have supplied the references, 'Jiang et al. (2022) preliminarily estimated the Arctic radar freeboard from October 2020 to April 2021, compared them with radar freeboard products from the Alfred Wegener Institute (AWI). The overall difference between the HY-2B radar freeboard estimates and the AWI data is $0.088 \pm 0.057$ m. The radar freeboards are generally higher for HY-2B than CS‐2' (line 66-69).

**Line 77: delete "as a supplementary means"**

**Response:** According to your valuable comment, we have removed the 'as a supplementary means' (line 70).

**Line 79: delete sentence starting with "Therefore…".**

**Response:** According to your valuable comment, we have removed the sentence (line 70).

**Line 90: Add some details about the HY-2B altimeter.**

**Response:** According to your valuable comment, we have added some details about the HY-2B altimeter. 'The HY-2 radar altimeter adopt the same reference ellipsoid of the TOPEX/Poseidon and the Jason-1/2/3. The HY-2B radar altimeter is a dual-band pulse-limited radar altimeter that comprised of the Ku band and C band to remove the impacts of ionospheric delays. The HY-2B satellite adopts an orbit with a repeat cycle of 14 days in the early stage, and an orbit with a repeat cycle of 168 days in the late stage. Waveforms have been sampled to 128 range bins, each of which is 0.4864 m in range' (line 87-92). 'The SGDR products contain waveform data and have been re-tracked using the Brown model. In addition, the HY-2B altimeter uses two different tracking modes:

suboptimal maximum likelihood estimation (SMLE) and offset center of gravity (OCOG). The two tacking modes can exchange according to the observation surfaces' (line 96-98).

**Line 91: delete "successfully".**

**Response:** According to your valuable comment, we have removed the vocabulary of 'successfully' (line 102).

**Line 168: "Snow depth" it is unclear which snow depth dataset is actually used. Please**

**clarify this and explain if you are using AWI snow depths or IS-2/CS2 snow depths.**

**Response:** According to your valuable comment, we have used AWI snow depth to retrieve radar freeboard.

**Line 224: delete "depth".**

**Response:** According to your valuable comment, we have revised to 'snowpack' (line 214).

**Line 226: Eq. 3 units in meters? Please add units, sometimes [m] sometimes [cm]⋯ use**

**the same units throughout.**

**Response:** According to your valuable comment, we have added the units in meter (line 204, 205, 212 and 213).

**Line 230: Eq. 4, what is the sea water density? Sometimes units are [kg m$^{-3}$] sometimes**

**[kg/m$^3$], stick to one form. Multiplication is sometimes a dot and sometimes x?**

**Response:** According to your valuable comment, we stick to use unit [kg m$^{-3}$] and a dot in multiplication (Eq. (4), Eq. (5), Eq. (6), Eq. (8), Eq. (10) and Eq. (13)). We have revised 'sea water density' to 'water density' (line 230).

**Line 322: I think that I know, but this is confusing: what is a "deviation"? standard**

**deviation? or bias? Throughout the MS.**

**Response:** According to your valuable comment, we have revised to 'In general, the HY-2B sea ice thicknesses exhibit a MAE of approximately 0.2 m with respect to CS-2' (line 315).

**Line 324: replace "independent" with "OIB".**

**Response:** According to your valuable comment, we have revised 'independent' to 'OIB and IS-2' (line 323).

---

## Author Comment (AC3)

Dear Editors and Reviewers,

Thank you for considering our manuscript, and the reviewers' comments concerning our manuscript entitled Assessment of Arctic Sea Ice Thickness Retrieval Ability of the Chinese HY-2B Radar Altimeter (manuscript ID: egusphere-2022-870). The comments are all valuable and very helpful for improving upon our paper. We have now carefully reviewed and addressed all of comments which we hope meet with approval, with revisions to the original manuscript shown in red. The primary corrections in the paper and the responds to the reviewer's comments are as flowing:

**Responses to Reviewer's Comments:**

**Reviewer #3:**

**General comments**

**This manuscript introduces the retrieval of Arctic sea ice freeboard and thickness with the radar altimeter onboard HY-2B. The inclination angle of HY-2B allows the coverage of up to 82° N, and the satellite potentially constitutes an important source of information for sea ice of both polar regions. Specifically in the manuscript, the authors focus on the processing of converting existing range (or elevation) product of HY-2B into radar/ice freeboard and thickness. I consider the data from HY2B and this submission a good contribution to the community, but I do have the following comments that in my opinion that should be addressed first.**

**GC0: The overall uncertainty analysis needs to be more clear and precise, especially in terms of the determination of freeboard uncertainty. First, the (claimed) uncertainty used for further analysis is $\sigma_{\text{SGDR}}$, which is only 2cm and should be a lower bound of the actual range uncertainty. This uncertainty, in its true quantity, applies to both SSHA and individual freeboard. Since SGDR is the upstream dataset this work relies on, it is not necessary a quantity that needs much elaboration within this work. However, its value needs to be justified. For example, what is the gate resolution of HY-2B? Is the 2cm uncertainty due to the combination of several footprints (therefore smaller)? Furthermore, the authors state that Ricker et al. (2014) `believed' the random uncertainty of radar freeboard to be determined by radar speckle noise. Is it possible to provide any other proof of how is the reference relevance to the treatment here?**

**Response:** Thank you for the valuable comments. $c$ is the speed of light in vacuum, $c$ = 299792485.0 m/s, $B$ is the measured chirp bandwidth, $B$ = 320,000,000 Hz. Each waveform sample covers a range of $\Delta R = c/2B$ = 0.4684 m or a delay of $\Delta t = 1/B$ = 3.125 ns. The instrument system error of HY-2B is 2 cm. It maybe caused by orbit error and the footprint resolution. According to Wingham et al. (2006) and CryoSat-2 Product Handbook Baseline D 1.1, there are three operating modes, low resolution mode (LRM), synthetic aperture mode (SARM), and synthetic aperture interferometric mode (SARInM). LRM provides for conventional, pulse-limited altimetry using a single antenna. The radio

frequency (13.575 GHz), pulse bandwidth (320 MHz), samples in echo (128 in LRM) and range bin sample (0.4684 m for LRM) are same with HY-2B altimeter. In addition, the instrument system error of LRM is 0.07 m (Wingham et al., 2006). Therefore, we adopted relevant treatment to HY-2B. Acording to the instructions for HY-2B satellite data, we know the range error is 0.02 m (https://osdds.nsoas.org.cn/HY2B_introduce).

**GC1: Second, the uncertainty of SSH is computed as the standard deviation (SD) of SSH points (of the along-track 25km segment). I think this is a crude guess, and unfortunately, probably an underestimation of SSH-induced uncertainty. Because the along-track points that are not used for determining SSH share the same uncertainty caused by the same set of SSH points, and therefore the uncertainty of the retrieved freeboard samples on the same track is systematic (rather than random). Therefore, in Eq. (11) it should be averaged out and diminished much more slowly.**

**Response:** Thank you for the valuable comments. If the points in the 25 km segment are more than 15, the average of the 15 lowest values is taken as the SSHA. Otherwise, the SSHA is obtained by nearest interpolation. We change to use the standard deviation of SSHAs within a 25-km moving window as the uncertainty of SSHA ($\sigma_{SSA}$). The all SSHAs are used for determining radar freeboard. The sea ice thickness uncertainty can be divided into random uncertainty and systematic uncertainty. According to Ricker et al (2014) and Hendricks et al (2020), the uncertainty of SSHA is random uncertainty. The gridded uncertainty of parameters with only random uncertainty is computed as the error of the weighted mean. This approach applies only to the radar freeboard, whose two error contributions range noise and sea surface height interpolation uncertainty are both defined as random error contributions (Hendricks et al., 2020). Ricker et al. (2014) hypothesized that the uncertainties of the modified W99 snow depth and snow density resulting from interannual variabilities are systematic and cannot be regarded as random uncertainty. However, the AWI snow depth product is a composite snow depth product obtained by integrating the W99 climatology snow depths and the daily average AMSR-2 snow depths of Bremen University. Therefore, we assumed that the uncertainties in the AWI snow depth and snow density products are systematic uncertainty. In addition, the density of snow and sea ice are also treated as systematic errors. Due to the variability in seawater density, the contribution of its uncertainty is ignored (Kurtz et al., 2014; Ricker et al., 2014) (line 1281-1288). The grid uncertainty of radar freeboard, sea ice freeboard and sea ice thickness can be expressed as shown in Eq. (1) to (4):

$$\hat{\sigma}_{l3,rf} = \sqrt{\frac{\sigma^2_{SSA} + \sigma^2_{SGDR}}{n}} \quad (1)$$

$$\sigma_{l3,f} = \sqrt{\left(\left(\frac{c}{c_S} - 1\right) \bullet \overline{\sigma}_{sd}\right)^2 + \left(\hat{\sigma}_{l3,rf}\right)^2} \quad (2)$$

$$c_S = c \bullet (1 + 5.1 \bullet 10^{-4} \rho_S)^{-1.5} \ (3)$$

$$\sigma_{I3,T} = \sqrt{(\frac{\overline{\rho}_w}{\overline{\rho}_w - \overline{\rho}_i}\sigma_{I3,f})^2 + (\frac{\overline{f} \bullet \overline{\rho}_w + \overline{sd} \bullet \overline{\rho}_i}{(\overline{\rho}_w - \overline{\rho}_i)^2}\overline{\sigma}_\rho^i)^2 + (\frac{\overline{\rho}_S}{\overline{\rho}_W - \overline{\rho}_i}\overline{\sigma}_{sd})^2 + (\frac{\overline{sd}}{\overline{\rho}_W - \overline{\rho}_i}\overline{\sigma}_\rho^s)^2} \ (4)$$

**GC2: Third, although the authors treat the snow-induced uncertainty as random error (possibly a typo on line 400), it is hardly the case, despite that the snow depth is based on both climatology and PMI retrieval. A simple counter argument is that: the PMI product is based on C-band PMI onboard AMSR-E/2, which is at about 60km in resolution. Not to mention the 8-grid smoothing that is carried out over the snow depth retrieval. Then there exists large local correlation of the snow depth uncertainty, given that 25km grid is used in this study. Arguably more importantly, the climatology of snow depth from W99 plays a very important role in the snow-depth composite of AWI, which is evident in the respective technical report. Given that W99 is halved for the combined product, its induced uncertainty is still very large and will dominate the portion caused by snow in the uncertainty of the final ice thickness product. Therefore, I will be very cautions to treat the snow depth related uncertainty as random errors.**

**Response:** Thank you for the valuable comments. Actually, we assumed that the uncertainties of AWI snow depth and snow density were systematic in the manuscript (line 1286-1287). We have recalculated the sea ice freeboard grid uncertainty and sea ice thickness uncertainty using averages.

**GC3: Finally, the density of snow and ice are also treated as random errors. This should be also a very 'optimistic' estimation, especially the effect on the final ice thickness uncertainty. Overall, I suggest that the authors reconsider the uncertainty quantification process and divide into the random part and the systematic part, and be cautious about which category each term belongs to. Differentiation of the uncertainty also ensures fairer comparison with the AWI product, and the author seemed to have used the random error of it, which is much lower in may part of the basin (e.g., Beaufort Sea).**

**Response:** Thank you for the valuable comments. The weighted mean ($\hat{\sigma}$) applies only to the radar freeboard, whose two error contributions range noise and sea surface height interpolation uncertainty are both defined as random error contributions. The error contribution of other physical variables to ice thickness is averaged ($\overline{\sigma}$). The grid uncertainty of radar freeboard, sea ice freeboard and sea ice thickness can be expressed as shown in Eq. (1) to (4).

**GC4: Another major issue I noticed is the limited data availability of HY-2B even in the later months of the winter. As shown in Figure 6 and 8, there are large areas with no valid radar freeboard on the monthly scale: on the Eurasia continental shelf,**

**Beaufort Sea, etc. What is the cause of the missing data? Invalid waveforms. I suppose that at this latitude HY-2B should have better coverage than CS2.**

**Response:** Thank you for the valuable comments. There are two main reasons for the missing data. Firstly, the HY-2B altimeter uses two different tracking modes: suboptimal maximum likelihood estimation (SMLE) and offset center of gravity (OCOG). The two tacking modes can exchange according to the observation surfaces. The HY-2B Level-2 altimetry products (SGDR products) we used do not have OCOG data. The measurement data point is sparse (line 96-98). Secondly, the SGDR data has nan values in the geophysical correction items such as the dry and wet tropospheric delay correction, inverse barometric correction, ionospheric correction, ocean tidal correction, ocean load tidal correction, earth tidal correction and polar tidal correction (line 187). We will try to use HY-2B L1 data to retrieve radar freeboard.

**GC5: Also, the authors mainly compared the HY-2B retrieval with CS2. However, Sentinel-3 and AltiKa have the same inclination angle as HY-2B, and especially Sentinel-3 satellites work on Ku-band (although they are of delay-Doppler type). Is it better to compare against Sentinel-3 retrievals? This is only a suggestion, out of my curiosity. The authors can decide whether this is an option or not.**

**Response:** Thank you for the valuable comments. We will conduct a comparative study between HY-2B and Sentinel-3 using HY-2B L1 data.

**GC6: The comparison against CS2-IS2 data contains inconsistencies in the methodology. The comparison of radar freeboard is inherently between two CS2 retrievals (one from AWI and the other one carried out by Kwok). Therefore, it's not a fair comparison, since the two products based on CS2 arises from the same source of information. So many issues are not present, such as limited representation by altimetry.**

**Response:** Thank you for the valuable comments. We use AWI snow depth to calculate HY-2B, CS-2 sea ice freeboard and sea ice thickness. We also used AWI snow depth to obtain IS-2 sea ice freeboard to make sure the final comparison is fair (line 339-340).

**Specific comments:**

**Some figures contain maps that are too small to read, such as Fig. 6 and 8. Pls increase the font and maps accordingly.**

**Response:** Thank you for the valuable comments. We have drawn Fig. 6 and 8 to make them as clear as possible (Fig. 5, 6,11 and 12).

**For the third-party and auxiliary datasets, I suggest that the authors just introduce them, without too much comments on the specific advantages. For each product there**

**are potentially uncertainties that are fully addressed. I think just stating the basic status quo of the products is already enough.**

**Response:** Thank you for the valuable comment. We have removed the description of Kwok snow depth product.

**The reference list is not strictly ordered.**

**Response:** Thank you for the valuable comments. We have checked the order of the references.

**The language usage needs improvements. General suggestions include avoiding long sentences, and avoiding complex statements. Several typos are present. I suggest that the authors give an overhaul of the manuscript after the major issues are addressed.**

**Response:** Thank you for the valuable comments. We have checked the manuscript, and we revised the manuscript using red fonts.

---

## Author Response (AR2)

Dear Editors and Reviewers,

The authors, above all, would like to thank the editor and the reviewers for their comments to help to improve the manuscript. We have now carefully reviewed and addressed all of comments which we hope meet with approval, with revisions to the manuscript shown in red. The primary corrections in the paper and the responds to the reviewer's comments are as flowing:

**Responses to Reviewer's Comments:**

**Reviewer #1:**

**One technical comment:**
**Line 629 in references change "Rasmus, T., John, L." to "Tonboe, R., J. Lavelle"**
**Response:** Thank you for the valuable comments. We have revised to 'Tonboe et al. (2016)' (Line 150, 154, 509 and 717-718). We have revised to 'Aaboe et al. (2021)'(Line 158, 509 and 524-526). We also have checked all referneces through the manuscript. We supplied the some references (Line 556-557, 625-627, 645-647, 649-651, 699-702, 715-716, and 734-736).

I'd like to thank the authors for the extensive edits, added work, and detailed response made to all the reviews. This is my second time reviewing this manuscript, so my comments will primarily focus on the edits made since the last review as well as the author's response to the reviews, but I will also include some additional comments that I realized when reading it a second time. Unfortunately, even with the extensive edits and comments provided by the authors, I still have concerns that should be considered before publication.

The authors have since the original submission changed the processing slightly to include not-a-number values when sea surface height anomalies (SSHA) were not available (instead of providing a value of 0), as well as used the 15 lowest (instead of 9) points within a 25-km segment as a measure of the sea level within leads (SSHA). This has improved their statistics when comparing with reference data (Operation IceBridge data) for both sea ice freeboard and sea ice thickness estimates. They have produced additional studies in response to reviews, where they have assessed waveform parameters/features to use for discriminating between leads/floes instead of their current methodology. However, this did not yield better results than their current methodology (and have therefore not been included in the manuscript). The authors have amended their uncertainty estimation, which now presents more feasible estimations, and provided a more extensive discussion on the limitations of their methodology and impacts hereof.

General comments

**I am happy to see the positive impact that the change in processing has shown in your comparison with OIB, however the statistics are still not a positive for HY-2B freeboard nor thickness estimates as the CryoSat-2 estimates. This, again, tells me that the processing chain of AWI (lead/floe discrimination, TFMRA50 re-tracker) is likely the way forward even with the different footprint and altimeter specifications of HY-2B compared with CryoSat-2. I do recommend the authors to investigate using waveform parameters for lead/floe discrimination as well as applying a re-tracker commonly used for sea ice (such as TFMRA50), however I understand that it will require quite the effort. If this is not possible, I think it is crucial that the manuscript clearly reflects (and mentions) the limitations of the methods and data used in this study. Below I have mentioned some suggestions for implementing this.**

**Response:** Thank you for the valuable comments. The processing chain of AWI (lead/floe discrimination, TFMRA50 re-tracker) provide a good reference for HY-2B L1 data. This will be the way of HY-2B forward.

**I think it is important to highlight that this study is a feasibility study. What I mean, is that you are in fact not truly assessing the retrieval ability of HY-2B for Arctic sea ice**

**thickness, since you are not using re-trackers commonly used, or designed, for sea ice–in fact, you use observations provided by an ocean-preferred re-tracker. As such, your results are also limited by this. I therefore propose you change the title of your manuscript to reflect this. Suggestions could be: "Feasibility of retrieving Arctic sea ice thickness from the Chinese HY-2B Ku-band radar altimeter" or "Assessment of Arctic sea ice thickness retrieval ability of the Chinese HY-2B Ku-band radar altimeter: a feasibility study". Also, I think it is crucial that you state the purpose of this study in the end of the introduction, highlighting that you are aiming to investigate whether HY-2B can be used for sea ice, but that you are limited to already provided higher-level products, and that it is not within the scope of the study to derive a freeboard product using your own re-tracker from the HY-2B product.**

**Response:** Thank you for the valuable comments. We have revised the title to "Feasibility of Retrieving Arctic Sea Ice Thickness from The Chinese HY-2B Ku-band Radar Altimeter". We have revised 'With the continuous development of China's Marine Dynamic Environment Satellite (Haiyang-2B, HY-2B), the HY-2B satellite can be used to observe polar sea ice' to 'With the continuous development of China's Marine Dynamic Environment Satellite, the feasibility of using the HY-2B satellite to map the polar sea ice must be explored.' (Line 72)

We have also supplied the limitation of this study. 'It is important to note in this study, however, that we are aiming to investigate whether HY-2B can be used for sea ice, but that we are limited to already provided higher-level (SGDR) product, and that it is not within the scope of the study to derive freeboard product using own re-tracker from the HY-2B SGDR product.' (line 72-75).

We have also revised 'In this study, we preliminarily tried to use HY-2B radar altimeter to retrieve reliable Arctic sea ice thicknesses. However, the shortcoming of this work is that we did not accurately distinguish between floes and lead. We did not re-track the SGDR products since they have been re-tracked using the Brown model.' to 'However, we are aiming to investigate whether HY-2B can be used for sea ice, but that we are limited to already provided higher-level (SGDR) product, and that it is not within the scope of the study to derive freeboard product using own re-tracker from the HY-2B SGDR product. The deficiency of this work is that we did not accurately distinguish between floes and lead.' (line 484-486).

**I'm happy to see a study using the waveforms and classifying the waveforms using pulse peakiness (PP), however slightly confused to see that it was in fact not satisfactory. Based on your response, I see some aspects that could potentially explain this: (1) PP can be calculated in different ways (that is to say, sometimes they are tweaked in different studies). I cannot see how you have calculated PP, but I do find it exceptionally odd that HY-2B is not able to use it to classify–whereas all other radar altimeters have in fact successfully used this. Therefore, it may be due to how you have calculated PP; (2) For HY-2B we do not yet know which PP thresholds to use. You have tried several PP thresholds (not sure what the selection of thresholds were based on) but choosing high PP's (which would normally yield a classification as lead)**

**resulted in almost no lead observations. This might be a result of the already low data coverage, which you also suggest, but I do propose that for the next time you look at this, that instead of pre-defined PP's, you choose a selection of waveforms and make a statistical analysis to derive the PP thresholds. I am aware that you state this is work for future studies, however it will unfortunately be those studies that have the true value then.**

**Response:** Thank you for the valuable comments. We used the equation (1) to calculate the PP value.

$$PP = \frac{\max(WF_i)}{\sum\limits_{i=1}^{128} WF_i} \times 128$$

(1)

where $WF_i$ represents the echo power at range bin index $i$.

We tried to use low, middle and high PP values (3,7,10 and 15) to extract lead points, but choosing high PP's (which would normally yield a classification as lead) resulted in almost no lead observations, choosing low PP's resulted in sparse floe observations (this might be a result of the already low data coverage). So we chose to use PP value greater than 10 to extract lead points. We will choose a selection of waveforms and make a statistical analysis to derive the PP thresholds in the future studies.

**There is very little data available by HY-2B. Why is that? It is based on the selected re-tracker where some restrictions/flags are applied? There must be applied some post-processing steps that remove the data. Could you provide paragraph explaining when in the processing this data is removed, and why it was removed? Surely, this should not be the case when a more appropriate re-tracker is used. Could you also provide a measure of how much coverage HY-2B has compared with CryoSat-2? (e.g., how many points within each grid cell when gridding).**

**Response:** Thank you for the valuable comments. The HY-2B altimeter uses two different tracking modes: suboptimal maximum likelihood estimation (SMLE) and offset center of gravity (OCOG). The two tacking modes can exchange according to the observation surfaces. For areas with slower changes in terrain height, such as the ocean and large areas of flat sea ice, the SMLE tracking mode is used. For areas with more dramatic changes in topographic height, such as land and sea ice areas, the OCOG tracking mode is used. The HY-2B Level-2 altimetry products (SGDR products) we used do not have OCOG data. Different re-trackers correspond to different flags in HY-2B L1 data. The data of OCOG re-tracking is not stored in SGDR products, because researchers only consider using it for ocean research when making SGDR products.

According to your valuable comments, we have conducted a statistic of the number of points within each grid cell when gridding the HY-2B radar freeboard, and have compared with CS-2 over the common area, as shown in Figure 1. As the data points of

the HY-2B SGDR products without OCOG re-track data is sparse, the number of points in HY-2B grids are generally less than CS-2. The mode of the data points within HY-2B grid cell is 5, and the mode of the data points within CS-2 grid cell is 104. The cumulative probability of measuring points greater than or equal to 15 within HY-2B grid cell is 58.3%.

[Figure]

Figure 1. A histogram of the number of points within each grid cell between HY-2B and CS-2 during two cycles (from October 2019 to April 2020 and from October 2020 to April 2021), bin=5.

**Your method of using 25 km segments and choosing leads based on the lowest 15 points within a segment still hasn't fully convinced me. As mentioned, several times throughout your study, this will have different impacts: higher freeboards during freeze-up, lower freeboards during spring, and a difference across first-year ice (FYI) and multi-year ice (MYI). So, essentially, you have a known bias due to your choice of methodology that you are not accounting for. I think it would be necessary to provide a measure (average, modal, median–take your pick) of how often this is the case, e.g., by providing a statistical value of how many points you usually have within a 25 km segment, as well as the standard deviation of this (or min-max range, quantiles–again, take your pick) to understand the variability. In truth, we do not know how often it is the case that your method will be impacted using the 15 lowest points.**

**Response:** Thank you for the valuable comments. We have conducted a statistic of the number of points within each 25 km segment, as shown in Figure 2. The mode of the data points within each 25 km segment is 5. The cumulative probability of measuring points greater than or equal to 15 within each 25 km segment is 43.4%. Meanwhile, we have supplied a histogram of the standard deviation of radar freeboard within each 25 km segment, as shown in Figure 3. The mode of the standard deviation within each 25 km segment is 0.11 m.

[Figure]

Figure 2. A histogram of the number of points within each 25 km segment during two cycles (from October 2019 to April 2020 and from October 2020 to April 2021), bin=1.

[Figure]

Figure 3. A histogram of the standard deviation of points within each 25 km segment during two cycles (from October 2019 to April 2020 and from October 2020 to April 2021).

**Furthermore, your method ensures a lead point every 25 km (if 15 points within a segment). Is this 'enough'? Here, I am considering the fact that HY-2B's footprint is 1.9 km across-track. In the study of Tilling et al. (2019), they compared CryoSat-2 and EnviSat, and saw that a lead-to-floe echo distance for EnviSat ranged from 0-20 km (average 11.3 km)–a satellite with a larger footprint, whereas for CryoSat-2 it ranged 0-4 km (mean of 1.0 km)–a satellite with comparable footprint. Somehow one lead observation every 25 km seems low.**

**Response:** Thank you for the valuable comments. We have used the average of the 15

lowest points as the sea surface height anomaly within each 25 km segment, but this does not necessitate that there is only one lead point within each 25km segment. This is just an estimation method, because there are too few data points in the HY-2B SGDR product to use conventional waveform characteristics to extract the true lead, so we have used the lowest point method to estimate sea surface height anomaly (the elevation of lead) within each 25 km segment. By comparing the lowest points of different values, it is found that the average of 15 lowest points have the best results. According to Figure 1, we have found the cumulative probability of measuring points greater than or equal to 15 within HY-2B grid cell is 58.3%. According to Figure 2, we have found the cumulative probability of measuring points greater than or equal to 15 within each 25 km segment is 43.4%. Therefore, it is enough for HY-2B SGDR product that at least contain 15 data points within 25 km segment to estimate sea surface height anomaly.

We have supplied the sentence in th revised manuscript. 'The cumulative probability of measuring points greater than or equal to 15 within each 25 km segment is 43.4%. Therefore, it is enough for HY-2B SGDR product that at least contain 15 data points within 25 km segment to estimate SSHA.' (Line 383)

**Figure 1 in response to Reviewer#1 suggests that something odd is going on due to the limited lead observations identified using the selected PP threshold, since you are simply using a nearest neighbor interpolation–which makes me wonder; how come you use this interpolation? Many other studies use either linear or cubic interpolations, making the interpolation dependent on several points rather than just one. Furthermore, do you have a limit on how far away points are allowed to be from a lead observation, e.g., 200 km, which other studies have required.**

**Response:** Thank you for the valuable comments. Firstly, when the spatical range of floe exceeds the spatical range of lead, the interpolation algorithms of matlab (cubic and linear) cannot interpolate. Secondly, we didn't have a limit on how far away points are allowed to be from a lead observation. Even if we have a limit on how far away points are allowed to be from a lead observation, e.g., 200 km, we cannot get the data through cubic and linear interpolations because the sptical range of floe exceeds the spatical range of lead within each 200 km segment. Compared with Figure 1 in response to Reviewer#1 (PP value greater than 10), we used PP value greater than 5 to obtain more SSHAs, as shown in Figure 4. We also compared the result of three interpolation algorithms (nearest, linear and cubic). All radar freeboard values interpolated by cubic and linear interpolation algorithms are nan.

[Figure]

Figure 4. A sample of the HY-2B elevation profile obtained for of track number 14418 on April 4, 2020. The green points in panel (a) are the relative elevation (h) values; the blue points in panel (a) are the h25km values, defined as the 25-km running mean of h; the black points in panel (b) are the modified relative elevation (hr) values; the red points in panel (b) are the sea surface height anomaly (SSHA) values **using nearest interpolation algorithm**; the azury points in panel (b) are lead points extracted by PP values greater 5; and the black points **in panel (c) are the radar freeboard values.**

**In response to GC2 for Reviewer#1, you present a comparison with TFMRA50 product–which is not available in the SGDR product. How come you have not used this sea-ice specific re-tracker for the study instead of the SGDR? Please provide some justification/explanation for the choice of this re-tracker in the manuscript.**

**Response:** Thank you for the valuable comments. Firstly, the range item in SGDR products has been re-tracked using the Brown model, so we did not re-tracked again using TFMRA when retrieving the radar freeboard. Secondly, we have used an implementation of the TFMRA to estimate the range to the main scattering horizon for each waveform based on SGDR product. But we didn't obtain satisfactory results, as shown in Figure 5. Finally, in response to GC2 for Reviewer#1, we have supplied a more reasonable result of **HY-2B L1 product** using TFMRA 50. We would like to explain that a more reasonable sea ice freeboard product can be obtained using L1 product instead of SGDR product. The processing chain of AWI (lead/floe discrimination, TFMRA50 re-tracker) provide a good reference for HY-2B L1 data. But it is difficult to obtain reasonable radar freeboard using SGDR products since it has sparse measurement points and the error of interpolation SSHA is larger than the method of 15 lowest points.

[Figure]

Figure 5. The retrieval of Arctic radar freeboard using TFMRA 50 and the average of 15 lowest points. Left: the spatial distribution of HY-2B radar freeboard in April 2019; Right: the histogram of HY-2B radar freeboard in April 2019.

**In response to GC2 from Reviewer#2, you conclude that the assumption of Ku-band penetrating to the snow-ice interface still holds based on former studies. However, more recent studies (Stroeve et al. 2020, 2022; Nab et al. 2022) have questioned this (with good reason). Since we do not currently have methods that can take into account this change of scattering horizon within the snowpack, I'd say the assumption is fair (and still widely used), however I do think that the topic warrants a discussion which is currently not given in the manuscript.**

**Response:** Thank you for the valuable comments. We have supplied the discussion of scattering horizon within the snowpack in the revised manuscript. We have revised 'We assumed that the radar pulses penetrate through any snow cover on floes and scatter from the snow-ice interface, which has been shown in laboratory experiments where the snow cover on sea ice is cold and dry (Beaven et al., 1995; Tilling et al., 2017). Despite some evidence that the scattering horizon migrates as temperature rises (Willatt et al., 2010), Tilling et al. (2017) did not observe any bias in their thickness retrieval when compared to year-round ice draft data, and so they thought that the impact of this effect was not significant.' to 'Several studies have found that radar freeboard uncertainty also pertains to inconsistent knowledge on how far the radar signal penetrates into the overlying snow cover (**Nandan et al., 2020; Willatt et al., 2011; Willatt et al., 2010; Drinkwater, 1995**). The general assumption is that the radar return primarily originates from the snow–sea ice interface at the Ku-band. While this may be applicable to cold, dry snow in a laboratory (**Beaven et al., 1995**), scientific evidence from observations and modeling indicates this assumption may not be valid even for a cold, homogeneous snowpack (**Nab et al., 2023; Nandan et al., 2020; Willatt et al., 2011; Willatt et al., 2010; Tonboe et al., 2010**). Morever, field campaigns have revealed that the dominant radar scattering actually occurs within the snowpack or at the snow surface rather than at the snow–ice interface (**Stroeve et al., 2020; Willatt et al., 2011; Willatt et al., 2010; Giles et al., 2007**). Since we do not currently have methods that can take into account this change of scattering horizon within the snowpack, we have assumed that the radar pulses

penetrate through any snow cover on ice floes and scatter from the snow-ice interface.' **(Line 220-230)**

**Also, based on several review comments, the uncertainty estimation procedure has been refined, and the results look more promising. However, when comparing with other studies (Ricker et al. 2014, Landy et al. 2020), the uncertainties estimated in this study (for both CS-2 and HY-2B) are in the lower range, which warrants a discussion. Also, consider separating your uncertainty estimates into ranges for MYI and FYI (like Ricker et al., 2014), since you also mention that you see different results depending on ice type.**

**Response:** Thank you for the valuable comments. The uncertainties of CS-2 are provided by the AWI sea ice thickness product. We have supplied a discussion in the revised manuscript. "However, the uncertainties estimated in this study for CS-2 and HY-2B are in the lower range when comparing with other studies (Ricker et al. 2014, Landy et al. 2020). This is because we just make a statistics of uncertainty over the common area for both CS-2 and HY-2B. Other studies do the statistics of CS-2 uncertainty with the upper limitation range of 88° N. In addition, Landy et al. (2020) also considered the following principal sources of systematic uncertainty: (i) partial wave penetration into the snowpack on MYI, for instance, due to metamorphic snow features; (ii) partial penetration into the snowpack on FYI, for instance due to brine wicking-induced snow basal salinity; and finally (iii) sea ice surface roughness. And they revealed sea ice surface roughness as a key overlooked feature of the conventional retrieval process (Landy et al. 2020). It is important to note that these key uncertainties limit the accuracy of the radar-based freeboard retrieval, which then propagate into the freeboard-to-thickness conversion." **(Line 436-444)**

We have supplied our uncertainty estimates for FYI and MYI, as shown in Table 1 and 2 (Table 9 and 10 in the revised manuscript). Here, we have used the monthly average grids of sea ice density and sea ice density uncertainty provided by AWI product. So we have updated the figure 12.

We have revised to '**from 0.61 m to 0.74 m**.' **(Line 22-23, 434-435 and 480)** We have revised to 'The sea ice thickness uncertainties over MYI are greater than over FYI for HY-2B and CS-2.' **(Line 426-427)** We have revised to 'Table 9: Mean sea ice freeboard uncertainties of HY-2B and CryoSat-2 on FYI, MYI and total sea ice.' **(Line 876)** We have also revised to 'Table 10: Mean sea ice thickness uncertainties of HY-2B and CryoSat-2 on FYI, MYI and total sea ice.' **(Line 876)**

Table 1. Mean sea ice freeboard uncertainties of HY-2B and CryoSat-2 on FYI, MYI and total sea ice.

| Unit: m | Oct 2019-April 2020 | | | | | | Oct 2020-April 2021 | | | | | |
| | HY-2B | | | CS-2 | | | HY-2B | | | CS-2 | | |
| | FYI | MYI | ALL | FYI | MYI | ALL | FYI | MYI | ALL | FYI | MYI | ALL |
|---|---|---|---|---|---|---|---|---|---|---|---|---|
| Oct | 0.025 | 0.028 | 0.027 | 0.026 | 0.028 | 0.028 | 0.021 | 0.027 | 0.025 | 0.026 | 0.028 | 0.028 |
| Nov | 0.019 | 0.028 | 0.022 | 0.021 | 0.027 | 0.023 | 0.018 | 0.026 | 0.022 | 0.020 | 0.026 | 0.023 |
| Dec | 0.020 | 0.030 | 0.023 | 0.021 | 0.028 | 0.023 | 0.018 | 0.029 | 0.022 | 0.020 | 0.028 | 0.023 |
| Jan | 0.019 | 0.029 | 0.022 | 0.021 | 0.028 | 0.023 | 0.018 | 0.028 | 0.021 | 0.020 | 0.027 | 0.022 |
| Feb | 0.021 | 0.033 | 0.024 | 0.022 | 0.030 | 0.024 | 0.019 | 0.032 | 0.023 | 0.021 | 0.030 | 0.024 |
| Mar | 0.022 | 0.036 | 0.025 | 0.023 | 0.033 | 0.025 | 0.021 | 0.036 | 0.025 | 0.022 | 0.033 | 0.025 |
| Apr | 0.023 | 0.037 | 0.025 | 0.022 | 0.033 | 0.024 | 0.022 | 0.039 | 0.025 | 0.022 | 0.034 | 0.024 |
| mean | 0.021 | 0.032 | 0.024 | 0.022 | 0.030 | 0.024 | 0.020 | 0.031 | 0.023 | 0.022 | 0.029 | 0.024 |

Table 2. Mean sea ice thickness uncertainties of HY-2B and CryoSat-2 on FYI, MYI and total sea ice.

| Unit: m | Oct 2019-April 2020 | | | | | | Oct 2020-April 2021 | | | | | |
| | HY-2B | | | CS-2 | | | HY-2B | | | CS-2 | | |
| | FYI | MYI | ALL | FYI | MYI | ALL | FYI | MYI | ALL | FYI | MYI | ALL |
|---|---|---|---|---|---|---|---|---|---|---|---|---|
| Oct | 0.81 | 0.58 | 0.67 | 0.45 | 0.51 | 0.49 | 0.80 | 0.56 | 0.61 | 0.46 | 0.47 | 0.47 |
| Nov | 0.64 | 0.68 | 0.65 | 0.44 | 0.50 | 0.46 | 0.62 | 0.66 | 0.64 | 0.40 | 0.45 | 0.42 |
| Dec | 0.69 | 0.73 | 0.70 | 0.44 | 0.51 | 0.46 | 0.61 | 0.72 | 0.65 | 0.47 | 0.54 | 0.49 |
| Jan | 0.66 | 0.76 | 0.69 | 0.48 | 0.54 | 0.50 | 0.63 | 0.73 | 0.66 | 0.51 | 0.54 | 0.52 |
| Feb | 0.71 | 0.80 | 0.73 | 0.55 | 0.58 | 0.56 | 0.67 | 0.77 | 0.70 | 0.57 | 0.58 | 0.57 |
| Mar | 0.71 | 0.88 | 0.74 | 0.63 | 0.65 | 0.63 | 0.70 | 0.84 | 0.73 | 0.63 | 0.64 | 0.63 |
| Apr | 0.71 | 0.88 | 0.73 | 0.68 | 0.77 | 0.69 | 0.68 | 0.87 | 0.71 | 0.68 | 0.72 | 0.69 |
| mean | 0.70 | 0.76 | 0.70 | 0.52 | 0.58 | 0.54 | 0.67 | 0.74 | 0.67 | 0.53 | 0.56 | 0.54 |

**Finally, I tried retrieving the data again. It was indeed necessary to create an account to access the FTP server, so I was not able to look at the data used for this study without creating a user (the user was not activated within deadline of this review). I suggest you write, in the data availability section, how to properly retrieve the data as you wrote in the response to reviews (especially since the website is not available with an English translation, thus limiting the potential user pool) to help other users get a hold of this data.**

**Response:** Thank you for the valuable comments. Please click the red button in the upper right corner to switch to the English interface, as shown in Figure 8. We have supplied the description of data availability in my revised manuscript (Line ). "If you haven't registered before, you'll need to create an account to access the FTP server at this website (https://osdds.nsoas.org.cn/register). Then, you can enter your account and password to log in to the official website to access the FTP folder with SDGR HY-2B data using filezilla (ftp://osdds-ftp.nsoas.org.cn/). The SGDR HY-2B data can also be accessed through https://osdds.nsoas.org.cn/MarineDynamic/." (Line 499-502)

[Figure]

Figure 8. The website of HY-2B satellite.

**Please check again that all references are properly written. E.g., line 144 (new manuscript) should be 'Tonboe et al. (2016)' not 'Rasmus et al. (2016)'.**

**Response:** Thank you for the valuable comments. We have revised to 'Tonboe et al. (2016)'(Line 150, 154, 509 and 717-718). We have revised to 'Aaboe et al. (2021)' (Line 158, 509 and 524-526). We also have checked all referneces through the manuscript. We supplied the some references (Line 556-557, 625-627, 645-647, 649-651, 699-702, 715-716, and 734-736).

**Specific comments (references to line numbers in the new manuscript)**

**I highly encourage the authors to look over the text again. With the introduction of a several new paragraphs, there are several places where it could benefit from some proofreading.**

**Response:** Thank you for the valuable comments. We have removed the sentences, 'Compared to conventional radar altimeters, CS-2 can achieve monthly observations of the Arctic with a coverage range of 88°N/S' (Line 112), 'It should be noted that the HY-2B SSHAs are slightly lower than those of the CS-2, the elevations of the floes are slightly higher than the CS-2, so the HY-2B radar freeboards are higher than those of the CS-2' (Line 300).
We have revised 'Fig. 10 shows comparisons of the HY-2B and CS-2 sea ice thicknesses with the IS-2 sea ice thicknesses from October 2019 to April 2020 and from October 2020 to April 2021, respectively.' to 'Fig. 10 shows comparisons of the HY-2B and CS-2 sea ice thicknesses with IS-2, respectively.' (Line 362-363)
We have revised 'the mean of these SSHA standard deviations' to 'the standard deviation of these SSHAs' (Line 394).

**Line 15-17. Your methodology has yet to be described, yet you mention specifics about how it is done. I suggest generalizing this sentence.**

Response: Thank you for the valuable comments. We have revised 'In spring, more of the lowest 15 points within 25 km segment are likely to originate from floes, while more points may originate from leads in early winter.' to 'A spring segment likely have more floe points than an early winter segment.' (Line 18) We also have supplied the description of methodology. 'Here, we derive radar freeboard by calculating the difference between the relative elevation of floe obtained by subtracting mean sea surface height (MSS) and sea surface height anomaly (SSHA) determined by an average of 15-lowest points method.' (Line 13-15)

**Line 17-20. Could you include a short sentence of how CryoSat-2 compare with OIB, for a perspective?**

Response: Thank you for the valuable comments. We have supplied the description of how CS-2 compare with OIB. 'The correlation between HY-2B (CS-2) sea ice freeboard retrievals and OIB values is 0.77 (0.84), with a root mean square error (RMSE) is 0.13 (0.10) m and a mean absolute error (MAE) is 0.12 (0.081) m. The correlation between HY-2B (CS-2) sea ice thickness retrievals and OIB values is 0.65 (0.80), with a RMSE is 1.86 (1.00) m and a MAE is 1.72 (0.75) m.' (Line 19-22)

**Line 22. I'm happy to see that the abstract has been shortened. Perhaps, you could include a short line on future work or current limitations for your work with HY-2B observations.**

Response: Thank you for the valuable comments. We have supplied the future work in the revised manuscript. 'We will reprocess the HY-2B L1 data to obtain more reliable polar sea ice thickness products.' (Line 24)

**Line 59. "corrections" -> "processing steps"**

Response: Thank you for the valuable comments. We have revised 'corrections' to 'processing steps'. (Line 61)

**Line 67. "The overall difference (…) AWI data is" -> "They noted the average difference (…) AWI data to be "**

Response: Thank you for the valuable comments. We have revised 'The overall difference between the HY-2B radar freeboard estimates and the AWI data is 0.088±0.057 m.' to 'They noted the average difference between Haiyang-2B (HY-2B) radar freeboard estimates and AWI data to be 0.088±0.057 m.' (Line 69-70)

**Line 69. "The radar freeboards are generally higher for HY-2B than CS-2" -> "They**

**generally observed higher radar freeboards for HY-2B than CS-2".**

**Response:** Thank you for the valuable comments. We have revised to 'They generally observed higher radar freeboards for HY-2B than CS-2.' (Line 70-71)

**Line 70. "(…), the HY-2B satellite can be used to observe polar sea ice" -> "(…), the feasibility of using the HY-2B satellite to map the polar sea ice must be explored". Please, also present limitations of your study (using already provided and re-tracked data rather than applying your own re-tracker etc.).**

**Response:** Thank you for the valuable comments. We have revised 'With the continuous development of China's Marine Dynamic Environment Satellite (Haiyang-2B, HY-2B), the HY-2B satellite can be used to observe polar sea ice' to 'With the continuous development of China's Marine Dynamic Environment Satellite, the feasibility of using the HY-2B satellite to map the polar sea ice must be explored.' (Line 72)

We have also supplied the limitation of this study. 'It is important to note in this study, however, that we are aiming to investigate whether HY-2B can be used for sea ice, but that we are limited to already provided higher-level (SGDR) product, and that it is not within the scope of the study to derive freeboard product using own re-tracker from the HY-2B SGDR product.' (Line 72-75).

**Line 84-85. Is "take into account the observations of sea ice" part of its main mission? I suggest rephrasing this sentence for clarity.**

**Response:** Thank you for the valuable comments. We have revised to 'Its main mission is to monitor and survey the marine environment, obtain a variety of marine dynamic environmental parameters, including sea surface winds, wave heights, sea surface heights, sea surface temperatures and other elements as well as the parameters of polar sea ice.' (Line 87-90)

**Line 88. Please provide a description of which stage the satellite is currently at and when the stages have started/ended.**

**Response:** Thank you for the valuable comments. The design life of HY-2B is 5 years. Generally, it will enter 168 days of repeated orbit in the later stage of operation. The specific time will be determined according to the satellite operation. We have supplied the description of the current stage of satellite. 'Currently, the repeat cycle of HY-2B is 14 days.' (Line 93)

**Line 90. Waveforms have not been mentioned yet. Consider rephrasing this for clarity.**

**Response:** Thank you for the valuable comments. We have removed this sentence. (Line 95)

**Line 96-97. Are the tracking modes named after the different re-trackers (OCOG, SMLE) or does HY-2B have to tracking modes based on surface, that tracks with different re-trackers? Consider rephrasing this, while also including a reference to what the "Brown" model is.**

**Response:** Thank you for the valuable comments. We have supplied a reference to explain Brown model. 'The SGDR products contain waveform data and have been re-tracked using the Brown model (Zhang et al., 2022).' (Line 101-102)
We have also revised 'In addition, the HY-2B altimeter uses two different tracking modes: suboptimal maximum likelihood estimation (SMLE) and offset center of gravity (OCOG). The two tacking modes can exchange according to the observation surfaces.' to 'The HY-2B altimeter will switch between suboptimal maximum likelihood estimation (SMLE) tracking mode and offset center of gravity (OCOG) tracking mode according to terrain changes. The SMLE tracking mode is suitable for areas with slower changes in terrain height, such as ocean and large areas of flat sea ice. The OCOG tracking mode is used for areas with dramatic changes in topographic height, such as land and sea ice areas.' (Line 102-105)

**Line 114. Include which version (baseline) of ESA CryoSat-2 product you are using.**

**Response:** Thank you for the valuable comments. We have revised to 'We mainly used the level-2 (L2) along-track data published by the ESA (processor baseline-D) and the monthly average products published by the AWI.' (Line 119-120)

**Line 196. MSS defined (by acronym) later than first used (used in MSS data section).**

**Response:** Thank you for the valuable comments. We had already defined the MSS in line 146. We have revised 'mean sea-surface (MSS)' to 'MSS'. (Line 199)

**Line 199. You mention residual error–be careful with terms like error/uncertainties. Consider using a different term. Also, this is the first time you are mentioning anything about this residual "error". Could you describe it more?**

**Response:** Thank you for the valuable comments. We have revised 'residual error' to 'estimation error' (Line 201, 203). We have revised 'The estimation error may be caused by the errors of orbit determination and different tracking algorithm.' to 'The estimation error does not include the modeled portion of the sea surface height, but includes all the unexplained static and time-varying components of the sea surface as well as noise introduced by our estimation process including the errors of orbit determination and different tracking algorithm (Kwok et al., 2007).' (Line 201-203)

**Line 207. Has SSHA been defined yet?**

**Response:** Thank you for the valuable comments. We had already defined 'SSHA' (Line

79).

**Line 209-210. "Since the (…)" -> consider rephrasing this or write the limitations in the introduction already, and then highlighting here, that you are simply using already provided elevations in the SGDR product.**

Response: Thank you for the valuable comments. We have supplied the limitation of this study. 'It is important to note in this study, however, that we are aiming to investigate whether HY-2B can be used for sea ice, but that we are limited to already provided higher-level (SGDR) product, and that it is not within the scope of the study to derive freeboard product using own re-tracker from the HY-2B SGDR product.' (line 72-75).
We have revised 'we did not re-track it again in this study.' to 'we are simply using the range terms from the satellite to the ground already provided in the SGDR product.' (Line 214-215)

**Line 232 (Section 3.2). Aren't these results? Consider moving them to the result section.**

Response: Thank you for the valuable comments. We have moved the section 3.2 to section 4.1 (Line 249-267). We have supplied the description. 'Firstly, we compared the parameters involved in the retrieval process with those in the CS-2 L2 along-track data released by the ESA.' (Line 244-245)
We have also revised to 'In this section, we described the sea ice thickness retrieval method applied for the SGDR data of the HY-2B pulse-limited radar altimeter in detail.' (Line 187-188) We have removed the '3.1 Retrieval process'.

**Line 241-242. "(…), which may have been caused by the fact that not all points within the 25 km segment are leads" -> "(…), which may have been caused by the fact that not all points used to estimate the SSHA within the 25 km segments originate from leads".**

Response: Thank you for the valuable comments. We have revised 'which may have been caused by the fact that not all points within the 25 km segment are leads' to 'which may have been caused by the fact that not all points used to estimate the SSHA within the 25 km segments originate from leads.' (Line 258-259)

**Line 248. "totally overlapped" -> "fully coincident"**

Response: Thank you for the valuable comments. We have revised 'totally overlapped' to 'fully coincident' (Line 266).

**Line 249. Could you provide the exact time difference and overall spatial difference? If the difference is significant (in hours), perhaps a measure of drift might be provided as well, to imply how big of an impact drift may have in this comparison.**

Response: Thank you for the valuable comments. The HY-2B track started to record at

18:05:48 on March 13, 2020, with the ending time at 18:58:02 on March 13, 2020. The spatial area of HY-2B track on Figure 4 (e) ranged from 97.7972 ° W-163.5410 ° W, 80.6271 ° N-63.3285 ° N. But the CS-2 track started to record at 15:10:52 on March 13, 2020, with the ending time at 15:20:54 on March 13, 2020. The spatial area of CS-2 track on Figure 4 (e) ranged from 123.0748 ° W-131.7198 ° W, 81.4975 ° N-70.5800 ° N. The time difference between HY-2B track and CS-2 track is as much as three hours. Considering the temporal differences between HY-2B track and CS-2 track indicates the sea ice drift likely impact the comparison between them. It is still a worthwhile question as to how big of an impact drift may have in this comparison.

**Line 308-310. Consider combining these two sentences ("Except (…)") in the sentence starting in line 307: "The HY2-B sea ice thicknesses are thicker (…)" for clarity. Also, throughout the text, be aware of too many repetitions of practically the same text.**

**Response:** Thank you for the valuable comments. We have revised 'These results are related to the accuracy of the extracted HY-2B SSHAs. Except in February, March and April, the monthly mean HY-2B sea ice thicknesses are thicker than AWI CS-2. The HY-2B modal thicknesses are thinner than AWI CS-2, except in December 2019, November 2020 and December 2020.' to 'The monthly mean sea ice thicknesses of HY-2B are thicker than CS-2 in early winter, while CS-2 sea ice thicknesses are greater than HY-2B in spring. The modal thicknesses of HY-2B are thinner than AWI CS-2, except in December 2019, November 2020 and December 2020. These results are related to the accuracy of the extracted HY-2B SSHAs.' (Line 318-321)

**Line 331. What underestimation? It can be seen on the figures but has not been described in the text so far. Please include it.**

**Response:** Thank you for the valuable comments. We have revised to 'The correlation between HY-2B and OIB is 0.65, with a RMSE of 1.86 m and a MAE of 1.72 m suggest that this underestimation of sea ice thickness could not only be attributed to sea ice freeboard but maybe also to snow depth or other parameters.' (Line 341-343)
We have also revised to 'The majority of the spread (shown by RMSE or MAE) in our HY-2B evaluation is caused by the underestimation of thickness over thick ice, which may have been caused by the fact that not all points used to estimate the SSHA within the 25 km segments originate from leads.' (Line 344-346)

**Line 334. "The majority of the spread" -> "The majority of the spread (shown by RMSE or MAE)"–or however you see this spread, but link to it.**

**Response:** Thank you for the valuable comments. We have revised to 'The majority of the spread (shown by RMSE or MAE) in our HY-2B evaluation is caused by the underestimation of thickness over thick ice, which may have been caused by the fact that not all points used to estimate the SSHA within the 25 km segments originate from leads.' (Line 344-346)

**Line 339. The IS-2 snow freeboard is not subtracted from the AWI snow depths to obtain the sea ice freeboard. The AWI snow depths are subtracted from the IS-2 snow freeboards to obtain the sea ice freeboard. Please correct.**

**Response:** Thank you for the valuable comments. We have revised to 'The AWI snow depths are subtracted from the IS-2 snow freeboards to obtain the sea ice freeboards.' (Line 351-352)

**Line 342. Add which section this slower wave propagation correction has already been explained in, in the end of the sentence -> (see Section …).**

**Response:** Thank you for the valuable comments. We have revised to 'To compare these values with IS-2 sea ice freeboard, we use AWI snow depth to perform a wave propagation speed correction for HY-2B and AWI CS-2 radar freeboard (see Section 3).' (Line 352-353)

**Line 348. "In addition (…)"–consider rephrasing this sentence for clarity.**

**Response:** Thank you for the valuable comments. We have revised 'In addition, the differences of measurement mode and footprint size maybe result the discrepancies between HY-2B and IS-2.' to 'In addition, the differences between HY-2B and IS-2 may be caused by inconsistent measurement modes and footprint sizes.' (Line 359-360)

**Line 367. What is meant by a "larger" SSHA? I suggest rephrasing for clarity.**

**Response:** Thank you for the valuable comments. We have revised 'As the table shows (Schemes 1-8), the mean deviation and MAE values first decrease and then increase with the gradual increase in SSHA, indicating that a larger SSHA does not necessitate a smaller mean deviation or MAE.' to 'As the table shows (Schemes 1-8), the mean deviation and MAE values first decrease and then increase with the gradual increase in SSHA, indicating that an increase in SSHA does not necessitate a linear reduction in mean deviation or MAE.' (Line 378)

**Line 434. Similar to the changes for the abstract, consider generalizing this sentence since you are talking specifics about a methodology that has not been described in your conclusion yet.**

**Response:** Thank you for the valuable comments. We have revised 'In spring, more of the lowest 15 points within 25 km segment are likely to originate from floes, while more points may originate from leads in early winter.' to 'A spring segment likely have more floe points than an early winter segment.' (Line 453-454)

References

Landy, J. C., Petty, A. A., Tsamados, M., & Stroeve, J. C. (2020). Sea ice roughness overlooked as a key source of uncertainty in CryoSat-2 ice freeboard retrievals. Journal of Geophysical Research: Oceans, 125, e2019JC015820. https://doi.org/10.1029/2019JC015820

Nab, C., Mallett, R., Gregory, W., Landy, J., Lawrence, I., Willatt, R., Stroeve, J., Tsamados, T.: Synoptic variability in satellite altimeter-derived radar freeboard of Arctic sea ice. Geophysical Research Letters, 50, e2022GL100696, https://doi.org/10.1029/2022GL100696, 2023.

Ricker, R., Hendricks, S., Helm, V., Skourup, H., and Davidson, M.: Sensitivity of CryoSat-2 Arctic sea-ice freeboard and thickness on radar-waveform interpretation, The Cryosphere, 8, 1607–1622, https://doi.org/10.5194/tc-8-1607-2014, 2014.

Stroeve, J., Nandan, V., Willatt, R., Dadic, R., Rostosky, P., Gallagher, M., Mallett, R., Barrett, A., Hendricks, S., Tonboe, R., McCrystall, M., Serreze, M., Thielke, L., Spreen, G., Newman, T., Yackel, J., Ricker, R., Tsamados, M., Macfarlane, A., Hannula, H.-R., and Schneebeli, M.: Rain on snow (ROS) understudied in sea ice remote sensing: a multi-sensor analysis of ROS during MOSAiC (Multidisciplinary drifting Observatory for the Study of Arctic Climate), The Cryosphere, 16, 4223–4250, https://doi.org/10.5194/tc-16-4223-2022, 2022.

Stroeve, J., Nandan, V., Willatt, R., Tonboe, R., Hendricks, S., Ricker, R., Mead, J., Mallett, R., Huntemann, M., Itkin, P., Schneebeli, M., Krampe, D., Spreen, G., Wilkinson, J., Matero, I., Hoppmann, M., and Tsamados, M.: Surface-based Ku- and Ka-band polarimetric radar for sea ice studies , The Cryosphere, 14, 4405–4426, https://doi.org/10.5194/tc-14-4405-2020, 2020.

Tilling, R., Ridout, A., & Shepherd, A. (2019). Assessing the impact of lead and floe sampling on Arctic sea ice thickness estimates from Envisat and CryoSat-2. Journal of Geophysical Research: Oceans, 124, 7473– 7485. https://doi.org/10.1029/2019JC015232

---

## Author Response (AR3)

Dear Editors and Reviewers,

The authors, above all, would like to thank the editor and the reviewers for their comments to help to improve the manuscript. We have now carefully reviewed and addressed all of comments which we hope meet with approval, with revisions to the manuscript shown in red. The primary corrections in the paper and the responds to the reviewer's comments are as flowing:

**Responses to Reviewer's Comments:**

**Reviewer #1:**

I'd once again like to thank the authors for their diligent response to all the reviews. This is my third time reviewing this paper, and I am happy to see that many of my concerns have been addressed in the current version of the manuscript.

The authors have now included statistics (cumulative probability) on how many points are used pr grid cell, clearly highlighting the limited coverage of HY-2B with the re-tracker provided in higher level products. They have also provided statistics (cumulative probability) on how often 25 km segments include 15 points or more. They have also included a discussion on their uncertainty estimates in relation to other studies, as well as provided a distinction in uncertainty estimates from multi-year ice and first-year ice.

Overall, I am happy with the changes that the authors have made based on my comments and within the scope of their study. I have a few comments, which are minor, but nonetheless should be considered by the authors to strengthen their publication in comparison to recently published studies.

**General comments**
**During the review process of this paper, it has come to my attention that a new paper (Jiang et al., 2023) with similar motivation/objectives and with a, to a large degree, similar processing chain has been published, although they have included more products in their comparison. I think this study now requires a short section or paragraph on how this study differs from the already published study. Here, I especially encourage a discussion on how they use the lowest 3 points in their study to estimate the SSHA with the argumentation that one other study on Envisat observation has been published using this and how this is often applied to laser altimetry, as well as their use of a different re-tracker and geophysical corrections (maybe, the results on DTU21 MSS represented in an earlier response to the reviewers of this study could be included to aid this discussion). You might want to have a look at their reviewer reports/response, where this has also been questioned. However, I believe it is imperative that the authors clearly highlight the differences, and present why the results of this study are still relevant.**

**Response:** Thank you for the valuable comments. We have supplied the description of Jiang et al. (2023) in the intruduction. 'Therefore, Jiang et al. (2023) used the AWI CS-2 sea ice thickness products to calibrate the HY-2B thickness estimates.' (Line 73-74)

We have supplied the description of the difference between this study and Jiang et al. (2023), as well as the relevance of this study in the discussion. 'The discrepancies between this study and Jiang et al. (2023) are mainly due to retrieval methods and data sources. The discrepancies of the methods are reflected in the re-tracking method, the estimation method of SSHA and whether the subsequent results need to be calibrated with AWI CS-2. The discrepancies of the datas are reflected in product levels of HY-2B and DTU MSS models. Jiang et al. (2023) used the lowest 3 points per 25 km to estimate SSHA with HY-2B L1 product, resulting the retrieval of sea ice thickness is thicker than AWI CS-2. So the retrieval of sea ice thickness need to be calibrated with AWI CS-2. It is worth noting that this study uses SGDR data, which only includes the SMLE re-tracking data. We don't deny that the L1 data Jiang et al. (2023) used is much more extensive in the Arctic. In this study, we try to explore the application of SGDR data released to the public in polar sea ice, but it can be seen from our study that it seems difficult to obtain reasonable results by using conventional methods. So we use 15 lowest points per 25 km to estimate SSHA to retrieve more reasonable Arctic radar freeboard and thickness. Through this study, we can see that the relative surface height after substracting MSS is relatively low compared with CS-2, which may be caused by the re-tracking algorithm and precision orbit determination. This is what we need to avoid when reprocessing HY-2B L1 data, which also provides reference for reprocessing L1 data.' (Line 491-503)

We have also revised 'We will reprocess the HY-2B L1 data to obtain more reliable polar sea ice thickness products.' to 'We hope to release products that are more reasonable and suitable for polar sea ice thickness retrieval, so as to better evaluate the potential application of HY-2B in polar sea ice'. (Line 512-513)

**The authors mention in response to the reviewers, that their estimation method of 15 points within the segment yielding one lead point does not necessitate that there is only one lead along the segment. This is inherently clear, however that was not the concern raised in that question. What is important here - especially considering that the interpolation of lead points into floes are done by nearest neighbor-is the accuracy of that lead estimation point every 25 km segment. This has already been partly discussed with the cumulative probability statistics and the analysis you made with using a combination of numbers of points used to estimate SSHA. But, what was questioned previously was: is one point pr. 25 km enough to represent the SSHA, especially if that point could be affected by floe points in the estimation? Showing the distance between leads identified for other missions was to feed into the discussion of whether the estimation method of SSHA seems reasonable, and whether one SSHA pr. 25 km (since this is your estimation method) is accurate enough. I'll leave it up to the authors to consider if more discussion is relevant here, but I strongly suggest the authors to reflect upon this-especially in relation to the recently published study, that used 3 points along the segment.**

**Response:** Thank you for the valuable comments. Actually, one point per 25 km isn't

enough to represent the SSHA, but currently we have tried other methods to retrieve radar freeboard, and we didn't obtain a satifactory result. This is the best result we can get so far. As the reviewer noted in his previous comments, Tilling et al. (2019) saw that a lead-to-floe echo distance for EnviSat ranged from 0-20 km (average 11.3 km). So there might be two or three leads within 25 km segment according to Tilling et al. (2019). However, it should be made clear that SGDR products have been re-tracking by Brown model, as well as the error of precision orbit determination, resulting in the relative surface elevations obtained by SGDR are lower than AWI CS-2 (Table 2, Figure 3 and Figure 4). The average of 3 lowest points within 25 km segment are lower than the average of 15 lowest points within 25 km segment, resulting in a higher radar freeboard retrieval. So we use the average of 15 lowest points within 25 km segment to estimate the SSHA. Although it is inconsistent with Tilling et al. (2019), the average of 15 lowest points within 25 km segment is more suitable to obtain reasonable radar freeboard than the average of 3 lowest points for the lower relative surface elevations. This method in this study is maybe only applicable to SGDR data and is not the mainstream method. We also pointed out in the paper that we hope to use HY-2B L1 data in the future to classify lead and floe to obtain more reasonable radar freeboard and sea ice thickness.

**With the introduction of new sentences, there are several places where a read-through would be beneficial. I am aware that this comment was also given the last time, but it is simply to ensure the readability of the manuscript, and since there have been a significant number of new/edited sentences, I encourage the authors to have a look at them again and see if they can improve on this.**

**Response:** Thank you for the valuable comments. We have read and checked the whole manuscript. We have added the reference of Jiang et al. (2023). (Line 605-606)

**Specific comments**

**Line 24: "we will …" -> I suggest rephrasing this sentence. Somehow it sounds like your not happy with your results. Perhaps mention that future work will include reprocessing the data with a dedicated sea ice re-tracker, and that a further investigation of using the radar waveforms directly to identify leads will be the next steps.**

**Response:** Thank you for the valuable comments. We have revised 'We will reprocess the HY-2B L1 data to obtain more reliable polar sea ice thickness products' to 'The future work will include reprocessing the HY-2B L1 data with a dedicated sea ice re-tracker, and using the radar waveforms to directly identify leads to release products that are more reasonable and suitable for polar sea ice thickness retrieval' (Line 24-26).

**Line 254: In the review comments you mention that the satellites passes with about 3 hour difference, but the way it is written here sounds like almost 2 weeks of difference. Could you please clarify that you are comparison two instances, and that for each date denote the time differences between HY-2B and CS-2 acquisition.**

**Response:** Thank you for the valuable comments. We have revised 'Fig. 4 (a) and (e) show the orbit positions of HY-2B and CS-2 obtained on April 4, 2020, and March 13,

2020, covering the Beaufort Sea and the northern Canadian Archipelago, respectively, to compare the relative surface elevation, SSHA, and radar freeboard estimates' to 'We selected two instances of different time for comparison acquired on April 4, 2020 and March 13, 2020, respectively, as shown in Fig.4 (a) and (e). For each date denote the time of CS-2 and HY-2B tracks. Both of them cover the Beaufort Sea and the northern Canadian Archipelago'(Line 255-257).

**References**

Jiang, M.; Zhong, W.; Xu, K.; Jia, Y. Estimation of Arctic Sea Ice Thickness from Chinese HY-2B Radar Altimetry Data. Remote Sens. 2023, 15, 1180. https://doi.org/10.3390/rs15051180